# BiAdam: Fast Adaptive Bilevel Optimization Methods

## Abstract

Bilevel optimization recently has attracted increased interest in machine learning due to its many applications such as hyper-parameter optimization and meta learning. Although many bilevel optimization methods recently have been proposed, these methods do not consider using adaptive learning rates. It is well known that adaptive learning rates can accelerate many optimization algorithms including (stochastic) gradient-based algorithms. To fill this gap, in the paper, we propose a novel fast adaptive bilevel framework for solving bilevel optimization problems that the outer problem is possibly nonconvex and the inner problem is strongly convex. Our framework uses unified adaptive matrices including many types of adaptive learning rates, and can flexibly use the momentum and variance reduced techniques. In particular, we provide a useful convergence analysis framework for the bilevel optimization. Specifically, we propose a fast single-loop adaptive bilevel optimization (BiAdam) algorithm based on the basic momentum technique, which achieves a sample complexity of $\tilde{O}(\epsilon^{-4})$ for finding an $\epsilon$-stationary point. Meanwhile, we propose an accelerated version of BiAdam algorithm (VR-BiAdam) by using variance reduced technique, which reaches the best known sample complexity of $\tilde{O}(\epsilon^{-3})$ without relying on large batch-size. To the best of our knowledge, we first study the adaptive bilevel optimization methods with adaptive learning rates. Some experimental results on data hyper-cleaning and hyper-representation learning tasks demonstrate the efficiency of the proposed algorithms.

## 1 Introduction

Bilevel optimization is known as a class of popular hierarchical optimization, which has been applied to a wide range of machine learning problems such as hyperparameter optimization Shaban et al. (2019), meta-learning Ji et al. (2021); Liu et al. (2021a) and policy optimization Hong et al. (2020). In the paper, we consider solving the following stochastic bilevel optimization problem, defined as

$$\min_{x \in \mathcal{X}} F(x) := \mathbb{E}_{\xi \sim \mathcal{D}}\Big[f\big(x, y^*(x); \xi\big)\Big] \qquad \text{(Outer)} \qquad (1)$$

$$\text{s.t. } y^*(x) \in \arg\min_{y \in \mathcal{Y}} \mathbb{E}_{\zeta \sim \mathcal{M}}\Big[g(x, y; \zeta)\Big], \qquad \text{(Inner)} \qquad (2)$$

where $F(x) = f(x, y^*(x)) = \mathbb{E}_{\xi}\big[f(x, y^*(x); \xi)\big]$ is a differentiable and possibly nonconvex function, and $g(x, y) = \mathbb{E}_{\zeta}\big[g(x, y; \zeta)\big]$ is a differentiable and strongly convex function in variable $y$, and $\xi$ and $\zeta$ are random variables follow unknown distributions $\mathcal{D}$ and $\mathcal{M}$, respectively. Here $\mathcal{X} \subseteq \mathbb{R}^d$ and $\mathcal{Y} \subseteq \mathbb{R}^p$ are convex closed sets. Problem (1) involves many machine learning problems with a hierarchical structure, which include hyper-parameter optimization Franceschi et al. (2018), meta-learning Franceschi et al. (2018), policy optimization Hong et al. (2020) and neural network architecture search Liu et al. (2018).

Since bilevel optimization has been widely applied in machine learning, some works recently have been begun to study the bilevel optimization. For example, Ghadimi & Wang (2018); Ji et al. (2021) proposed a class of double-loop methods to solve the problem (1). However, to obtain an accurate estimate, the BSA in Ghadimi & Wang (2018) needs to solve the inner problem to a high accuracy, and the stocBiO in Ji et al. (2021) requires large batch-sizes in solving the inner problem.

Table 1: Sample complexity of the representative bilevel optimization methods for finding an $\epsilon$-stationary point of the bilevel problem (1), i.e., $\mathbb{E}\|\nabla F(x)\| \leq \epsilon$ or its equivalent variants. **BSize** denotes mini-batch size; **ALR** denotes adaptive learning rate. $C(x,y)$ denotes the constraint sets in $x$ and $y$, where **Y** denotes the fact that there exists a convex constraint on variable, otherwise is **N**. **DD** denotes dimension dependence in the gradient estimators, and $p$ denotes the dimension of variable $y$. **1** denotes Lipschitz continuous of $\nabla_x f(x,y;\xi)$, $\nabla_y f(x,y;\xi)$, $\nabla_y g(x,y;\zeta)$, $\nabla^2_{xy} g(x,y;\zeta)$ and $\nabla^2_{yy} g(x,y;\zeta)$ for all $\xi, \zeta$; **2** denotes Lipschitz continuous of $\nabla_x f(x,y)$, $\nabla_y f(x,y)$, $\nabla_y g(x,y)$, $\nabla^2_{xy} g(x,y)$ and $\nabla^2_{yy} g(x,y)$; **3** denotes bounded stochastic partial derivatives $\nabla_y f(x,y;\xi)$ and $\nabla^2_{xy} g(x,y;\zeta)$; **4** denotes bounded stochastic partial derivatives $\nabla_x f(x,y;\xi)$, and $\nabla^2_{yy} g(x,y;\zeta)$; **5** denotes the bounded true partial derivatives $\nabla_y f(x,y)$ and $\nabla^2_{xy} g(x,y)$; **6** denotes Lipschitz continuous of function $f(x,y;\xi)$; **7** denotes $g(x,y;\zeta)$ is $L_g$-smooth and $\mu$-strongly convex function *w.r.t.* $y$ for all $\zeta$; **8** denotes $g(x,y)$ is $L_g$-smooth and $\mu$-strongly convex function *w.r.t.* $y$.

| Algorithm | Reference | Complexity | BSize | Loop | $C(x,y)$ | DD | ALR | Conditions |
|---|---|---|---|---|---|---|---|---|
| BSA | Ghadimi & Wang (2018) | $O(\epsilon^{-6})$ | $O(1)$ | Double | Y, N | $p^2$ | | **2, 5, 7** |
| TTSA | Hong et al. (2020) | $\tilde{O}(\epsilon^{-5})$ | $O(1)$ | Single | Y, N | $p^2$ | | **1, 3, 7** |
| stocBiO | Ji et al. (2021) | $O(\epsilon^{-4})$ | $\tilde{O}(\epsilon^{-2})$ | Double | N, N | $p^2$ | | **1, 6, 7** |
| STABLE | Chen et al. (2022) | $\tilde{O}(\epsilon^{-4})$ | $O(1)$ | Single | N, N | $p^3$ | | **1, 3, 4, 8** |
| SMB | Guo et al. (2021b) | $\tilde{O}(\epsilon^{-4})$ | $\tilde{O}(1)$ | Single | N, Y | $p^2$ | | **2, 5, 7** |
| SUSTAIN | Khanduri et al. (2021) | $\tilde{O}(\epsilon^{-3})$ | $\tilde{O}(1)$ | Single | N, N | $p^2$ | | **1, 3, 7** |
| SVRB | Guo & Yang (2021) | $\tilde{O}(\epsilon^{-3})$ | $O(1)$ | Single | N, N | $p^3$ | | **1, 5, 8** |
| MRBO | Yang et al. (2021) | $\tilde{O}(\epsilon^{-3})$ | $\tilde{O}(1)$ | Single | N, N | $p^2$ | | **1, 6, 7** |
| VRBO | Yang et al. (2021) | $\tilde{O}(\epsilon^{-3})$ | $\tilde{O}(\epsilon^{-2})$ | Double | N, N | $p^2$ | | **1, 6, 7** |
| BiAdam | Ours | $\tilde{O}(\epsilon^{-4})$ | $\tilde{O}(1)$ | Single | Y/N, Y/N | $p^2$ | $\checkmark$ | **2, 5, 7** |
| VR-BiAdam | Ours | $\tilde{O}(\epsilon^{-3})$ | $\tilde{O}(1)$ | Single | Y/N, Y/N | $p^2$ | $\checkmark$ | **1, 5, 7** |

Hong et al. (2020) proposed a class of single-loop methods to solve the bilevel problems. Subsequently, Khanduri et al. (2021); Guo & Yang (2021); Yang et al. (2021); Chen et al. (2022) presented some accelerated single-loop methods by using the momentum-based variance reduced technique of STORM Cutkosky & Orabona (2019). More recently, Dagréou et al. (2022) developed a novel framework for bilevel optimization based on the linear system, and proposed a fast SABA algorithm for finite-sum bilevel problems based on the varaince reduced technique of SAGA (Defazio et al., 2014). Although these methods can effectively solve the bilevel problems, they do not consider using the adaptive learning rates and only consider the bilevel problems under unconstrained setting. Since using generally different learning rates for the inner and outer problems to ensure the convergence of bilevel optimization problems, we will consider using different adaptive learning rates for the inner and outer problems with convergence guarantee. Clearly, this can not follow the exiting adaptive methods for single-level problems. Thus, there exists a natural question:

> **How to design the effective optimization methods with adaptive learning rates for the bilevel problems ?**

In the paper, we provide an affirmative answer to this question and propose a class of fast single-loop adaptive bilevel optimization methods based on unified adaptive matrices, which including many types of adaptive learning rates. Moreover, our framework can flexibly use the momentum and variance reduced techniques. Our main **contributions** are summarized as follows:

1) We propose a fast single-loop adaptive bilevel optimization algorithm (BiAdam) based on the basic momentum technique, which achieves a sample complexity of $\tilde{O}(\epsilon^{-4})$ for finding an $\epsilon$-stationary point.

2) Meanwhile, we propose a single-loop accelerated version of BiAdam algorithm (VR-BiAdam) by using the momentum-based variance reduced technique, which reaches the best known sample complexity of $\tilde{O}(\epsilon^{-3})$.

3) Moreover, we provide a useful convergence analysis framework for both the constrained and unconstrained bilevel programming under some mild conditions (Please see Table 1).

4) The experimental results on hyper-parameter learning demonstrate the efficiency of the proposed algorithms.

## 2 PRELIMINARIES

### 2.1 NOTATIONS

$\mathcal{U}\{1, 2, \cdots, K\}$ denotes a uniform distribution over a discrete set $\{1, 2, \cdots, K\}$. $\|\cdot\|$ denotes the $\ell_2$ norm for vectors and spectral norm for matrices. $\langle x, y \rangle$ denotes the inner product of two vectors $x$ and $y$. For vectors $x$ and $y$, $x^r$ $(r > 0)$ denotes the element-wise power operation, $x/y$ denotes the element-wise division and $\max(x, y)$ denotes the element-wise maximum. $I_d$ denotes a $d$-dimensional identity matrix. $A \succ 0$ denotes that the matrix $A$ is positive definite. Given function $f(x, y)$, $f(x, \cdot)$ denotes function *w.r.t.* the second variable with fixing $x$, and $f(\cdot, y)$ denotes function *w.r.t.* the first variable with fixing $y$. $a = O(b)$ denotes that $a \leq Cb$ for some constant $C > 0$. The notation $\tilde{O}(\cdot)$ hides logarithmic terms. Given a convex closed set $\mathcal{X}$, we define a projection operation to $\mathcal{X}$ as $\mathcal{P}_{\mathcal{X}}(z) = \arg\min_{x \in \mathcal{X}} \frac{1}{2}\|x - z\|^2$.

### 2.2 SOME MILD ASSUMPTIONS

In this subsection, we give some mild assumptions on the problem (1).

**Assumption 1.** *For any $x$ and $\zeta$, $g(x, y; \zeta)$ is $L_g$-smooth and $\mu$-strongly convex function, i.e., $L_g I_p \succeq \nabla^2_{yy} g(x, y; \zeta) \succeq \mu I_p$.*

**Assumption 2.** *For functions $f(x, y)$ and $g(x, y)$ for all $x \in \mathcal{X}$ and $y \in \mathcal{Y}$, we assume the following conditions hold: $\nabla_x f(x, y)$ and $\nabla_y f(x, y)$ are $L_f$-Lipschitz continuous, $\nabla_y g(x, y)$ is $L_g$-Lipschitz continuous, $\nabla^2_{xy} g(x, y)$ is $L_{gxy}$-Lipschitz continuous, $\nabla^2_{yy} g(x, y)$ is $L_{gyy}$-Lipschitz continuous. For example, for all $x, x_1, x_2 \in \mathcal{X}$ and $y, y_1, y_2 \in \mathcal{Y}$, we have*

$$\|\nabla_x f(x_1, y) - \nabla_x f(x_2, y)\| \leq L_f \|x_1 - x_2\|, \quad \|\nabla_x f(x, y_1) - \nabla_x f(x, y_2)\| \leq L_f \|y_1 - y_2\|.$$

**Assumption 3.** *For functions $f(x, y; \xi)$ and $g(x, y; \zeta)$ for all $x \in \mathcal{X}$, $y \in \mathcal{Y}$, $\xi$ and $\zeta$, we assume the following conditions hold: $\nabla_x f(x, y; \xi)$ and $\nabla_y f(x, y; \xi)$ are $L_f$-Lipschitz continuous, $\nabla_y g(x, y; \zeta)$ is $L_g$-Lipschitz continuous, $\nabla^2_{xy} g(x, y; \zeta)$ is $L_{gxy}$-Lipschitz continuous, $\nabla^2_{yy} g(x, y; \zeta)$ is $L_{gyy}$-Lipschitz continuous. For example, for all $x, x_1, x_2 \in \mathcal{X}$ and $y, y_1, y_2 \in \mathcal{Y}$, we have*

$$\|\nabla_x f(x_1, y; \xi) - \nabla_x f(x_2, y; \xi)\| \leq L_f \|x_1 - x_2\|, \quad \|\nabla_x f(x, y_1; \xi) - \nabla_x f(x, y_2; \xi)\| \leq L_f \|y_1 - y_2\|.$$

**Assumption 4.** *The partial derivatives $\nabla_y f(x, y)$ and $\nabla^2_{xy} g(x, y)$ are bounded, i.e., $\|\nabla_y f(x, y)\|^2 \leq C^2_{fy}$ and $\|\nabla^2_{xy} g(x, y)\|^2 \leq C^2_{gxy}$.*

**Assumption 5.** *Stochastic functions $f(x, y; \xi)$ and $g(x, y; \zeta)$ have unbiased stochastic partial derivatives with bounded variance, e.g.,*

$$\mathbb{E}[\nabla_x f(x, y; \xi)] = \nabla_x f(x, y), \quad \mathbb{E}\|\nabla_x f(x, y; \xi) - \nabla_x f(x, y)\|^2 \leq \sigma^2.$$

*The same assumptions hold for $\nabla_y f(x, y; \xi)$, $\nabla_y g(x, y; \zeta)$, $\nabla^2_{xy} g(x, y; \zeta)$ and $\nabla^2_{yy} g(x, y; \zeta)$.*

Assumptions 1-5 are commonly used in stochastic bilevel optimization problems Ghadimi & Wang (2018); Hong et al. (2020); Ji et al. (2021); Chen et al. (2022); Khanduri et al. (2021). Note that Assumption 3 is clearly stricter than Assumption 2. For example, given Assumption 3, we have $\|\nabla_x f(x_1, y) - \nabla_x f(x_2, y)\| = \|\mathbb{E}[\nabla_x f(x_1, y; \xi) - \nabla_x f(x_2, y; \xi)]\| \leq \mathbb{E}\|\nabla_x f(x_1, y; \xi) - \nabla_x f(x_2, y; \xi)\| \leq L_f\|x_1 - x_2\|\|$ for any $x, y, \xi$. At the same time, based on Assumptions 4-5, we also have $\|\nabla_y f(x, y; \xi)\|^2 = \|\nabla_y f(x, y; \xi) - \nabla_y f(x, y) - \nabla_y f(x, y)\|^2 \leq 2\|\nabla_y f(x, y; \xi) - \nabla_y f(x, y)\|^2 + 2\|\nabla_y f(x, y)\|^2 \leq 2\sigma^2 + 2C^2_{fy}$ and $\|\nabla^2_{xy} g(x, y; \zeta)\|^2 \leq 2\sigma^2 + 2C^2_{gxy}$. Thus we argue that under Assumption 5, the bounded $\nabla_y f(x, y)$ and $\nabla^2_{xy} g(x, y)$ are not milder than the bounded $\nabla_y f(x, y; \xi)$ and $\nabla^2_{xy} g(x, y; \zeta)$ for all $\xi$ and $\zeta$.

### 2.3 BILEVEL OPTIMIZATION

In this subsection, we review the basic first-order method for solving the problem (1). Naturally, we give the following iteration to update the variables $x, y$: at the $t$-th step

$$y_{t+1} = \mathcal{P}_{\mathcal{Y}}\Big(y_t - \lambda \nabla_y g(x_t, y_t)\Big), \quad x_{t+1} = \mathcal{P}_{\mathcal{X}}\Big(x_t - \gamma \nabla_x f(x_t, y^*(x_t))\Big),$$

where $\lambda > 0$ and $\gamma > 0$ denote the step sizes. Clearly, if there does not exist a closed form solution of the inner problem in the problem (1), i.e., $y_{t+1} \neq y^*(x_t)$, we can not easily obtain the gradient $\nabla F(x_t) = \nabla f(x_t, y^*(x_t))$. Thus, one of key points in solving the problem (1) is to estimate the gradient $\nabla F(x_t)$.

**Lemma 1.** *(Lemma 2.1 in Ghadimi & Wang (2018)) Under the above Assumption 2, we have, for any $x \in \mathcal{X}$*

$$\nabla F(x) = \nabla_x f(x, y^*(x)) - \nabla^2_{xy} g(x, y^*(x)) \left[ \nabla^2_{yy} g(x, y^*(x)) \right]^{-1} \nabla_y f(x, y^*(x)).$$

From the above Lemma 1, it is natural to use the following form to estimate $\nabla F(x)$, defined as,

$$\bar{\nabla} f(x, y) = \nabla_x f(x, y) - \nabla^2_{xy} g(x, y) \left( \nabla^2_{yy} g(x, y) \right)^{-1} \nabla_y f(x, y), \ \forall x \in \mathcal{X}, y \in \mathcal{Y}$$

Note that although the inner problem of the problem (1) is a constrained optimization, we assume that the optimal condition of the inner problem still is $\nabla_y g(x, y^*(x)) = 0$ and $y^*(x) \in \mathcal{Y}$.

**Lemma 2.** *(Lemma 2.2 in Ghadimi & Wang (2018)) Under the above Assumptions (1, 2, 4), for all $x, x_1, x_2 \in \mathcal{X}$ and $y \in \mathcal{Y}$, we have*

$$\|\bar{\nabla} f(x, y) - \nabla F(x)\| \leq L_y \|y^*(x) - y\|, \ \|y^*(x_1) - y^*(x_2)\| \leq \kappa \|x_1 - x_2\|,$$
$$\|\nabla F(x_1) - \nabla F(x_2)\| \leq L \|x_1 - x_2\|,$$

*where $L_y = L_f + L_f C_{gxy}/\mu + C_{fy}\big(L_{gxy}/\mu + L_{gyy}C_{gxy}/\mu^2\big)$, $\kappa = C_{gxy}/\mu$, and $L = L_f + (L_f + L_y)C_{gxy}/\mu + C_{fy}\big(L_{gxy}/\mu + L_{gyy}C_{gxy}/\mu^2\big)$.*

For the stochastic bilevel optimization, Yang et al. (2021); Hong et al. (2020) provided a stochastic estimator $\nabla F(x)$ as follows:

$$\hat{\nabla} f(x, y; S) = \nabla_x f(x, y; \xi) - \nabla^2_{xy} g(x, y; \zeta) \vartheta \sum_{q=-1}^{Q-1} \prod_{i=Q-q}^{Q} \left( I_p - \vartheta \nabla^2_{yy} g(x, y; \zeta^i) \right) \nabla_y f(x, y; \xi),$$

(3)

where $\vartheta > 0$ and $Q \geq 1$. Here $S = \big\{ \xi, \zeta, \zeta^1, \cdots \zeta^Q \big\}$, where $\xi$ is drawn from distribution $\mathcal{D}$, and $\{\zeta, \zeta^1, \cdots \zeta^Q\}$ are drawn from distribution $\mathcal{M}$.

## 3 ADAPTIVE BILEVEL OPTIMIZATION METHODS

In this section, we propose a class of fast single-loop adaptive bilevel optimization methods to solve the problem equation 1. Specifically, our methods adopt the universal adaptive learning rates as in Huang et al. (2021). Moreover, our methods can be flexibly incorporate the momentum and variance reduced techniques.

### 3.1 BIADAM ALGORITHM

In this subsection, we propose a fast single-loop adaptive bilevel optimization method (BiAdam) based on the basic momentum technique. Algorithm 1 shows the algorithmic framework of our BiAdam algorithm.

At the line 4 of Algorithm 1, we generate the adaptive matrices $A_t$ and $B_t$ for updating variables $x$ and $y$, respectively. Specifically, we use the general adaptive matrix $A_t \succeq \rho I_d$ ($\rho > 0$) for variable $x$, and the global adaptive matrix $B_t = b_t I_p$ ($b_t > 0$). For example, we can generate the matrix $A_t$ as the Adam Kingma & Ba (2014), and generate the matrix $B_t$ as a novel version of AdaGrad-Norm Ward et al. (2019), defined as

$$\tilde{w}_t = \alpha \tilde{w}_{t-1} + (1 - \alpha) \nabla_x f(x_t, y_t; \xi_t)^2, \ \tilde{w}_0 = 0, \ A_t = \text{diag}\big(\sqrt{\tilde{w}_t} + \rho\big), \ t \geq 1 \quad (4)$$
$$b_t = \beta b_{t-1} + (1 - \beta) \|\nabla_y g(x_t, y_t; \zeta_t)\|, \ b_0 > 0, \ B_t = (b_t + \varepsilon) I_p, \ t \geq 1, \quad (5)$$

where $\alpha, \ \beta \in (0, 1)$ and $\rho > 0, \ \varepsilon > 0$.

At the lines 5-6 of Algorithm 1, we use the generalized projection gradient iteration with Bregman distance Censor & Zenios (1992); Huang et al. (2021) to update the variables $x$ and $y$, respectively.

---

**Algorithm 1** BiAdam Algorithm

1: **Input:** $T, K \in \mathbb{N}$, parameters $\{\gamma, \lambda, \eta_t, \alpha_t, \beta_t\}$ and initial input $x_1 \in \mathcal{X}$ and $y_1 \in \mathcal{Y}$;
2: **initialize:** Draw $K + 2$ independent samples $\bar{\xi}_1 = \{\xi_1, \zeta_1^0, \zeta_1^1, \cdots, \zeta_1^{K-1}\}$ and $\zeta_1$, and then
   compute $v_1 = \nabla_y g(x_1, y_1; \zeta_1)$, and $w_1 = \bar{\nabla} f(x_1, y_1; \bar{\xi}_1)$ generated from (6);
3: **for** $t = 1, 2, \ldots, T$ **do**
4:    Generate adaptive matrices $A_t \in \mathbb{R}^{d \times d}$, $B_t \in \mathbb{R}^{p \times p}$;
5:    $\tilde{x}_{t+1} = \arg\min_{x \in \mathcal{X}} \{\langle w_t, x \rangle + \frac{1}{2\gamma}(x - x_t)^T A_t(x - x_t)\}$, and $x_{t+1} = x_t + \eta_t(\tilde{x}_{t+1} - x_t)$;
6:    $\tilde{y}_{t+1} = \arg\min_{y \in \mathcal{Y}} \{\langle v_t, y \rangle + \frac{1}{2\lambda}(y - y_t)^T B_t(y - y_t)\}$, and $y_{t+1} = y_t + \eta_t(\tilde{y}_{t+1} - y_t)$;
7:    Draw $K + 2$ independent samples $\bar{\xi}_{t+1} = \{\xi_{t+1}, \zeta_{t+1}^0, \cdots, \zeta_{t+1}^{K-1}\}$ and $\zeta_{t+1}$:
8:    $v_{t+1} = \alpha_{t+1} \nabla_y g(x_{t+1}, y_{t+1}; \zeta_{t+1}) + (1 - \alpha_{t+1})v_t$;
9:    $w_{t+1} = \beta_{t+1} \bar{\nabla} f(x_{t+1}, y_{t+1}; \bar{\xi}_{t+1}) + (1 - \beta_{t+1})w_t$;
10: **end for**
11: **Output:** Chosen uniformly random from $\{x_t, y_t\}_{t=1}^T$.

---

When $\mathcal{X} = \mathbb{R}^d$ and $\mathcal{Y} = \mathbb{R}^p$, i.e., unconstrained optimization problem (1), we have $x_{t+1} = x_t - \gamma\eta_t A_t^{-1} w_t$ and $y_{t+1} = y_t - \lambda\eta_t B_t^{-1} v_t$.

At the line 7 of Algorithm 1, we draw $K + 1$ independent samples $\bar{\xi} = \{\xi, \zeta^0, \zeta^1, \cdots, \zeta^{K-1}\}$ from distributions $\mathcal{D}$ and $\mathcal{M}$, then we define a stochastic gradient estimator as in Khanduri et al. (2021):

$$\bar{\nabla} f(x, y, \bar{\xi}) = \nabla_x f(x, y; \xi) - \nabla_{xy}^2 g(x, y; \zeta^0) \left[ \frac{K}{L_g} \prod_{i=1}^k \left( I_p - \frac{1}{L_g} \nabla_{yy}^2 g(x, y; \zeta^i) \right) \right] \nabla_y f(x, y; \xi),$$
(6)

where $K \geq 1$ and $k \sim \mathcal{U}\{0, 1, \cdots, K-1\}$ is a uniform random variable independent on $\bar{\xi}$. In fact, the estimator (6) is a specific case of the above estimator (3). In practice, thus, we can use a tuning parameter $\vartheta \in (0, \frac{1}{L_g}]$ instead of $\frac{1}{L_g}$ in the estimator (6) as in Yang et al. (2021). Here we use the term $\frac{K}{L_g} \prod_{i=1}^k \left( I_p - \frac{1}{L_g} \nabla_{yy}^2 g(x, y; \zeta^i) \right)$ to approximate the Hessian inverse, i.e., $\left( \nabla_{yy}^2 g(x, y; \zeta) \right)^{-1}$. Clearly, the above $\bar{\nabla} f(x, y, \bar{\xi})$ is a biased estimator in estimating $\bar{\nabla} f(x, y)$, i.e. $\mathbb{E}_{\bar{\xi}}[\bar{\nabla} f(x, y; \bar{\xi})] \neq \bar{\nabla} f(x, y)$. In the following, we give Lemma 3, which shows that the bias $R(x, y) = \bar{\nabla} f(x, y) - \mathbb{E}_{\bar{\xi}}[\bar{\nabla} f(x, y; \bar{\xi})]$ in the gradient estimator (6) decays exponentially fast with number $K$.

**Lemma 3.** *(Lemma 2.1 in Khanduri et al. (2021) and Lemma 11 in Hong et al. (2020)) Under the about Assumptions (1, 4), for any $K \geq 1$, the gradient estimator in equation 6 satisfies*

$$\|R(x, y)\| \leq \frac{C_{gxy} C_{fy}}{\mu} \left( 1 - \frac{\mu}{L_g} \right)^K,$$
(7)

*where $R(x, y) = \bar{\nabla} f(x, y) - \mathbb{E}_{\bar{\xi}}[\bar{\nabla} f(x, y; \bar{\xi})]$.*

From Lemma 3, choose $K = \frac{L_g}{\mu} \log(C_{gxy} C_{fy} T / \mu)$ in Algorithm 1, we have $\|R(x, y)\| \leq \frac{1}{T}$ for all $t \geq 1$. Thus, this result guarantees convergence of our algorithms only requiring a small mini-batch samples. For notational simplicity, let $R_t = R(x_t, y_t)$ for all $t \geq 1$.

**Lemma 4.** *(Lemma 3.1 in Khanduri et al. (2021)) Under the above Assumptions (1, 3, 4), stochastic gradient estimate $\bar{\nabla} f(x, y; \bar{\xi})$ is Lipschitz continuous, such that for $x, x_1, x_2 \in \mathcal{X}$ and $y, y_1, y_2 \in \mathcal{Y}$,*

$$\mathbb{E}_{\bar{\xi}} \|\bar{\nabla} f(x_1, y; \bar{\xi}) - \bar{\nabla} f(x_2, y; \bar{\xi})\|^2 \leq L_K^2 \|x_1 - x_2\|^2,$$

$$\mathbb{E}_{\bar{\xi}} \|\bar{\nabla} f(x, y_1; \bar{\xi}) - \bar{\nabla} f(x, y_2; \bar{\xi})\|^2 \leq L_K^2 \|y_1 - y_2\|^2,$$

*where $L_K^2 = 2L_f^2 + 6C_{gxy}^2 L_f^2 \frac{K}{2\mu L_g - \mu^2} + 6C_{fy}^2 L_{gxy}^2 \frac{K}{2\mu L_g - \mu^2} + 6C_{gxy}^2 L_f^2 \frac{K^3 L_{gyy}^2}{(L_g - \mu)^2 (2\mu L_g - \mu^2)}$.*

### 3.2 VR-BiAdam Algorithm

In this subsection, we propose an accelerated version of BiAdam method (VR-BiAdam) by using the momentum-based variance reduced technique. Algorithm 2 demonstrates the algorithmic framework of our VR-BiAdam algorithm.

---

**Algorithm 2** VR-BiAdam Algorithm

1: **Input:** $T, K \in \mathbb{N}$, parameters $\{\gamma, \lambda, \eta_t, \alpha_t, \beta_t\}$ and initial input $x_1 \in \mathcal{X}$ and $y_1 \in \mathcal{Y}$;
2: **initialize:** Draw $K + 2$ independent samples $\bar{\xi}_1 = \{\xi_1, \zeta_1^0, \zeta_1^1, \cdots, \zeta_1^{K-1}\}$ and $\zeta_1$, and then compute $v_1 = \nabla_y g(x_1, y_1; \zeta_1)$, and $w_1 = \bar{\nabla} f(x_1, y_1; \bar{\xi}_1)$ generated from (6);
3: **for** $t = 1, 2, \ldots, T$ **do**
4: $\quad$ Generate adaptive matrices $A_t \in \mathbb{R}^{d \times d}$, $B_t \in \mathbb{R}^{p \times p}$;
5: $\quad \tilde{x}_{t+1} = \arg\min_{x \in \mathcal{X}} \left\{ \langle w_t, x \rangle + \frac{1}{2\gamma}(x - x_t)^T A_t (x - x_t) \right\}$, and $x_{t+1} = x_t + \eta_t(\tilde{x}_{t+1} - x_t)$;
6: $\quad \tilde{y}_{t+1} = \arg\min_{y \in \mathcal{Y}} \left\{ \langle v_t, y \rangle + \frac{1}{2\lambda}(y - y_t)^T B_t (y - y_t) \right\}$, and $y_{t+1} = y_t + \eta_t(\tilde{y}_{t+1} - y_t)$;
7: $\quad$ Draw $K + 2$ independent samples $\bar{\xi}_{t+1} = \{\xi_{t+1}, \zeta_{t+1}^0, \cdots, \zeta_{t+1}^{K-1}\}$ and $\zeta_{t+1}$;
8: $\quad v_{t+1} = \nabla_y g(x_{t+1}, y_{t+1}; \zeta_{t+1}) + (1 - \alpha_{t+1})[v_t - \nabla_y g(x_t, y_t; \zeta_{t+1})]$;
9: $\quad w_{t+1} = \bar{\nabla} f(x_{t+1}, y_{t+1}; \bar{\xi}_{t+1}) + (1 - \beta_{t+1})[w_t - \bar{\nabla} f(x_t, y_t; \bar{\xi}_{t+1})]$;
10: **end for**
11: **Output:** Chosen uniformly random from $\{x_t, y_t\}_{t=1}^T$.

---

At the lines 8-9 of Algorithm 2, we use the momentum-based variance reduced technique to estimate the stochastic partial derivatives $v_t$ and $w_t$. For example, the estimator of partial derivative $\bar{\nabla} f(x_{t+1}, y_{t+1})$ is defined as

$$w_{t+1} = \bar{\nabla} f(x_{t+1}, y_{t+1}; \bar{\xi}_{t+1}) + (1 - \beta_{t+1})[w_t - \bar{\nabla} f(x_t, y_t; \bar{\xi}_{t+1})],$$
$$= \beta_{t+1} \bar{\nabla} f(x_{t+1}, y_{t+1}; \bar{\xi}_{t+1}) + (1 - \beta_{t+1})[w_t + \bar{\nabla} f(x_{t+1}, y_{t+1}; \bar{\xi}_{t+1}) - \bar{\nabla} f(x_t, y_t; \bar{\xi}_{t+1})].$$

Compared with the estimator $w_{t+1}$ in Algorithm 1, $w_{t+1}$ in Algorithm 2 adds the term $\bar{\nabla} f(x_{t+1}, y_{t+1}; \bar{\xi}_{t+1}) - \bar{\nabla} f(x_t, y_t; \bar{\xi}_{t+1})$ to control the variances of estimator.

## 4 THEORETICAL ANALYSIS

In this section, we study the convergence properties of our algorithms (BiAdam and VR-BiAdam) under some mild conditions. All proofs are provided in the Appendix A.

### 4.1 ADDITIONAL MILD ASSUMPTIONS

**Assumption 6.** *The estimated stochastic partial derivative $\bar{\nabla} f(x, y; \bar{\xi})$ satisfies*

$$\mathbb{E}_{\bar{\xi}}[\bar{\nabla} f(x, y; \bar{\xi})] = \bar{\nabla} f(x, y) + R(x, y), \quad \mathbb{E}_{\bar{\xi}} \|\bar{\nabla} f(x, y; \bar{\xi}) - \bar{\nabla} f(x, y) - R(x, y)\|^2 \leq \sigma^2.$$

*The stochastic partial derivative $\nabla_y g(x, y; \zeta)$ satisfies*

$$\mathbb{E}[\nabla_y g(x, y; \zeta)] = \nabla_y g(x, y), \quad \mathbb{E} \|\nabla_y g(x, y; \zeta) - \nabla_y g(x, y)\|^2 \leq \sigma^2.$$

**Assumption 7.** *In our algorithms, the adaptive matrices $A_t$ and $B_t$ for all $t \geq 1$ satisfy $A_t \succeq \rho I_d$ $(\rho > 0)$ and $B_t = b I_p$ $(b_u \geq b \geq b_l > 0)$, respectively, where $\rho$, $b_u$ and $b_l$ are appropriate positive numbers.*

Assumption 6 is commonly used in the stochastic bilevel optimization methods Ji et al. (2021); Yang et al. (2021); Khanduri et al. (2021). In the paper, we consider the general adaptive learning rates (including the coordinate-wise and global learning rates) for variable $x$ and the global learning rate for variable $y$. Assumption 7 ensures that the adaptive matrices $A_t$ for all $t \geq 1$ are positive definite as in Huang et al. (2021). Assumption 7 also guarantees the global adaptive matrices $B_t$ for all $t \geq 1$ are positive definite and bounded. In fact, Assumption 7 is mild. For example, in the problem $\min_{x \in \mathbb{R}^p} \mathbb{E}[f(x; \xi)]$, Ward et al. (2019) apply a global adaptive learning rate to the update form $x_t = x_{t-1} - \eta \frac{\nabla f(x_{t-1}; \xi_{t-1})}{b_t}$, $b_t^2 = b_{t-1}^2 + \|\nabla f(x_{t-1}; \xi_{t-1})\|^2$, $b_0 > 0, \eta > 0$ for all $t \geq 1$, which is equivalent to the form $x_t = x_{t-1} - \eta B_t^{-1} \nabla f(x_{t-1}; \xi_{t-1})$ with $B_t = b_t I_p$ and $b_t \geq \cdots \geq b_0 > 0$. Li & Orabona (2019); Cutkosky & Orabona (2019) use a global adaptive learning rate to the update form $x_{t+1} = x_t - \eta g_t / b_t$, where $g_t$ is stochastic gradient and $b_t = (\omega + \sum_{i=1}^t \|\nabla f(x_i; \xi_i)\|^2)^\alpha / k$, $k > 0$, $\omega > 0$ and $\alpha \in (0, 1)$, which is equivalent to $x_{t+1} = x_t - \eta B_t^{-1} g_t$ with $B_t = b_t I_p$ and $b_t \geq \cdots \geq b_0 = \frac{\omega^\alpha}{k} > 0$. At the same time, the problem $\min_{x \in \mathbb{R}^p} f(x) = \mathbb{E}[f(x; \xi)]$ approaches the stationary points, i.e., $\nabla f(x) = 0$ or even $\nabla f(x; \xi) = 0$ for all $\xi$. Thus, these global adaptive learning rates are generally bounded, i.e., $b_u \geq b_t \geq b_l > 0$ for all $t \geq 1$.

### 4.2 Useful Convergence Metric

In the subsection, we define a useful convergence metric for our algorithms and some useful lemmas.

**Lemma 5.** *Given gradient estimator $w_t$ is generated from Algorithms 1 or 2, for all $t \geq 1$, we have*

$$\|w_t - \nabla F(x_t)\|^2 \leq L_0^2 \|y^*(x_t) - y_t\|^2 + 2\|w_t - \bar{\nabla} f(x_t, y_t)\|^2,$$

*where $L_0^2 = 8\big(L_f^2 + \frac{L_{gxy}^2 C_{fy}^2}{\mu^2} + \frac{L_{gyy}^2 C_{gxy}^2 C_{fy}^2}{\mu^4} + \frac{L_f^2 C_{gxy}^2}{\mu^2}\big)$.*

For our Algorithms 1 and 2, based on Lemma 5, we provide a convergence metric $\mathbb{E}[\mathcal{G}_t]$, defined as

$$\mathcal{G}_t = \frac{1}{\gamma}\|\tilde{x}_{t+1} - x_t\| + \frac{1}{\rho}\Big(\sqrt{2}\|w_t - \bar{\nabla} f(x_t, y_t)\| + L_0\|y^*(x_t) - y_t\|\Big),$$

where the first two terms of $\mathcal{G}_t$ measure the convergence of the iteration solutions $\{x_t\}_{t=1}^T$, and the last term measures the convergence of the iteration solutions $\{y_t\}_{t=1}^T$.

Let $\phi_t(x) = \frac{1}{2}x^T A_t x$, we define a prox-function (a.k.a., Bregman distance) Censor & Lent (1981); Censor & Zenios (1992); Ghadimi et al. (2016) associated with $\phi_t(x)$ as follows:

$$V_t(x, x_t) = \phi_t(x) - \big[\phi_t(x_t) + \langle \nabla \phi_t(x_t), x - x_t \rangle\big] = \frac{1}{2}(x - x_t)^T A_t (x - x_t). \tag{8}$$

The line 5 of Algorithm 1 or 2 is equivalent to the following generalized projection problem:

$$\tilde{x}_{t+1} = \min_{x \in \mathcal{X}}\Big\{\langle w_t, x \rangle + \frac{1}{\gamma}V_t(x, x_t)\Big\}. \tag{9}$$

As in Ghadimi et al. (2016), we define a generalized projected gradient $\mathcal{G}_{\mathcal{X}}(x_t, w_t, \gamma) = \frac{1}{\gamma}(x_t - \tilde{x}_{t+1})$. At the same time, we define a gradient mapping $\mathcal{G}_{\mathcal{X}}(x_t, \nabla F(x_t), \gamma) = \frac{1}{\gamma}(x_t - x_{t+1}^+)$ with

$$x_{t+1}^+ = \arg\min_{x \in \mathcal{X}}\Big\{\langle \nabla F(x_t), x \rangle + \frac{1}{\gamma}V_t(x, x_t)\Big\}. \tag{10}$$

According to the Proposition 1 of Ghadimi et al. (2016), we have $\|\mathcal{G}_{\mathcal{X}}(x_t, w_t, \gamma) - \mathcal{G}_{\mathcal{X}}(x_t, \nabla F(x_t), \gamma)\| \leq \frac{1}{\rho}\|\nabla F(x_t) - w_t\|$. Since $\|\mathcal{G}_{\mathcal{X}}(x_t, \nabla F(x_t), \gamma)\| \leq \|\mathcal{G}_{\mathcal{X}}(x_t, w_t, \gamma)\| + \|\mathcal{G}_{\mathcal{X}}(x_t, w_t, \gamma) - \mathcal{G}_{\mathcal{X}}(x_t, \nabla F(x_t), \gamma)\|$, we have

$$\|\mathcal{G}_{\mathcal{X}}(x_t, \nabla F(x_t), \gamma)\| \leq \|\mathcal{G}_{\mathcal{X}}(x_t, w_t, \gamma)\| + \frac{1}{\rho}\|\nabla F(x_t) - w_t\| \tag{11}$$

$$\leq \frac{1}{\gamma}\|\tilde{x}_{t+1} - x_t\| + \frac{1}{\rho}\Big(\sqrt{2}\|w_t - \bar{\nabla} f(x_t, y_t)\| + L_0\|y^*(x_t) - y_t\|\Big) = \mathcal{G}_t,$$

where the last inequality holds by the above Lemma 5. Thus, our new convergence measure $\mathbb{E}[\mathcal{G}_t]$ is tighter than the standard gradient mapping $\mathbb{E}\|\mathcal{G}_{\mathcal{X}}(x_t, \nabla F(x_t), \gamma)\|$ used in Hong et al. (2020). When $\mathcal{G}_t \to 0$, we have $\|\mathcal{G}_{\mathcal{X}}(x_t, \nabla F(x_t), \gamma)\| \to 0$, where $x_t$ is a stationary point or local minimum of the bilevel problem equation 1 Ghadimi et al. (2016); Hong et al. (2020).

### 4.3 Convergence Analysis of BiAdam Algorithm

In this subsection, we study the convergence properties of our BiAdam algorithm. The detailed proofs are provided in the Appendix A.5.

**Theorem 1.** *Under the above Assumptions (1, 2, 4, 6, 7), in the Algorithm 1, given $\mathcal{X} \subset \mathbb{R}^d$, $\eta_t = \frac{k}{(m+t)^{1/2}}$ for all $t \geq 0$, $\alpha_{t+1} = c_1\eta_t$, $\beta_{t+1} = c_2\eta_t$, $m \geq \max\big(k^2, (c_1k)^2, (c_2k)^2\big)$, $k > 0$, $\frac{125L_0^2}{6\mu^2} \leq c_1 \leq \frac{m^{1/2}}{k}$, $\frac{9}{2} \leq c_2 \leq \frac{m^{1/2}}{k}$, $0 < \lambda \leq \min\big(\frac{15b_lL_0^2}{4L_1^2\mu}, \frac{b_l}{6L_g}\big)$, $0 < \gamma \leq \min\big(\frac{\sqrt{6}\lambda\mu\rho}{\sqrt{6L_1^2\lambda^2\mu^2 + 125b_u^2L_0^2\kappa^2}}, \frac{m^{1/2}\rho}{4Lk}\big)$ and $K = \frac{L_g}{\mu}\log(C_{gxy}C_{fy}T/\mu)$, we have*

$$\frac{1}{T}\sum_{t=1}^T \mathbb{E}\|\mathcal{G}_{\mathcal{X}}(x_t, \nabla F(x_t), \gamma)\| \leq \frac{1}{T}\sum_{t=1}^T \mathbb{E}[\mathcal{G}_t] \leq \frac{2\sqrt{3G}m^{1/4}}{T^{1/2}} + \frac{2\sqrt{3G}}{T^{1/4}} + \frac{\sqrt{2}}{T} = \tilde{O}\big(\frac{1}{T^{1/4}}\big), \tag{12}$$

*where $G = \frac{F(x_1) - F^*}{\rho k\gamma} + \frac{5b_1L_0^2\Delta_0}{\rho^2 k\lambda\mu} + \frac{2\sigma^2}{\rho^2 k} + \frac{2m\sigma^2}{\rho^2 k}\ln(m+T) + \frac{4(m+T)}{9\rho^2 kT^2} + \frac{8k}{\rho^2 T}$ and $\Delta_0 = \|y_1 - y^*(x_1)\|^2$, $L_1^2 = \frac{12L_g^2\mu^2}{125L_0^2} + \frac{2L_0^2}{3}$.*

**Remark 1.** *Without loss of generality, let $k = O(1)$ and $m = O(1)$, we have $G = O(\ln(m + T)) = \tilde{O}(1)$. Thus our BiAdam algorithm has a convergence rate of $\tilde{O}(\frac{1}{T^{1/4}})$. Let $\mathbb{E}[\mathcal{G}_\zeta] = \frac{1}{T}\sum_{t=1}^{T}\mathbb{E}[\mathcal{G}_t] = \tilde{O}(\frac{1}{T^{1/4}}) \leq \epsilon$, we have $T = \tilde{O}(\epsilon^{-4})$. Since our BiAdam algorithm only requires $K + 2 = \frac{L_g}{\mu}\log(C_{gxy}C_{fy}T/\mu) + 2 = \tilde{O}(1)$ samples to estimate stochastic partial derivatives in each iteration, and needs $T$ iterations. Thus our BiAdam algorithm requires sample complexity of $(K + 2)T = \tilde{O}(\epsilon^{-4})$ for finding an $\epsilon$-stationary point of the problem (1). Note that the convergence properties of our BiAdam algorithm for* unconstrained *bilevel optimization are provided in the Appendix A.3.*

## 4.4 CONVERGENCE ANALYSIS OF VR-BiAdam ALGORITHM

In this subsection, we study convergence properties of our VR-BiAdam algorithm. The detailed proofs are provided in the Appendix A.6.

**Theorem 2.** *Under the above Assumptions (1, 3, 4, 6, 7), in the Algorithm 2, given $\mathcal{X} \subset \mathbb{R}^d$, $\eta_t = \frac{k}{(m+t)^{1/3}}$ for all $t \geq 0$, $\alpha_{t+1} = c_1\eta_t^2$, $\beta_{t+1} = c_2\eta_t^2$, $m \geq \max\left(2, k^3, (c_1 k)^3, (c_2 k)^3\right)$, $k > 0$, $c_1 \geq \frac{2}{3k^3} + \frac{125L_0^2}{6\mu^2}$, $c_2 \geq \frac{2}{3k^3} + \frac{9}{2}$, $0 < \lambda \leq \min\left(\frac{15b_l L_0^2}{16L_2^2\mu}, \frac{b_l}{6L_g}\right)$, $0 < \gamma \leq \min\left(\frac{\sqrt{6}\lambda\mu\rho}{2\sqrt{24L_2^2\lambda^2\mu^2 + 125b_u^2L_0^2\kappa^2}}, \frac{m^{1/3}\rho}{4Lk}\right)$ and $K = \frac{L_g}{\mu}\log(C_{gxy}C_{fy}T/\mu)$, we have*

$$\frac{1}{T}\sum_{t=1}^{T}\mathbb{E}\|\mathcal{G}_{\mathcal{X}}(x_t, \nabla F(x_t), \gamma)\| \leq \frac{1}{T}\sum_{t=1}^{T}\mathbb{E}[\mathcal{G}_t] \leq \frac{2\sqrt{3M}m^{1/6}}{T^{1/2}} + \frac{2\sqrt{3M}}{T^{1/3}} + \frac{\sqrt{2}}{T} = \tilde{O}(\frac{1}{T^{1/3}}),$$
(13)

*where $M = \frac{F(x_1)-F^*}{\rho k\gamma} + \frac{5b_1 L_0^2 \Delta_0}{\rho^2 k\lambda\mu} + \frac{2m^{1/3}\sigma^2}{\rho^2 k^2} + \frac{2k^2(c_1^2+c_2^2)\sigma^2\ln(m+T)}{\rho^2} + \frac{6k(m+T)^{1/3}}{\rho^2 T}$, $\Delta_0 = \|y_1 - y^*(x_1)\|^2$ and $L_2^2 = L_g^2 + L_K^2$.*

**Remark 2.** *Without loss of generality, let $k = O(1)$ and $m = O(1)$, we have $M = O(\ln(m + T)) = \tilde{O}(1)$. Thus our VR-BiAdam algorithm has a convergence rate of $\tilde{O}(\frac{1}{T^{1/3}})$. Let $\mathbb{E}[\mathcal{G}_\zeta] = \frac{1}{T}\sum_{t=1}^{T}\mathbb{E}[\mathcal{G}_t] = \tilde{O}(\frac{1}{T^{1/3}}) \leq \epsilon$, we have $T = \tilde{O}(\epsilon^{-3})$. Since our VR-BiAdam algorithm requires $K + 2 = \frac{L_g}{\mu}\log(C_{gxy}C_{fy}T/\mu) + 2 = \tilde{O}(1)$ samples to estimate stochastic partial derivatives in each iteration, and needs $T$ iterations. Thus our VR-BiAdam algorithm requires sample complexity of $(K + 2)T = \tilde{O}(\epsilon^{-3})$ for finding an $\epsilon$-stationary point of the problem (1). Note that the convergence properties of our VR-BiAdam algorithm for* unconstrained *bilevel optimization are provided in the Appendix A.3.*

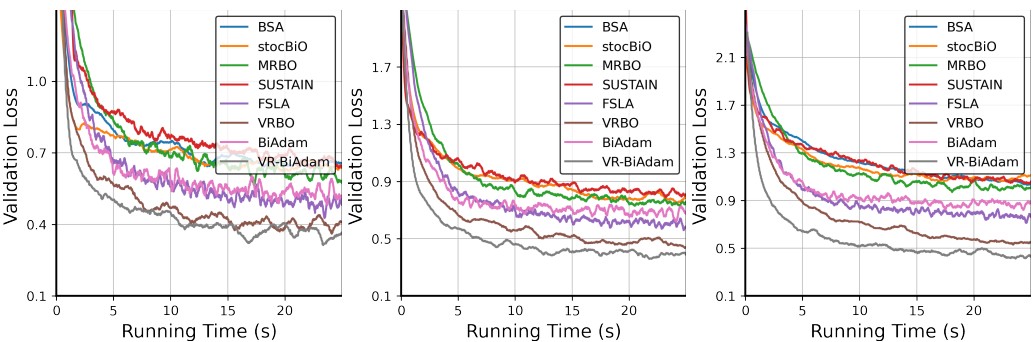

Figure 1: Validation Loss $vs.$ Running Time. We test three values of $\rho$: 0.8, 0.6, 0.2 from left to right. Larger value of $\rho$ represents a more noisy setting.

## 5 NUMERICAL EXPERIMENTS

In this section, we perform two hyper-parameter optimization tasks to demonstrate the efficiency of our algorithms: 1) data hyper-cleaning task over the MNIST dataset; 2) hyper-representation learning task over the Omniglot dataset. In all experiments, we use a server with AMD EPYC 7763 64-Core CPU and 1 NVIDIA RTX A5000.

## 5.1 DATA HYPER-CLEANING

In the hyper-cleaning task, we clean a noisy dataset through a bilevel formulation. The precise formulation of the problem is included in the Appendix A.1.1. In particular, we use a training set and a validation set, where each contains 5000 images in our experiments. A portion of the training data are corrupted by randomly changing their labels, and we denote the portion of corrupted images as $\rho$. We compare our algorithms (*i.e.*, BiAdam and VR-BiAdam) with various baselines. See the Appendix A.1 for a brief introduction of the baselines. For all methods, we perform grid search over hyper-parameters and choose the best setting. The detailed experimental setup is described in the Appendix A.1.1. The experimental results are summarized in Figure 1. As shown by the figure, our BiAdam algorithm outperforms its non-adaptive counterparts such as stocBiO, MRBO and SUSTAIN, furthermore, our VR-BiAdam gets the best performance, where it outperforms VRBO, which requires using large batch-sizes every a few iterations.

## 5.2 HYPER-REPRESENTATION LEARNING

In the hyper-representation learning task, we learn a hyper-representation of the data such that a linear classifier can be learned quickly with a small number of data samples. The precise formulation of the problem is included in Appendix A.1.2. We compare our algorithms (*i.e.*, BiAdam and VR-BiAdam) with various baselines. See Appendix A.1 for a brief introduction of the baselines. For all methods, we perform grid search over hyper-parameters and choose the best setting. The detailed experimental setup is described in the Appendix A.1.2. The experimental results are summarized in Figure 2. As shown by the figure, both our BiAdam and VR-BiAdam algorithms outperform other baselines.

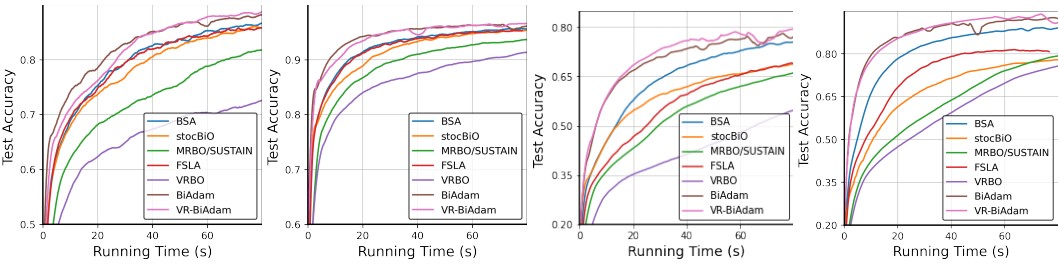

Figure 2: Test Accuracy *vs.* Running Time. The plots corresponds to 5-way-1-shot, 5-way-5-shot, 20-way-1-shot and 20-way-5-shots from left to right.

## 6 CONCLUSIONS

In this paper, we proposed a class of novel adaptive bilevel optimization methods for nonconvex-strongly-convex bilevel optimization problems. Our methods use unified adaptive matrices including many types of adaptive learning rates, and can flexibly use the momentum and variance reduced techniques. Moreover, we provided a useful convergence analysis framework for both the constrained and unconstrained bilevel optimization. Our VR-BiAdam algorithm reaches the best known sample complexity of $\tilde{O}(\epsilon^{-3})$ for finding an $\epsilon$-stationary point.

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

# A APPENDIX

In this section, we provide the additional experiment results, related works and additional theoretical results. We also provide the detailed convergence analysis.

## A.1 ADDITIONAL EXPERIMENTAL DETAILS AND RESULTS

In this subsection, we introduce more details of our experiments. We compare our BiAdam and VR-BiAdam algorithms with the following bilevel optimization algorithms: reverse Franceschi et al. (2018), AID-CG Grazzi et al. (2020), AID-FP Grazzi et al. (2020), stocBio Ji et al. (2021)), MRBO Ji & Liang (2021), VRBO Ji & Liang (2021), FSLA Li et al. (2021), MSTSA/SUSTAIN Khanduri et al. (2021), SMB Guo et al. (2021b), SVRB Guo & Yang (2021).

### A.1.1 DATA HYPER-CLEANING

In this task, we perform data hyper-cleaning over the MNIST dataset LeCun et al. (1998). The formulation of this problem is as follows:

$$\min_\tau f_{val}\big(\tau, w^*(\tau)\big) := \frac{1}{|D_\mathcal{V}|} \sum_{(x_i,y_i)\in D_\mathcal{V}} f\big(x_i^T w^*(\tau), y_i\big)$$

$$\text{s.t. } w^*(\tau) = \arg\min_w f_{tr}(\tau, w) := \frac{1}{|D_\mathcal{T}|} \sum_{(x_i,y_i)\in D_\mathcal{T}} \sigma(\tau_i) f(x_i^T w, y_i) + C\|w\|^2,$$

where $f(\cdot)$ denotes the cross entropy loss, $D_\mathcal{T}$ and $D_\mathcal{V}$ are training and validation dataset, respectively. Here $\tau = \{\tau_i\}_{i\in\mathcal{D}_\mathcal{T}}$ are hyper-parameters and $C \geq 0$ is a tuning parameter, $\sigma(\cdot)$ denotes the sigmoid function. In the experiment, we set $C = 0.001$.

For training/validation batch-size, we use batch-size of 32, while for VRBO, we choose larger batch-size 5000 and sampling interval is set as 3. For AID-FP, AID-CG and reverse, we use the warm-start trick as our algorithms, *i.e.* the inner variable starts from the state of last iteration. We fine tune the number of inner-loop iterations and set it to be 50 for these algorithms. For MRBO, VRBO, SUSTAIN and our BiAdam/VR-BiAdam, we set $K = 3$ to evaluate the hyper-gradient. For FSLA, $K = 1$ as the hyper-gradient is evaluated recursively. As for learning rates, we set 1000 as the outer learning rate for all algorithms except our algorithms which use 0.5 as we change the learning rate adaptively. As for the inner learning rates, we set the stepsize as 0.05 for reverse, AID-CG, stocBiO/AID-FP, MRBO/SUSTAIN, FSLA; we set the stepsize as 0.2 for VRBO; we set the stepsize as 1 for SUSTAIN; we set the stepsize as 0.00025 for BiAdam and 0.0005 for VR-BiAdam.

### A.1.2 HYPER-REPRESENTATION LEARNING

In this task, we perform the hyper-representation learning task over the Omniglot dataset Lake et al. (2015). The formulation of this problem is as follows:

$$\min_\tau f_{val}\big(\tau, w^*(\tau)\big) := \mathbb{E}\big[f_{val}\big(\tau, w^*(\tau); \xi\big)\big] := \frac{1}{|D_{\mathcal{V},\xi}|} \sum_{(x_i,y_i)\in D_{\mathcal{V},\xi}} f\big((\omega_\tau^*)^T \phi(x_i; \tau), y_i\big)$$

$$\text{s.t. } w^*(\tau) = \arg\min_w f_{tr}(\tau, w) := \frac{1}{|D_{\mathcal{T},\xi}|} \sum_{(x_i,y_i)\in D_{\mathcal{T},\xi}} f\big(\omega^T \phi(x_i; \tau), y_i\big) + C\|w\|^2,$$

where $f(\cdot)$ denotes the cross entropy loss, $D_{\mathcal{T},\xi}$ and $D_{\mathcal{V},\xi}$ are training and validation dataset for a randomly sampled meta task. Here $\tau = \{\tau_i\}_{i\in\mathcal{D}_\mathcal{T}}$ are hyper-representations and $C \geq 0$ is a tuning parameter to gaurantee the inner problem to be strongly convex. In experiment, we set $C = 0.01$.

In every hyper-iteration, we choose 4 meta tasks, while for VRBO, we choose larger batch-size 16 and sampling interval is set as 3. For stocBiO/AID-FP, AID-CG and reverse, we use the warm-start trick as our algorithms, *i.e.* the inner variable starts from the state of last iteration. We fine tune the number of inner-loop iterations and set it to be 16 for these algorithms. For MRBO, VRBO, SUSTAIN and our algorithms, we set $K = 5$ to evaluate the hyper-gradient. For FSLA, $K = 1$ as the hyper-gradient is evaluated recursively. As for learning rates, we set 1000 as the outer learning rate for all algorithms except our algorithms which use 0.001 as we change the learning rate adaptively. As for the inner learning rates, we set the stepsize as 0.4 for all algorithms.

## A.2 RELATED WORKS

In this subsection, we overview the existing bilevel optimization methods and adaptive methods for single-level optimization, respectively.

### A.2.1 BILEVEL OPTIMIZATION METHODS

Bilevel optimization has shown successes in many machine learning problems with hierarchical structures such as policy optimization Hong et al. (2020), model-agnostic meta-learning Liu et al. (2021a); Ji et al. (2021); Lu et al. (2022) and adversarial training Zhang et al. (2021). Thus, its researches have become active in the machine learning community, and some bilevel optimization methods recently have been proposed. For example, one class of successful methods Colson et al. (2007); Kunapuli et al. (2008) are to reformulate the bilevel problem as a single-level problem by replacing the inner problem by its optimality conditions. Another class of successful methods Ghadimi & Wang (2018); Hong et al. (2020); Ji et al. (2021); Chen et al. (2022); Khanduri et al. (2021); Guo & Yang (2021); Liu et al. (2021b; 2022); Li et al. (2021) for bilevel optimization are to iteratively approximate the (stochastic) gradient of the outer problem either in forward or backward. Specifically, Liu et al. (2022) proposed a general gradient-based descent aggregation framework for bilevel optimization. Moreover, the non-asymptotic analysis of these gradient-based bilevel optimization methods has been recently studied. For example, Ghadimi & Wang (2018) first studied the sample complexity of $O(\epsilon^{-6})$ of the proposed double-loop algorithm for the bilevel problem (1) (Please see Table 1). Subsequently, Ji et al. (2021) proposed an accelerated double-loop algorithm that reaches the sample complexity of $O(\epsilon^{-4})$ relying on large batches. At the same time, Hong et al. (2020) studied a single-loop algorithm that reaches the sample complexity of $O(\epsilon^{-5})$ without relying on large batches. Moreover, Khanduri et al. (2021); Guo & Yang (2021); Yang et al. (2021) proposed a class of accelerated single-loop methods for the bilevel problem (1) by using momentum-based variance reduced technique, which achieve the best known sample complexity of $O(\epsilon^{-3})$. More recently, Dagréou et al. (2022) proposed a novel framework for bilevel optimization based on the linear system. Meanwhile, Lu et al. (2022); Li et al. (2022); Tarzanagh et al. (2022) studied the distributed bilevel optimization.

### A.2.2 ADAPTIVE GRADIENT METHODS

Adaptive gradient methods recently have been shown great successes in current machine learning problems such as training Deep Neural Networks (DNNs). Recently, thus many adaptive gradient methods Duchi et al. (2011); Kingma & Ba (2014); Loshchilov & Hutter (2018); Zhuang et al. (2020) have been developed and studied. For example, Adagrad Duchi et al. (2011) is the first adaptive gradient method that shows good performances under the sparse gradient setting. One variant of Adagrad, i.e., Adam Kingma & Ba (2014) is a very popular adaptive gradient method and basically is a default method of choice for training DNNs. Subsequently, some variants of Adam algorithm Reddi et al. (2019); Chen et al. (2019) have been developed and studied, and especially they have convergence guarantee under the nonconvex setting. At the same time, some adaptive gradient methods Loshchilov & Hutter (2018); Chen et al. (2018); Zhuang et al. (2020) have been presented to improve the generalization performance of Adam algorithm. The norm version of AdaGrad (i.e., AdaGrad-Norm) Ward et al. (2019) has been presented to accelerate AdaGrad without sacrificing generalization. Moreover, some accelerated adaptive gradient methods such as STORM Cutkosky & Orabona (2019) and SUPER-ADAM Huang et al. (2021) have been proposed by using variance-reduced technique. Meanwhile, Huang et al. (2021); Guo et al. (2021a) studied the convergence analysis framework for adaptive gradient methods.

## A.3 ADDITIONAL THEORETICAL RESULTS

In this subsection, we further give the convergence properties of our BiAdam algorithm for *unconstrained* bilevel optimization.

**Theorem 3.** *Under the above Assumptions (1, 2, 4, 6, 7), in the Algorithm 1, given $\mathcal{X} = \mathbb{R}^d$, $\eta_t = \frac{k}{(m+t)^{1/2}}$ for all $t \geq 0$, $\alpha_{t+1} = c_1 \eta_t$, $\beta_{t+1} = c_2 \eta_t$, $m \geq \max\left(k^2, (c_1 k)^2, (c_2 k)^2\right)$, $k > 0$, $\frac{125 L_0^2}{6\mu^2} \leq c_1 \leq \frac{m^{1/2}}{k}$, $\frac{9}{2} \leq c_2 \leq \frac{m^{1/2}}{k}$, $0 < \lambda \leq \min\left(\frac{15 b_l L_0^2}{4 L_1^2 \mu}, \frac{b_l}{6 L_g}\right)$, $0 < \gamma \leq$*

$\min\big(\frac{\sqrt{6}\lambda\mu\rho}{\sqrt{6L_1^2\lambda^2\mu^2+125b_u^2L_0^2\kappa^2}}, \frac{m^{1/2}\rho}{4Lk}\big)$ *and* $K = \frac{L_g}{\mu}\log(C_{gxy}C_{fy}T/\mu)$, *we have*

$$\frac{1}{T}\sum_{t=1}^T \mathbb{E}\|\nabla F(x_t)\| \leq \frac{\sqrt{\frac{1}{T}\sum_{t=1}^T \mathbb{E}\|A_t\|^2}}{\rho}\Big(\frac{2\sqrt{6G'm}}{T^{1/2}} + \frac{2\sqrt{6G'}}{T^{1/4}} + \frac{2\sqrt{3}}{T}\Big) = \tilde{O}(\frac{1}{T^{1/4}}), \quad (14)$$

*where* $G' = \frac{\rho(F(x_1)-F^*)}{k\gamma} + \frac{5b_1L_0^2\Delta_0}{k\lambda\mu} + \frac{2\sigma^2}{k} + \frac{2m\sigma^2}{k}\ln(m+T) + \frac{4(m+T)}{9kT^2} + \frac{8k}{T}$.

**Remark 3.** *Under the same conditions in Theorem 1, based on the metric* $\frac{1}{T}\sum_{t=1}^T \mathbb{E}\|\nabla F(x_t)\|$, *our BiAdam algorithm still has a gradient complexity of* $\tilde{O}(\epsilon^{-4})$ *without relying on the large mini-batches. Interestingly, the right hand side of the above inequality (14) includes a term* $\frac{\sqrt{\frac{1}{T}\sum_{t=1}^T \mathbb{E}\|A_t\|^2}}{\rho}$ *that can be seen as an upper bound of the expected condition number of adaptive matrices* $\{A_t\}_{t=1}^T$. *When* $A_t$ *given in the above (4), we have* $\frac{\sqrt{\frac{1}{T}\sum_{t=1}^T \mathbb{E}\|A_t\|^2}}{\rho} \leq \frac{G_1+\lambda}{\lambda}$ *as in the existing adaptive gradient methods assuming the standard bounded stochastic gradient* $\|\nabla f(x;\xi)\| \leq G_1$.

Next, we further give the convergence properties of our VR-BiAdam algorithm for *unconstrained bilevel optimization.*

**Theorem 4.** *Under the above Assumptions (1, 3, 4, 6, 7), in the Algorithm 2, given* $\mathcal{X} = \mathbb{R}^d$, $\eta_t = \frac{k}{(m+t)^{1/3}}$ *for all* $t \geq 0$, $\alpha_{t+1} = c_1\eta_t^2$, $\beta_{t+1} = c_2\eta_t^2$, $m \geq \max\big(2, k^3, (c_1k)^3, (c_2k)^3\big)$, $k > 0$, $c_1 \geq \frac{2}{3k^3} + \frac{125L_0^2}{6\mu^2}$, $c_2 \geq \frac{2}{3k^3} + \frac{9}{2}$, $0 < \lambda \leq \min\big(\frac{15b_1L_0^2}{16L_2^2\mu}, \frac{b_l}{6L_g}\big)$, $0 < \gamma \leq \min\big(\frac{\sqrt{6}\lambda\mu\rho}{2\sqrt{24L_2^2\lambda^2\mu^2+125b_u^2L_0^2\kappa^2}}, \frac{m^{1/3}\rho}{4Lk}\big)$ *and* $K = \frac{L_g}{\mu}\log(C_{gxy}C_{fy}T/\mu)$, *we have*

$$\frac{1}{T}\sum_{t=1}^T \mathbb{E}\|\nabla F(x_t)\| \leq \frac{\sqrt{\frac{1}{T}\sum_{t=1}^T \mathbb{E}\|A_t\|^2}}{\rho}\Big(\frac{2\sqrt{6M'm}}{T^{1/2}} + \frac{2\sqrt{6M'}}{T^{1/3}} + \frac{2\sqrt{3}}{T}\Big) = \tilde{O}(\frac{1}{T^{1/3}}), \quad (15)$$

*where* $M' = \frac{\rho(F(x_1)-F^*)}{k\gamma} + \frac{5b_1L_0^2\Delta_0}{k\lambda\mu} + \frac{2m^{1/3}\sigma^2}{k^2} + 2k^2(c_1^2+c_2^2)\sigma^2\ln(m+T) + \frac{6k(m+T)^{1/3}}{T}$.

**Remark 4.** *Clearly, the adaptive matrix* $A_t$ *generated from the above (4), we have* $\frac{\sqrt{\frac{1}{T}\sum_{t=1}^T \mathbb{E}\|A_t\|^2}}{\rho} \leq \frac{G_1+\lambda}{\lambda}$ *as in the existing adaptive gradient methods assuming the standard bounded stochastic gradient* $\|\nabla f(x;\xi)\| \leq G_1$. *Under the same conditions in Theorem 2, based on the metric* $\frac{1}{T}\sum_{t=1}^T \mathbb{E}\|\nabla F(x_t)\|$, *our VR-BiAdam algorithm still has a convergence rate of* $\tilde{O}(\frac{1}{T^{1/3}})$ *and a sample complexity of* $\tilde{O}(\epsilon^{-3})$ *without relying on the large mini-batches.*

### A.4 DETAILED CONVERGENCE ANALYSIS

In this subsection, we provide the detailed convergence analysis of our algorithms. We first review and provide some useful lemmas.

Given a $\rho$-strongly convex function $\phi(x)$, we define a prox-function (Bregman distance) Censor & Lent (1981); Censor & Zenios (1992) associated with $\phi(x)$ as follows:

$$V(z,x) = \phi(z) - \big[\phi(x) + \langle\nabla\phi(x), z-x\rangle\big]. \quad (16)$$

Then we define a generalized projection problem as in Ghadimi et al. (2016):

$$x^* = \arg\min_{z\in\mathcal{X}}\big\{\langle z,w\rangle + \frac{1}{\gamma}V(z,x) + h(z)\big\}, \quad (17)$$

where $\mathcal{X} \subseteq \mathbb{R}^d$, $w \in \mathbb{R}^d$ and $\gamma > 0$. Here $h(x)$ is convex and possibly nonsmooth function. At the same time, we define a generalized gradient as follows:

$$\mathcal{G}_{\mathcal{X}}(x,w,\gamma) = \frac{1}{\gamma}(x-x^*). \quad (18)$$

**Lemma 6.** *(Lemma 1 in Ghadimi et al. (2016)) Let $x^*$ be given in (17). Then, for any $x \in \mathcal{X}$, $w \in \mathbb{R}^d$ and $\gamma > 0$, we have*

$$\langle w, \mathcal{G}_{\mathcal{X}}(x, w, \gamma) \rangle \geq \rho \|\mathcal{G}_{\mathcal{X}}(x, w, \gamma)\|^2 + \frac{1}{\gamma}\big[ h(x^*) - h(x) \big], \tag{19}$$

*where $\rho > 0$ depends on $\rho$-strongly convex function $\phi(x)$.*

When $h(x) = 0$, in the above lemma 6, we have

$$\langle w, \mathcal{G}_{\mathcal{X}}(x, w, \gamma) \rangle \geq \rho \|\mathcal{G}_{\mathcal{X}}(x, w, \gamma)\|^2. \tag{20}$$

**Lemma 7.** *(Restatement of Lemma 5) When the gradient estimator $w_t$ generated from Algorithm 1 or 2, for all $t \geq 1$, we have*

$$\|w_t - \nabla F(x_t)\|^2 \leq L_0^2 \|y^*(x_t) - y_t\|^2 + 2\|w_t - \bar{\nabla} f(x_t, y_t)\|^2, \tag{21}$$

*where $L_0^2 = 8\big(L_f^2 + \frac{L_{gxy}^2 C_{fy}^2}{\mu^2} + \frac{L_{gyy}^2 C_{gxy}^2 C_{fy}^2}{\mu^4} + \frac{L_f^2 C_{gxy}^2}{\mu^2}\big)$.*

*Proof.* We first consider the term $\|\nabla F(x_t) - \bar{\nabla} f(x_t, y_t)\|^2$. Since $\nabla f(x_t, y^*(x_t)) = \nabla F(x_t)$, we have

$$\|\nabla f(x_t, y^*(x_t)) - \bar{\nabla} f(x_t, y_t)\|^2$$
$$= \|\nabla_x f(x_t, y^*(x_t)) - \nabla_{xy}^2 g(x_t, y^*(x_t))\big(\nabla_{yy}^2 g(x_t, y^*(x_t))\big)^{-1} \nabla_y f(x_t, y^*(x))$$
$$\quad - \nabla_x f(x_t, y_t) + \nabla_{xy}^2 g(x_t, y_t)\big(\nabla_{yy}^2 g(x_t, y_t)\big)^{-1} \nabla_y f(x_t, y_t)\|^2$$
$$= \|\nabla_x f(x_t, y^*(x_t)) - \nabla_x f(x_t, y_t) - \nabla_{xy}^2 g(x_t, y^*(x_t))\big(\nabla_{yy}^2 g(x_t, y^*(x_t))\big)^{-1} \nabla_y f(x_t, y^*(x_t))$$
$$\quad + \nabla_{xy}^2 g(x_t, y_t)\big(\nabla_{yy}^2 g(x_t, y^*(x_t))\big)^{-1} \nabla_y f(x_t, y^*(x_t)) - \nabla_{xy}^2 g(x_t, y_t)\big(\nabla_{yy}^2 g(x_t, y^*(x_t))\big)^{-1} \nabla_y f(x_t, y^*(x_t))$$
$$\quad + \nabla_{xy}^2 g(x_t, y_t)\big(\nabla_{yy}^2 g(x_t, y_t)\big)^{-1} \nabla_y f(x_t, y^*(x_t)) - \nabla_{xy}^2 g(x_t, y_t)\big(\nabla_{yy}^2 g(x_t, y_t)\big)^{-1} \nabla_y f(x_t, y^*(x_t))$$
$$\quad + \nabla_{xy}^2 g(x_t, y_t)\big(\nabla_{yy}^2 g(x_t, y_t)\big)^{-1} \nabla_y f(x_t, y_t)\|^2$$
$$\leq 4\|\nabla_x f(x_t, y^*(x_t)) - \nabla_x f(x_t, y_t)\|^2 + \frac{4C_{fy}^2}{\mu^2}\|\nabla_{xy}^2 g(x_t, y^*(x_t)) - \nabla_{xy}^2 g(x_t, y_t)\|^2$$
$$\quad + \frac{4C_{gxy}^2 C_{fy}^2}{\mu^4}\|\nabla_{yy}^2 g(x_t, y^*(x_t)) - \nabla_{yy}^2 g(x_t, y_t)\|^2 + \frac{4C_{gxy}^2}{\mu^2}\|\nabla_y f(x_t, y^*(x_t)) - \nabla_y f(x_t, y_t)\|^2$$
$$\leq 4\big(L_f^2 + \frac{L_{gxy}^2 C_{fy}^2}{\mu^2} + \frac{L_{gyy}^2 C_{gxy}^2 C_{fy}^2}{\mu^4} + \frac{L_f^2 C_{gxy}^2}{\mu^2}\big)\|y^*(x_t) - y_t\|^2$$
$$= 4\bar{L}^2 \|y^*(x_t) - y_t\|^2, \tag{22}$$

where the second last inequality is due to Assumptions 1, 2 and 4; the last equality holds by $\bar{L}^2 = L_f^2 + \frac{L_{gxy}^2 C_{fy}^2}{\mu^2} + \frac{L_{gyy}^2 C_{gxy}^2 C_{fy}^2}{\mu^4} + \frac{L_f^2 C_{gxy}^2}{\mu^2}$.

Then we have

$$\|w_t - \nabla F(x_t)\|^2 = \|w_t - \bar{\nabla} f(x_t, y_t) + \bar{\nabla} f(x_t, y_t) - \nabla F(x_t)\|^2$$
$$\leq 2\|w_t - \bar{\nabla} f(x_t, y_t)\|^2 + 2\|\bar{\nabla} f(x_t, y_t) - \nabla F(x_t)\|^2$$
$$\leq 2\|w_t - \bar{\nabla} f(x_t, y_t)\|^2 + 8\bar{L}^2 \|y^*(x_t) - y_t\|^2. \tag{23}$$

$\square$

**Lemma 8.** *Under the Assumptions 1, 2, 4, we have*

$$\|\bar{\nabla} f(x_{t+1}, y_{t+1}) - \bar{\nabla} f(x_t, y_t)\|^2 \leq L_0^2 \big(\|x_{t+1} - x_t\|^2 + \|y_{t+1} - y_t\|^2\big), \tag{24}$$

*where $L_0^2 = 8\big(L_f^2 + \frac{L_{gxy}^2 C_{fy}^2}{\mu^2} + \frac{L_{gyy}^2 C_{gxy}^2 C_{fy}^2}{\mu^4} + \frac{L_f^2 C_{gxy}^2}{\mu^2}\big)$.*

*Proof.*

$$\|\bar{\nabla}f(x_{t+1}, y_{t+1}) - \bar{\nabla}f(x_t, y_t)\|^2$$

$$= \|\nabla_x f(x_{t+1}, y_{t+1}) - \nabla_{xy}^2 g(x_{t+1}, y_{t+1})\big(\nabla_{yy}^2 g(x_{t+1}, y_{t+1})\big)^{-1}\nabla_y f(x_{t+1}, y_{t+1})$$
$$\quad - \nabla_x f(x_t, y_t) + \nabla_{xy}^2 g(x_t, y_t)\big(\nabla_{yy}^2 g(x_t, y_t)\big)^{-1}\nabla_y f(x_t, y_t)\|^2$$

$$= \|\nabla_x f(x_{t+1}, y_{t+1}) - \nabla_x f(x_t, y_t) - \nabla_{xy}^2 g(x_{t+1}, y_{t+1})\big(\nabla_{yy}^2 g(x_{t+1}, y_{t+1})\big)^{-1}\nabla_y f(x_{t+1}, y_{t+1})$$
$$\quad + \nabla_{xy}^2 g(x_t, y_t)\big(\nabla_{yy}^2 g(x_{t+1}, y_{t+1})\big)^{-1}\nabla_y f(x_{t+1}, y_{t+1}) - \nabla_{xy}^2 g(x_t, y_t)\big(\nabla_{yy}^2 g(x_{t+1}, y_{t+1})\big)^{-1}\nabla_y f(x_{t+1}, y_{t+1})$$
$$\quad + \nabla_{xy}^2 g(x_t, y_t)\big(\nabla_{yy}^2 g(x_t, y_t)\big)^{-1}\nabla_y f(x_{t+1}, y_{t+1}) - \nabla_{xy}^2 g(x_t, y_t)\big(\nabla_{yy}^2 g(x_t, y_t)\big)^{-1}\nabla_y f(x_{t+1}, y_{t+1})$$
$$\quad + \nabla_{xy}^2 g(x_t, y_t)\big(\nabla_{yy}^2 g(x_t, y_t)\big)^{-1}\nabla_y f(x_t, y_t)\|^2$$

$$\leq 4\|\nabla_x f(x_{t+1}, y_{t+1}) - \nabla_x f(x_t, y_t)\|^2 + \frac{4C_{fy}^2}{\mu^2}\|\nabla_{xy}^2 g(x_{t+1}, y_{t+1}) - \nabla_{xy}^2 g(x_t, y_t)\|^2$$
$$\quad + \frac{4C_{gxy}^2 C_{fy}^2}{\mu^4}\|\nabla_{yy}^2 g(x_{t+1}, y_{t+1}) - \nabla_{yy}^2 g(x_t, y_t)\|^2 + \frac{4C_{gxy}^2}{\mu^2}\|\nabla_y f(x_{t+1}, y_{t+1}) - \nabla_y f(x_t, y_t)\|^2$$

$$\leq 8L_f^2\big(\|x_{t+1} - x_t\|^2 + \|y_{t+1} - y_t\|^2\big) + \frac{8L_{gxy}^2 C_{fy}^2}{\mu^2}\big(\|x_{t+1} - x_t\|^2 + \|y_{t+1} - y_t\|^2\big)$$
$$\quad + \frac{8L_{gyy}^2 C_{gxy}^2 C_{fy}^2}{\mu^4}\big(\|x_{t+1} - x_t\|^2 + \|y_{t+1} - y_t\|^2\big) + \frac{8L_f^2 C_{gxy}^2}{\mu^2}\big(\|x_{t+1} - x_t\|^2 + \|y_{t+1} - y_t\|^2\big)$$

$$= L_0^2\big(\|x_{t+1} - x_t\|^2 + \|y_{t+1} - y_t\|^2\big), \tag{25}$$

where the first inequality holds by the Assumptions 1 and 4, and the second inequality holds by the Assumption 2.

$\square$

**Lemma 9.** *Suppose that the sequence $\{x_t, y_t\}_{t=1}^T$ be generated from Algorithm 1 or 2. Let $0 < \eta_t \leq 1$ and $0 < \gamma \leq \frac{\rho}{2L\eta_t}$, then we have*

$$F(x_{t+1}) \leq F(x_t) + \frac{\eta_t \gamma}{\rho}\|\nabla F(x_t) - w_t\|^2 - \frac{\rho \eta_t}{2\gamma}\|\tilde{x}_{t+1} - x_t\|^2. \tag{26}$$

*Proof.* According to Lemma 2, the function $F(x)$ is $L$-smooth. Thus we have

$$F(x_{t+1}) \leq F(x_t) + \langle \nabla F(x_t), x_{t+1} - x_t \rangle + \frac{L}{2}\|x_{t+1} - x_t\|^2 \tag{27}$$

$$= F(x_t) + \langle \nabla F(x_t), \eta_t(\tilde{x}_{t+1} - x_t) \rangle + \frac{L}{2}\|\eta_t(\tilde{x}_{t+1} - x_t)\|^2$$

$$= F(x_t) + \eta_t \underbrace{\langle w_t, \tilde{x}_{t+1} - x_t \rangle}_{=T_1} + \eta_t \underbrace{\langle \nabla F(x_t) - w_t, \tilde{x}_{t+1} - x_t \rangle}_{=T_2} + \frac{L\eta_t^2}{2}\|\tilde{x}_{t+1} - x_t\|^2,$$

where the second equality is due to $x_{t+1} = x_t + \eta_t(\tilde{x}_{t+1} - x_t)$.

According to Assumption 7, i.e., $A_t \succ \rho I_d$ for any $t \geq 1$, the function $\phi_t(x) = \frac{1}{2}x^T A_t x$ is $\rho$-strongly convex, then we define a prox-function (a.k.a. Bregman distance) associated with $\phi_t(x)$ as in Censor & Zenios (1992); Ghadimi et al. (2016),

$$V_t(x, x_t) = \phi_t(x) - \big[\phi_t(x_t) + \langle \nabla \phi_t(x_t), x - x_t \rangle\big] = \frac{1}{2}(x - x_t)^T A_t(x - x_t). \tag{28}$$

By using the above Lemma 6 to the problem $\tilde{x}_{t+1} = \arg\min_{x \in \mathcal{X}}\big\{\langle w_t, x \rangle + \frac{1}{2\gamma}(x - x_t)^T A_t(x - x_t)\big\}$ at the line 5 of Algorithm 1 or 2, we have

$$\langle w_t, \frac{1}{\gamma}(x_t - \tilde{x}_{t+1}) \rangle \geq \rho\|\frac{1}{\gamma}(x_t - \tilde{x}_{t+1})\|^2. \tag{29}$$

Then we obtain

$$T_1 = \langle w_t, \tilde{x}_{t+1} - x_t \rangle \le -\frac{\rho}{\gamma} \|\tilde{x}_{t+1} - x_t\|^2. \tag{30}$$

Next, consider the bound of the term $T_2$, we have

$$
\begin{aligned}
T_2 &= \langle \nabla F(x_t) - w_t, \tilde{x}_{t+1} - x_t \rangle \\
&\le \|\nabla F(x_t) - w_t\| \cdot \|\tilde{x}_{t+1} - x_t\| \\
&\le \frac{\gamma}{\rho} \|\nabla F(x_t) - w_t\|^2 + \frac{\rho}{4\gamma} \|\tilde{x}_{t+1} - x_t\|^2,
\end{aligned}
\tag{31}
$$

where the first inequality is due to the Cauchy-Schwarz inequality and the last is due to Young's inequality. By combining the above inequalities (27), (30) with (31), we obtain

$$F(x_{t+1}) \le F(x_t) + \eta_t \langle \nabla F(x_t) - w_t, \tilde{x}_{t+1} - x_t \rangle + \eta_t \langle w_t, \tilde{x}_{t+1} - x_t \rangle + \frac{L\eta_t^2}{2} \|\tilde{x}_{t+1} - x_t\|^2$$

$$\le F(x_t) + \frac{\eta_t \gamma}{\rho} \|\nabla F(x_t) - w_t\|^2 + \frac{\rho \eta_t}{4\gamma} \|\tilde{x}_{t+1} - x_t\|^2 - \frac{\rho \eta_t}{\gamma} \|\tilde{x}_{t+1} - x_t\|^2 + \frac{L\eta_t^2}{2} \|\tilde{x}_{t+1} - x_t\|^2$$

$$= F(x_t) + \frac{\eta_t \gamma}{\rho} \|\nabla F(x_t) - w_t\|^2 - \frac{\rho \eta_t}{2\gamma} \|\tilde{x}_{t+1} - x_t\|^2 - \left(\frac{\rho \eta_t}{4\gamma} - \frac{L\eta_t^2}{2}\right) \|\tilde{x}_{t+1} - x_t\|^2$$

$$\le F(x_t) + \frac{\eta_t \gamma}{\rho} \|\nabla F(x_t) - w_t\|^2 - \frac{\rho \eta_t}{2\gamma} \|\tilde{x}_{t+1} - x_t\|^2, \tag{32}$$

where the last inequality is due to $0 < \gamma \le \frac{\rho}{2L\eta_t}$.

$\square$

**Lemma 10.** *Suppose the sequence $\{x_t, y_t\}_{t=1}^{T}$ be generated from Algorithm 1 or 2. Under the above assumptions, given $0 < \eta_t \le 1$, $B_t = b_t I_p$ ($b_u \ge b_t \ge b_l > 0$) for all $t \ge 1$, and $0 < \lambda \le \frac{b_l}{6L_g}$, we have*

$$\|y_{t+1} - y^*(x_{t+1})\|^2 \le (1 - \frac{\eta_t \mu \lambda}{4b_t}) \|y_t - y^*(x_t)\|^2 - \frac{3\eta_t}{4} \|\tilde{y}_{t+1} - y_t\|^2$$

$$+ \frac{25\eta_t \lambda}{6\mu b_t} \|\nabla_y g(x_t, y_t) - v_t\|^2 + \frac{25\kappa^2 \eta_t b_t}{6\mu \lambda} \|\tilde{x}_{t+1} - x_t\|^2, \tag{33}$$

*where $\kappa = L_g/\mu$.*

*Proof.* According to Assumption 1, i.e., the function $g(x, y)$ is $\mu$-strongly convex w.r.t $y$, we have

$$
\begin{aligned}
g(x_t, y) &\ge g(x_t, y_t) + \langle \nabla_y g(x_t, y_t), y - y_t \rangle + \frac{\mu}{2} \|y - y_t\|^2 \\
&= g(x_t, y_t) + \langle v_t, y - \tilde{y}_{t+1} \rangle + \langle \nabla_y g(x_t, y_t) - v_t, y - \tilde{y}_{t+1} \rangle \\
&\quad + \langle \nabla_y g(x_t, y_t), \tilde{y}_{t+1} - y_t \rangle + \frac{\mu}{2} \|y - y_t\|^2.
\end{aligned}
\tag{34}
$$

According to the Assumption 2, i.e., the function $g(x, y)$ is $L_g$-smooth, we have

$$g(x_t, \tilde{y}_{t+1}) \le g(x_t, y_t) + \langle \nabla_y g(x_t, y_t), \tilde{y}_{t+1} - y_t \rangle + \frac{L_g}{2} \|\tilde{y}_{t+1} - y_t\|^2. \tag{35}$$

Combining the about inequalities (34) with (35), we have

$$g(x_t, y) \ge g(x_t, \tilde{y}_{t+1}) + \langle v_t, y - \tilde{y}_{t+1} \rangle + \langle \nabla_y g(x_t, y_t) - v_t, y - \tilde{y}_{t+1} \rangle$$

$$+ \frac{\mu}{2} \|y - y_t\|^2 - \frac{L_g}{2} \|\tilde{y}_{t+1} - y_t\|^2. \tag{36}$$

By the optimality condition of the problem $\tilde{y}_{t+1} = \arg\min_{y \in \mathcal{Y}} \left\{ \langle v_t, y \rangle + \frac{1}{2\lambda}(y - y_t)^T B_t (y - y_t) \right\}$ at the line 6 of Algorithm 1 or 2, given $B_t = b_t I_p$, we have

$$\langle v_t + \frac{b_t}{\lambda}(\tilde{y}_{t+1} - y_t), y - \tilde{y}_{t+1} \rangle \ge 0, \quad \forall y \in \mathcal{Y}. \tag{37}$$

Then we obtain

$$\langle v_t, y - \tilde{y}_{t+1} \rangle \geq \frac{b_t}{\lambda} \langle \tilde{y}_{t+1} - y_t, \tilde{y}_{t+1} - y \rangle$$

$$= \frac{b_t}{\lambda} \|\tilde{y}_{t+1} - y_t\|^2 + \frac{b_t}{\lambda} \langle \tilde{y}_{t+1} - y_t, y_t - y \rangle. \tag{38}$$

By pugging the inequalities (38) into (36), we have

$$g(x_t, y) \geq g(x_t, \tilde{y}_{t+1}) + \frac{b_t}{\lambda} \langle \tilde{y}_{t+1} - y_t, y_t - y \rangle + \frac{b_t}{\lambda} \|\tilde{y}_{t+1} - y_t\|^2$$

$$+ \langle \nabla_y g(x_t, y_t) - v_t, y - \tilde{y}_{t+1} \rangle + \frac{\mu}{2} \|y - y_t\|^2 - \frac{L_g}{2} \|\tilde{y}_{t+1} - y_t\|^2. \tag{39}$$

Let $y = y^*(x_t)$, then we have

$$g(x_t, y^*(x_t)) \geq g(x_t, \tilde{y}_{t+1}) + \frac{b_t}{\lambda} \langle \tilde{y}_{t+1} - y_t, y_t - y^*(x_t) \rangle + (\frac{b_t}{\lambda} - \frac{L_g}{2}) \|\tilde{y}_{t+1} - y_t\|^2$$

$$+ \langle \nabla_y g(x_t, y_t) - v_t, y^*(x_t) - \tilde{y}_{t+1} \rangle + \frac{\mu}{2} \|y^*(x_t) - y_t\|^2. \tag{40}$$

Due to the strongly-convexity of $g(\cdot, y)$ and $y^*(x_t) = \arg\min_{y \in \mathcal{Y}} g(x_t, y)$, we have $g(x_t, y^*(x_t)) \leq g(x_t, \tilde{y}_{t+1})$. Thus, we obtain

$$0 \geq \frac{b_t}{\lambda} \langle \tilde{y}_{t+1} - y_t, y_t - y^*(x_t) \rangle + \langle \nabla_y g(x_t, y_t) - v_t, y^*(x_t) - \tilde{y}_{t+1} \rangle$$

$$+ (\frac{b_t}{\lambda} - \frac{L_g}{2}) \|\tilde{y}_{t+1} - y_t\|^2 + \frac{\mu}{2} \|y^*(x_t) - y_t\|^2. \tag{41}$$

By $y_{t+1} = y_t + \eta_t(\tilde{y}_{t+1} - y_t)$, we have

$$\|y_{t+1} - y^*(x_t)\|^2 = \|y_t + \eta_t(\tilde{y}_{t+1} - y_t) - y^*(x_t)\|^2$$

$$= \|y_t - y^*(x_t)\|^2 + 2\eta_t \langle \tilde{y}_{t+1} - y_t, y_t - y^*(x_t) \rangle + \eta_t^2 \|\tilde{y}_{t+1} - y_t\|^2. \tag{42}$$

Then we obtain

$$\langle \tilde{y}_{t+1} - y_t, y_t - y^*(x_t) \rangle = \frac{1}{2\eta_t} \|y_{t+1} - y^*(x_t)\|^2 - \frac{1}{2\eta_t} \|y_t - y^*(x_t)\|^2 - \frac{\eta_t}{2} \|\tilde{y}_{t+1} - y_t\|^2. \tag{43}$$

Consider the upper bound of the term $\langle \nabla_y g(x_t, y_t) - v_t, y^*(x_t) - \tilde{y}_{t+1} \rangle$, we have

$$\langle \nabla_y g(x_t, y_t) - v_t, y^*(x_t) - \tilde{y}_{t+1} \rangle$$

$$= \langle \nabla_y g(x_t, y_t) - v_t, y^*(x_t) - y_t \rangle + \langle \nabla_y g(x_t, y_t) - v_t, y_t - \tilde{y}_{t+1} \rangle$$

$$\geq -\frac{1}{\mu} \|\nabla_y g(x_t, y_t) - v_t\|^2 - \frac{\mu}{4} \|y^*(x_t) - y_t\|^2 - \frac{1}{\mu} \|\nabla_y g(x_t, y_t) - v_t\|^2 - \frac{\mu}{4} \|y_t - \tilde{y}_{t+1}\|^2$$

$$= -\frac{2}{\mu} \|\nabla_y g(x_t, y_t) - v_t\|^2 - \frac{\mu}{4} \|y^*(x_t) - y_t\|^2 - \frac{\mu}{4} \|y_t - \tilde{y}_{t+1}\|^2. \tag{44}$$

By plugging the inequalities (43) and (44) into (41), we obtain

$$\frac{b_t}{2\eta_t \lambda} \|y_{t+1} - y^*(x_t)\|^2$$

$$\leq (\frac{b_t}{2\eta_t \lambda} - \frac{\mu}{4}) \|y_t - y^*(x_t)\|^2 + (\frac{b_t \eta_t}{2\lambda} + \frac{\mu}{4} + \frac{L_g}{2} - \frac{b_t}{\lambda}) \|\tilde{y}_{t+1} - y_t\|^2 + \frac{2}{\mu} \|\nabla_y g(x_t, y_t) - v_t\|^2$$

$$\leq (\frac{b_t}{2\eta_t \lambda} - \frac{\mu}{4}) \|y_t - y^*(x_t)\|^2 + (\frac{3L_g}{4} - \frac{b_t}{2\lambda}) \|\tilde{y}_{t+1} - y_t\|^2 + \frac{2}{\mu} \|\nabla_y g(x_t, y_t) - v_t\|^2$$

$$= (\frac{b_t}{2\eta_t \lambda} - \frac{\mu}{4}) \|y_t - y^*(x_t)\|^2 - (\frac{3b_t}{8\lambda} + \frac{b_t}{8\lambda} - \frac{3L_g}{4}) \|\tilde{y}_{t+1} - y_t\|^2 + \frac{2}{\mu} \|\nabla_y g(x_t, y_t) - v_t\|^2$$

$$\leq (\frac{b_t}{2\eta_t \lambda} - \frac{\mu}{4}) \|y_t - y^*(x_t)\|^2 - \frac{3b_t}{8\lambda} \|\tilde{y}_{t+1} - y_t\|^2 + \frac{2}{\mu} \|\nabla_y g(x_t, y_t) - v_t\|^2, \tag{45}$$

where the second inequality holds by $L_g \geq \mu$ and $0 < \eta_t \leq 1$, and the last inequality is due to $0 < \lambda \leq \frac{b_l}{6L_g} \leq \frac{b_t}{6L_g}$. It implies that

$$\|y_{t+1} - y^*(x_t)\|^2 \leq (1 - \frac{\eta_t \mu \lambda}{2b_t})\|y_t - y^*(x_t)\|^2 - \frac{3\eta_t}{4}\|\tilde{y}_{t+1} - y_t\|^2 + \frac{4\eta_t \lambda}{\mu b_t}\|\nabla_y g(x_t, y_t) - v_t\|^2.$$
(46)

Next, we decompose the term $\|y_{t+1} - y^*(x_{t+1})\|^2$ as follows:

$$\|y_{t+1} - y^*(x_{t+1})\|^2 = \|y_{t+1} - y^*(x_t) + y^*(x_t) - y^*(x_{t+1})\|^2$$
$$= \|y_{t+1} - y^*(x_t)\|^2 + 2\langle y_{t+1} - y^*(x_t), y^*(x_t) - y^*(x_{t+1})\rangle + \|y^*(x_t) - y^*(x_{t+1})\|^2$$
$$\leq (1 + \frac{\eta_t \mu \lambda}{4b_t})\|y_{t+1} - y^*(x_t)\|^2 + (1 + \frac{4b_t}{\eta_t \mu \lambda})\|y^*(x_t) - y^*(x_{t+1})\|^2$$
$$\leq (1 + \frac{\eta_t \mu \lambda}{4b_t})\|y_{t+1} - y^*(x_t)\|^2 + (1 + \frac{4b_t}{\eta_t \mu \lambda})\kappa^2\|x_t - x_{t+1}\|^2, \quad (47)$$

where the first inequality holds by Cauchy-Schwarz inequality and Young's inequality, and the second inequality is due to Lemma 3, and the last equality holds by $x_{t+1} = x_t + \eta_t(\tilde{x}_{t+1} - x_t)$.

By combining the above inequalities (46) and (47), we have

$$\|y_{t+1} - y^*(x_{t+1})\|^2 \leq (1 + \frac{\eta_t \mu \lambda}{4b_t})(1 - \frac{\eta_t \mu \lambda}{2b_t})\|y_t - y^*(x_t)\|^2 - (1 + \frac{\eta_t \mu \lambda}{4b_t})\frac{3\eta_t}{4}\|\tilde{y}_{t+1} - y_t\|^2$$
$$+ (1 + \frac{\eta_t \mu \lambda}{4b_t})\frac{4\eta_t \lambda}{\mu b_t}\|\nabla_y g(x_t, y_t) - v_t\|^2 + (1 + \frac{4b_t}{\eta_t \mu \lambda})\kappa^2\|x_t - x_{t+1}\|^2.$$

Since $0 < \eta_t \leq 1$, $0 < \lambda \leq \frac{b_l}{6L_g} \leq \frac{b_t}{6L_g}$ and $L_g \geq \mu$, we have $\lambda \leq \frac{b_t}{6L_g} \leq \frac{b_t}{6\mu}$ and $\eta_t \leq 1 \leq \frac{b_t}{6\mu\lambda}$. Then we have

$$(1 + \frac{\eta_t \mu \lambda}{4b_t})(1 - \frac{\eta_t \mu \lambda}{2b_t}) = 1 - \frac{\eta_t \mu \lambda}{2b_t} + \frac{\eta_t \mu \lambda}{4b_t} - \frac{\eta_t^2 \mu^2 \lambda^2}{8b_t^2} \leq 1 - \frac{\eta_t \mu \lambda}{4b_t},$$

$$-(1 + \frac{\eta_t \mu \lambda}{4b_t})\frac{3\eta_t}{4} \leq -\frac{3\eta_t}{4},$$

$$(1 + \frac{\eta_t \mu \lambda}{4b_t})\frac{4\eta_t \lambda}{\mu b_t} \leq (1 + \frac{1}{24})\frac{4\eta_t \lambda}{\mu b_t} = \frac{25\eta_t \lambda}{6\mu b_t},$$

$$(1 + \frac{4b_t}{\eta_t \mu \lambda})\kappa^2 \leq \frac{b_t \kappa^2}{6\eta_t \mu \lambda} + \frac{4b_t \kappa^2}{\eta_t \mu \lambda} = \frac{25 b_t \kappa^2}{6\eta_t \mu \lambda},$$

where the second last inequality is due to $\frac{\eta_t \mu \lambda}{b_t} \leq \frac{1}{6}$ and the last inequality holds by $\frac{b_t}{6\mu\lambda\eta_t} \geq 1$. By using $x_{t+1} = x_t + \eta_t(\tilde{x}_{t+1} - x_t)$, then we have

$$\|y_{t+1} - y^*(x_{t+1})\|^2 \leq (1 - \frac{\eta_t \mu \lambda}{4b_t})\|y_t - y^*(x_t)\|^2 - \frac{3\eta_t}{4}\|\tilde{y}_{t+1} - y_t\|^2$$
$$+ \frac{25\eta_t \lambda}{6\mu b_t}\|\nabla_y g(x_t, y_t) - v_t\|^2 + \frac{25\kappa^2 \eta_t b_t}{6\mu \lambda}\|\tilde{x}_{t+1} - x_t\|^2. \quad (48)$$

$\square$

## A.5 CONVERGENCE ANALYSIS OF BIADAM ALGORITHM

In the subsection, we provide the detail convergence analysis of BiAdam algorithm. For notational simplicity, let $R_t = R(x_t, y_t)$ for all $t \geq 1$.

**Lemma 11.** *Assume that the stochastic partial derivatives $v_{t+1}$, and $w_{t+1}$ be generated from Algorithm 1, we have*

$$\mathbb{E}\|w_{t+1} - \bar{\nabla} f(x_{t+1}, y_{t+1}) - R_{t+1}\|^2 \leq (1 - \beta_{t+1})\mathbb{E}\|w_t - \bar{\nabla} f(x_t, y_t) - R_t\|^2 + \beta_{t+1}^2 \sigma^2 \quad (49)$$
$$+ \frac{3L_0^2 \eta_t^2}{\beta_{t+1}}\left(\|\tilde{x}_{t+1} - x_t\|^2 + \|\tilde{y}_{t+1} - y_t\|^2\right)$$
$$+ \frac{3}{\beta_{t+1}}\left(\|R_t\|^2 + \|R_{t+1}\|^2\right),$$

$$\mathbb{E}\|v_{t+1} - \nabla_y g(x_{t+1}, y_{t+1})\|^2 \le (1 - \alpha_{t+1})\mathbb{E}\|v_t - \nabla_y g(x_t, y_t)\|^2 + \alpha_{t+1}^2 \sigma^2 \qquad (50)$$
$$+ 2L_g^2 \eta_t^2 / \alpha_{t+1}\big(\mathbb{E}\|\tilde{x}_{t+1} - x_t\|^2 + \mathbb{E}\|\tilde{y}_{t+1} - y_t\|^2\big),$$

where $L_0^2 = 8\big(L_f^2 + \frac{L_{gxy}^2 C_{fy}^2}{\mu^2} + \frac{L_{gyy}^2 C_{gxy}^2 C_{fy}^2}{\mu^4} + \frac{L_f^2 C_{gxy}^2}{\mu^2}\big)$ and $R_t = \bar{\nabla} f(x_t, y_t) - \mathbb{E}_{\bar{\xi}}[\bar{\nabla} f(x_t, y_t; \bar{\xi})]$ for all $t \ge 1$.

*Proof.* Without loss of generality, we only prove the term $\mathbb{E}\|w_{t+1} - \bar{\nabla} f(x_{t+1}, y_{t+1}) - R_{t+1}\|^2$. The other term is similar for this term. Since $w_{t+1} = \beta_{t+1} \bar{\nabla} f(x_{t+1}, y_{t+1}; \bar{\xi}_{t+1}) + (1 - \beta_{t+1})w_t$, we have

$$\mathbb{E}\|w_{t+1} - \bar{\nabla} f(x_{t+1}, y_{t+1}) - R_{t+1}\|^2 \qquad (51)$$
$$= \mathbb{E}\|\beta_{t+1} \bar{\nabla} f(x_{t+1}, y_{t+1}; \bar{\xi}_{t+1}) + (1 - \beta_{t+1})w_t - \bar{\nabla} f(x_{t+1}, y_{t+1}) - R_{t+1}\|^2$$
$$= \mathbb{E}\|(1 - \beta_{t+1})(w_t - \bar{\nabla} f(x_t, y_t) - R_t) + \beta_{t+1}\big(\bar{\nabla} f(x_{t+1}, y_{t+1}; \bar{\xi}_{t+1}) - \bar{\nabla} f(x_{t+1}, y_{t+1}) - R_{t+1}\big)$$
$$+ (1 - \beta_{t+1})\big(\bar{\nabla} f(x_t, y_t) + R_t - (\bar{\nabla} f(x_{t+1}, y_{t+1}) + R_{t+1})\big)\|^2$$
$$= \beta_{t+1}^2 \mathbb{E}\|\bar{\nabla} f(x_{t+1}, y_{t+1}; \bar{\xi}_{t+1}) - \bar{\nabla} f(x_{t+1}, y_{t+1}) - R_{t+1}\|^2$$
$$+ \mathbb{E}\|(1 - \beta_{t+1})\big(w_t - \bar{\nabla} f(x_t, y_t) - R_t + \bar{\nabla} f(x_t, y_t) + R_t - (\bar{\nabla} f(x_{t+1}, y_{t+1}) + R_{t+1})\big)\|^2$$
$$\le \beta_{t+1}^2 \mathbb{E}\|\bar{\nabla} f(x_{t+1}, y_{t+1}; \bar{\xi}_{t+1}) - \bar{\nabla} f(x_{t+1}, y_{t+1}) - R_{t+1}\|^2 + (1 - \beta_{t+1})^2(1 + \beta_{t+1})\mathbb{E}\|w_t - \bar{\nabla} f(x_t, y_t) - R_t\|^2$$
$$+ (1 - \beta_{t+1})^2(1 + \frac{1}{\beta_{t+1}})\mathbb{E}\|\bar{\nabla} f(x_t, y_t) + R_t - (\bar{\nabla} f(x_{t+1}, y_{t+1}) + R_{t+1})\|^2$$
$$\le (1 - \beta_{t+1})\mathbb{E}\|w_t - \bar{\nabla} f(x_t, y_t) - R_t\|^2 + \beta_{t+1}^2 \sigma^2 + \frac{1}{\beta_{t+1}}\|\bar{\nabla} f(x_t, y_t) + R_t - (\bar{\nabla} f(x_{t+1}, y_{t+1}) + R_{t+1})\|^2$$
$$\le (1 - \beta_{t+1})\mathbb{E}\|w_t - \bar{\nabla} f(x_t, y_t) - R_t\|^2 + \beta_{t+1}^2 \sigma^2$$
$$+ \frac{3}{\beta_{t+1}}\|\bar{\nabla} f(x_{t+1}, y_{t+1}) - \bar{\nabla} f(x_t, y_t)\|^2 + \frac{3}{\beta_{t+1}}\big(\|R_t\|^2 + \|R_{t+1}\|^2\big)$$
$$\le (1 - \beta_{t+1})\mathbb{E}\|w_t - \bar{\nabla} f(x_t, y_t) - R_t\|^2 + \beta_{t+1}^2 \sigma^2 + \frac{3}{\beta_{t+1}}\big(\|R_t\|^2 + \|R_{t+1}\|^2\big)$$
$$+ \frac{3L_0^2 \eta_t^2}{\beta_{t+1}}\big(\|\tilde{x}_{t+1} - x_t\|^2 + \|\tilde{y}_{t+1} - y_t\|^2\big), \qquad (52)$$

where the third equality is due to $\mathbb{E}_{\bar{\xi}_{t+1}}[\bar{\nabla} f(x_{t+1}, y_{t+1}; \bar{\xi}_{t+1})] = \bar{\nabla} f(x_{t+1}, y_{t+1}) + R_{t+1}$; the second last inequality holds by $0 \le \beta_{t+1} \le 1$ such that $(1 - \beta_{t+1})^2(1 + \beta_{t+1}) = 1 - \beta_{t+1} - \beta_{t+1}^2 + \beta_{t+1}^3 \le 1 - \beta_{t+1}$ and $(1 - \beta_{t+1})^2(1 + \frac{1}{\beta_{t+1}}) \le (1 - \beta_{t+1})(1 + \frac{1}{\beta_{t+1}}) = -\beta_{t+1} + \frac{1}{\beta_{t+1}} \le \frac{1}{\beta_{t+1}}$, and the last inequality holds by the above Lemma 8 and $x_{t+1} = x_t + \eta_t(\tilde{x}_{t+1} - x_t)$, $y_{t+1} = y_t + \eta_t(\tilde{y}_{t+1} - y_t)$. $\qquad\square$

**Theorem 5.** *(Restatement of Theorem 1) Under the above Assumptions (1, 2, 4, 6, 7), in the Algorithm 1, given $\mathcal{X} \subset \mathbb{R}^d$, $\eta_t = \frac{k}{(m+t)^{1/2}}$ for all $t \ge 0$, $\alpha_{t+1} = c_1 \eta_t$, $\beta_{t+1} = c_2 \eta_t$, $m \ge \max\big(k^2, (c_1 k)^2, (c_2 k)^2\big)$, $k > 0$, $\frac{125 L_0^2}{6\mu^2} \le c_1 \le \frac{m^{1/2}}{k}$, $\frac{9}{2} \le c_2 \le \frac{m^{1/2}}{k}$, $0 < \lambda \le \min\big(\frac{15 b_l L_0^2}{4 L_1^2 \mu}, \frac{b_l}{6 L_g}\big)$, $0 < \gamma \le \min\big(\frac{\sqrt{6}\lambda\mu\rho}{\sqrt{6 L_1^2 \lambda^2 \mu^2 + 125 b_u^2 L_0^2 \kappa^2}}, \frac{m^{1/2}\rho}{4 L k}\big)$ and $K = \frac{L_g}{\mu} \log(C_{gxy} C_{fy} T / \mu)$, we have*

$$\frac{1}{T}\sum_{t=1}^{T} \mathbb{E}\|\mathcal{G}_{\mathcal{X}}(x_t, \nabla F(x_t), \gamma)\| \le \frac{1}{T}\sum_{t=1}^{T} \mathbb{E}[\mathcal{G}_t] \le \frac{2\sqrt{3G} m^{1/4}}{T^{1/2}} + \frac{2\sqrt{3G}}{T^{1/4}} + \frac{\sqrt{2}}{T}, \qquad (53)$$

*where $G = \frac{F(x_1) - F^*}{\rho k \gamma} + \frac{5 b_1 L_0^2 \Delta_0}{\rho^2 k \lambda \mu} + \frac{2\sigma^2}{\rho^2 k} + \frac{2 m \sigma^2}{\rho^2 k} \ln(m + T) + \frac{4(m+T)}{9\rho^2 k T^2} + \frac{8k}{\rho^2 T}$, $\Delta_0 = \|y_1 - y^*(x_1)\|^2$ and $L_1^2 = \frac{12 L_g^2 \mu^2}{125 L_0^2} + \frac{2 L_0^2}{3}$.*

*Proof.* Since $\eta_t = \frac{k}{(m+t)^{1/2}}$ on $t$ is decreasing and $m \ge k^2$, we have $\eta_t \le \eta_0 = \frac{k}{m^{1/2}} \le 1$ and $\gamma \le \frac{m^{1/2}\rho}{4 L k} \le \frac{\rho}{2 L \eta_0} \le \frac{\rho}{2 L \eta_t}$ for any $t \ge 0$. Due to $0 < \eta_t \le 1$ and $m \ge (c_1 k)^2$, we have

$\alpha_{t+1} = c_1 \eta_t \leq \frac{c_1 k}{m^{1/2}} \leq 1$. Similarly, due to $m \geq (c_2 k)^2$, we have $\beta_{t+1} \leq 1$. At the same time, we have $c_1, c_2 \leq \frac{m^{1/2}}{k}$. According to Lemma 11, we have

$$
\begin{aligned}
\mathbb{E}\|v_{t+1} &- \nabla_y g(x_{t+1}, y_{t+1})\|^2 - \mathbb{E}\|v_t - \nabla_y g(x_t, y_t)\|^2 \qquad (54) \\
&\leq -c_1 \eta_t \mathbb{E}\|\nabla_y g(x_t, y_t) - v_t\|^2 + 2L_g^2 \eta_t/c_1 \left(\|\tilde{x}_{t+1} - x_t\|^2 + \|\tilde{y}_{t+1} - y_t\|^2\right) + c_1^2 \eta_t^2 \sigma^2 \\
&\leq -\frac{125 L_0^2}{6\mu^2} \mathbb{E}\|\nabla_y g(x_t, y_t) - v_t\|^2 + \frac{12 L_g^2 \mu^2 \eta_t}{125 L_0^2}\left(\|\tilde{x}_{t+1} - x_t\|^2 + \|\tilde{y}_{t+1} - y_t\|^2\right) + \frac{m \eta_t^2 \sigma^2}{k^2},
\end{aligned}
$$

where the above equality holds by $\alpha_{t+1} = c_1 \eta_t$, and the last inequality is due to $\frac{125 L_0^2}{6\mu^2} \leq c_1 \leq \frac{m^{1/2}}{k}$. Similarly, we have

$$
\begin{aligned}
\mathbb{E}\|w_{t+1} &- \bar{\nabla} f(x_{t+1}, y_{t+1}) - R_{t+1}\|^2 - \mathbb{E}\|w_t - \bar{\nabla} f(x_t, y_t) - R_t\|^2 \qquad (55) \\
&\leq -\beta_{t+1} \mathbb{E}\|w_t - \bar{\nabla} f(x_t, y_t) - R_t\|^2 + \frac{3 L_0^2 \eta_t^2}{\beta_{t+1}}\left(\|\tilde{x}_{t+1} - x_t\|^2 + \|\tilde{y}_{t+1} - y_t\|^2\right) \\
&\quad + \frac{3}{\beta_{t+1}}\left(\|R_t\|^2 + \|R_{t+1}\|^2\right) + \beta_{t+1}^2 \sigma^2 \\
&\leq -\frac{9 \eta_t}{2} \mathbb{E}\|w_t - \bar{\nabla} f(x_t, y_t) - R_t\|^2 + \frac{2 L_0^2 \eta_t}{3}\left(\|\tilde{x}_{t+1} - x_t\|^2 + \|\tilde{y}_{t+1} - y_t\|^2\right) \\
&\quad + \frac{2}{3\eta_t}\left(\|R_t\|^2 + \|R_{t+1}\|^2\right) + \frac{m \eta_t^2 \sigma^2}{k^2},
\end{aligned}
$$

where the last inequality holds by $\beta_{t+1} = c_2 \eta_t$ and $\frac{9}{2} \leq c_2 \leq \frac{m^{1/2}}{k}$.

According to Lemmas 7 and 9, we have

$$
\begin{aligned}
F(x_{t+1}) &- F(x_t) \qquad (56) \\
&\leq \frac{2\eta_t \gamma}{\rho} \|w_t - \bar{\nabla} f(x_t, y_t)\|^2 + \frac{L_0^2 \eta_t \gamma}{\rho} \|y^*(x_t) - y_t\|^2 - \frac{\rho \eta_t}{2\gamma} \|\tilde{x}_{t+1} - x_t\|^2 \\
&\leq \frac{4\eta_t \gamma}{\rho} \|w_t - \bar{\nabla} f(x_t, y_t) - R_t\|^2 + \frac{4\eta_t \gamma}{\rho} \|R_t\|^2 + \frac{L_0^2 \eta_t \gamma}{\rho} \|y^*(x_t) - y_t\|^2 - \frac{\rho \eta_t}{2\gamma} \|\tilde{x}_{t+1} - x_t\|^2.
\end{aligned}
$$

According to Lemma 10, we have

$$
\begin{aligned}
\|y_{t+1} &- y^*(x_{t+1})\|^2 - \|y_t - y^*(x_t)\|^2 \qquad (57) \\
&\leq -\frac{\eta_t \mu \lambda}{4 b_t} \|y_t - y^*(x_t)\|^2 - \frac{3\eta_t}{4} \|\tilde{y}_{t+1} - y_t\|^2 + \frac{25\eta_t \lambda}{6\mu b_t} \|\nabla_y g(x_t, y_t) - v_t\|^2 + \frac{25\kappa^2 \eta_t b_t}{6\mu\lambda} \|\tilde{x}_{t+1} - x_t\|^2.
\end{aligned}
$$

Next, we define a *Lyapunov* function (i.e., potential function), for any $t \geq 1$,

$$
\Gamma_t = \mathbb{E}\left[F(x_t) + \frac{5 b_t L_0^2 \gamma}{\lambda \mu \rho} \|y_t - y^*(x_t)\|^2 + \frac{\gamma}{\rho}\left(\|v_t - \nabla_y g(x_t, y_t)\|^2 + \|w_t - \bar{\nabla} f(x_t, y_t) - R_t\|^2\right)\right].
$$

For notational simplicity, let $L_1^2 = \frac{12L_g^2\mu^2}{125L_0^2} + \frac{2L_0^2}{3}$. Then we have

$$
\begin{aligned}
&\Gamma_{t+1} - \Gamma_t \\
&= F(x_{t+1}) - F(x_t) + \frac{5b_t L_0^2 \gamma}{\lambda\mu\rho}\big(\|y_{t+1} - y^*(x_{t+1})\|^2 - \|y_t - y^*(x_t)\|^2\big) + \frac{\gamma}{\rho}\big(\|v_{t+1} - \nabla_y g(x_{t+1}, y_{t+1})\|^2 \\
&\quad - \|v_t - \nabla_y g(x_t, y_t)\|^2 + \|w_{t+1} - \bar{\nabla}f(x_{t+1}, y_{t+1}) - R_{t+1}\|^2 - \|w_t - \bar{\nabla}f(x_t, y_t) - R_t\|^2\big) \\
&\leq \frac{L_0^2 \eta_t \gamma}{\rho}\|y^*(x_t) - y_t\|^2 + \frac{4\eta_t\gamma}{\rho}\|w_t - \bar{\nabla}f(x_t, y_t) - R_t\|^2 + \frac{4\eta_t\gamma}{\rho}\|R_t\|^2 - \frac{\rho\eta_t}{2\gamma}\|\tilde{x}_{t+1} - x_t\|^2 \\
&\quad + \frac{5b_t L_0^2 \gamma}{\lambda\mu\rho}\Big(-\frac{\eta_t\mu\lambda}{4b_t}\|y_t - y^*(x_t)\|^2 - \frac{3\eta_t}{4}\|\tilde{y}_{t+1} - y_t\|^2 + \frac{25\eta_t\lambda}{6\mu b_t}\|\nabla_y g(x_t, y_t) - v_t\|^2 + \frac{25\kappa^2\eta_t b_t}{6\mu\lambda}\|\tilde{x}_{t+1} - x_t\|^2\Big) \\
&\quad + \frac{\gamma}{\rho}\Big(-\frac{125 L_0^2}{6\mu^2}\mathbb{E}\|\nabla_y g(x_t, y_t) - v_t\|^2 + \frac{12L_g^2\mu^2\eta_t}{125L_0^2}\big(\|\tilde{x}_{t+1} - x_t\|^2 + \|\tilde{y}_{t+1} - y_t\|^2\big) + \frac{m\eta_t^2\sigma^2}{k^2} \\
&\quad - \frac{9\eta_t}{2}\mathbb{E}\|w_t - \bar{\nabla}f(x_t, y_t) - R_t\|^2 + \frac{2L_0^2\eta_t}{3}\big(\|\tilde{x}_{t+1} - x_t\|^2 + \|\tilde{y}_{t+1} - y_t\|^2\big) + \frac{2}{3\eta_t}\big(\|R_t\|^2 + \|R_{t+1}\|^2\big) + \frac{m\eta_t^2\sigma^2}{k^2}\Big) \\
&= -\frac{\gamma\eta_t}{4\rho}\Big(L_0^2\|y_t - y^*(x_t)\|^2 + 2\mathbb{E}\|w_t - \bar{\nabla}f(x_t, y_t) - R_t\|^2\Big) - \Big(\frac{\rho}{2\gamma} - \frac{L_1^2\gamma}{\rho} - \frac{125 b_t^2 L_0^2\kappa^2\gamma}{6\mu^2\lambda^2\rho}\Big)\eta_t\|\tilde{x}_{t+1} - x_t\|^2 \\
&\quad - \Big(\frac{15 b_t L_0^2 \gamma}{4\lambda\mu\rho} - \frac{L_1^2\gamma}{\rho}\Big)\eta_t\|\tilde{y}_{t+1} - y_t\|^2 + + \frac{2m\gamma\sigma^2}{k^2\rho}\eta_t^2 + \frac{2\gamma}{3\rho\eta_t}\big(\|R_t\|^2 + \|R_{t+1}\|^2\big) + \frac{4\eta_t\gamma}{\rho}\|R_t\|^2 \\
&\leq -\frac{\gamma\eta_t}{4\rho}\Big(L_0^2\|y_t - y^*(x_t)\|^2 + 2\mathbb{E}\|w_t - \bar{\nabla}f(x_t, y_t) - R_t\|^2\Big) - \frac{\rho\eta_t}{4\gamma}\|\tilde{x}_{t+1} - x_t\|^2 + \frac{2m\gamma\sigma^2}{k^2\rho}\eta_t^2 \\
&\quad + \frac{2\gamma}{3\rho\eta_t}\big(\|R_t\|^2 + \|R_{t+1}\|^2\big) + \frac{4\eta_t\gamma}{\rho}\|R_t\|^2, \tag{58}
\end{aligned}
$$

where the first inequality holds by the above inequalities (54), (55), (56) and (57); the last inequality is due to $0 < \gamma \leq \frac{\sqrt{6}\lambda\mu\rho}{\sqrt{6L_1^2\lambda^2\mu^2 + 125 b_u^2 L_0^2\kappa^2}} \leq \frac{\sqrt{6}\lambda\mu\rho}{\sqrt{6L_1^2\lambda^2\mu^2 + 125 b_t^2 L_0^2\kappa^2}}$, $0 < \lambda \leq \frac{15 b_t L_0^2}{4L_1^2\mu} \leq \frac{15 b_t L_0^2}{4L_1^2\mu}$ for all $t \geq 1$.

Let $\Phi_t = L_0^2\|y_t - y^*(x_t)\|^2 + 2\|w_t - \bar{\nabla}f(x_t, y_t) - R_t\|^2$, we have

$$
\frac{\gamma\eta_t}{4\rho}\Phi_t + \frac{\rho\eta_t}{4\gamma}\|\tilde{x}_{t+1} - x_t\|^2 \leq \Gamma_t - \Gamma_{t+1} + \frac{2m\gamma\sigma^2}{k^2\rho}\eta_t^2 + \frac{2\gamma}{3\rho\eta_t}\big(\|R_t\|^2 + \|R_{t+1}\|^2\big) + \frac{4\gamma\eta_t}{\rho}\|R_t\|^2. \tag{59}
$$

Taking average over $t = 1, 2, \cdots, T$ on both sides of (59), we have

$$
\begin{aligned}
\frac{1}{T}\sum_{t=1}^{T}\mathbb{E}\big[\frac{\eta_t}{4}\Phi_t + \frac{\rho^2\eta_t}{4\gamma^2}\|\tilde{x}_{t+1} - x_t\|^2\big] &\leq \sum_{t=1}^{T}\frac{\rho(\Gamma_t - \Gamma_{t+1})}{T\gamma} + \frac{1}{T}\sum_{t=1}^{T}\frac{2m\sigma^2}{k^2}\eta_t^2 \\
&\quad + \frac{1}{T}\sum_{t=1}^{T}\Big(\frac{2}{3\eta_t}\big(\|R_t\|^2 + \|R_{t+1}\|^2\big) + 4\eta_t\|R_t\|^2\Big).
\end{aligned}
$$

Given $x_1 \in \mathcal{X}$ and $y_1 \in \mathcal{Y}$, let $\Delta_0 = \|y_1 - y^*(x_1)\|^2$, we have

$$
\begin{aligned}
\Gamma_1 &= \mathbb{E}\big[F(x_t) + \frac{5b_1 L_0^2\gamma}{\lambda\mu\rho}\|y_1 - y^*(x_1)\|^2 + \frac{\gamma}{\rho}\big(\|v_1 - \nabla_y g(x_1, y_1)\|^2 + \|w_1 - \bar{\nabla}f(x_1, y_1) - R_1\|^2\big)\big] \\
&\leq F(x_1) + \frac{5b_1 L_0^2\gamma\Delta_0}{\lambda\mu\rho} + \frac{2\gamma\sigma^2}{\rho}, \tag{60}
\end{aligned}
$$

where the last inequality holds by Assumption 2. Since $\eta_t$ is decreasing on $t$, i.e., $\eta_T^{-1} \geq \eta_t^{-1}$ for any $0 \leq t \leq T$, we have

$$
\frac{1}{T} \sum_{t=1}^{T} \mathbb{E}\Big(\frac{\Phi_t}{4} + \frac{\rho^2}{4\gamma^2}\|\tilde{x}_{t+1} - x_t\|^2\Big)
$$

$$
\leq \frac{\rho}{T\gamma\eta_T} \sum_{t=1}^{T}\big(\Gamma_t - \Gamma_{t+1}\big) + \frac{1}{T\eta_T}\sum_{t=1}^{T}\frac{2m\sigma^2}{k^2}\eta_t^2 + \frac{1}{T}\sum_{t=1}^{T}\Big(\frac{2}{3\eta_t}\big(\|R_t\|^2 + \|R_{t+1}\|^2\big) + 4\eta_t\|R_t\|^2\Big)
$$

$$
\leq \frac{\rho}{T\gamma\eta_T}\Big(F(x_1) + \frac{5b_1 L_0^2 \gamma\Delta_0}{\lambda\mu\rho} + \frac{2\gamma\sigma^2}{\rho} - F^*\Big) + \frac{1}{T\eta_T}\sum_{t=1}^{T}\frac{2m\sigma^2}{k^2}\eta_t^2 + \frac{2}{3T^3}\sum_{t=1}^{T}\frac{1}{\eta_t} + \frac{4}{T^2}\sum_{t=1}^{T}\eta_t
$$

$$
\leq \frac{\rho(F(x_1) - F^*)}{T\gamma\eta_T} + \frac{5b_1 L_0^2\Delta_0}{T\eta_T\lambda\mu} + \frac{2\sigma^2}{T\eta_T} + \frac{2m\sigma^2}{T\eta_T k^2}\int_1^T \frac{k^2}{m+t}dt + \frac{2}{3T^3}\int_1^T \frac{(m+t)^{1/2}}{k}dt
$$

$$
\quad + \frac{4}{T^2}\int_1^T \frac{k}{(m+t)^{1/2}}dt
$$

$$
\leq \frac{\rho(F(x_1) - F^*)}{T\gamma\eta_T} + \frac{5b_1 L_0^2\Delta_0}{T\eta_T\lambda\mu} + \frac{2\sigma^2}{T\eta_T} + \frac{2m\sigma^2}{T\eta_T}\ln(m+T) + \frac{4}{9kT^3}(m+T)^{3/2} + \frac{8k}{T^2}(m+T)^{1/2}
$$

$$
= \Big(\frac{\rho(F(x_1) - F^*)}{k\gamma} + \frac{5b_1 L_0^2\Delta_0}{k\lambda\mu} + \frac{2\sigma^2}{k} + \frac{2m\sigma^2}{k}\ln(m+T) + \frac{4(m+T)}{9kT^2} + \frac{8k}{T}\Big)\frac{(m+T)^{1/2}}{T},
$$

where the second inequality holds by the above inequality (60) and $\|R_t\| \leq \frac{1}{T}$ for all $t \geq 1$ by choosing $K = \frac{L_g}{\mu}\log(C_{gxy}C_{fy}T/\mu)$ in Algorithm 1. Let $G = \frac{F(x_1)-F^*}{\rho k\gamma} + \frac{5b_1 L_0^2\Delta_0}{\rho^2 k\lambda\mu} + \frac{2\sigma^2}{\rho^2 k} + \frac{2m\sigma^2}{\rho^2 k}\ln(m+T) + \frac{4(m+T)}{9\rho^2 kT^2} + \frac{8k}{\rho^2 T}$, we have

$$
\frac{1}{T}\sum_{t=1}^{T}\mathbb{E}\Big[\frac{\Phi_t}{4\rho^2} + \frac{1}{4\gamma^2}\|\tilde{x}_{t+1} - x_t\|^2\Big] \leq \frac{G}{T}(m+T)^{1/2}. \tag{61}
$$

According to the Jensen's inequality, we have

$$
\frac{1}{T}\sum_{t=1}^{T}\mathbb{E}\Big[\frac{1}{2\gamma}\|\tilde{x}_{t+1} - x_t\| + \frac{1}{2\rho}\big(L_0\|y_t - y^*(x_t)\| + \sqrt{2}\|w_t - \bar{\nabla}f(x_t, y_t) - R_t\|\big)\Big]
$$

$$
\leq \Big(\frac{3}{T}\sum_{t=1}^{T}\big(\frac{1}{4\gamma^2}\|\tilde{x}_{t+1} - x_t\|^2 + \frac{L_0^2}{4\rho^2}\|y_t - y^*(x_t)\|^2 + \frac{2}{4\rho^2}\mathbb{E}\|w_t - \bar{\nabla}f(x_t, y_t) - R_t\|^2\big)\Big)^{1/2}
$$

$$
= \Big(\frac{3}{T}\sum_{t=1}^{T}\big(\frac{\Phi_t}{4\rho^2} + \frac{1}{4\gamma^2}\|\tilde{x}_{t+1} - x_t\|^2\big)\Big)^{1/2}
$$

$$
\leq \frac{\sqrt{3G}}{T^{1/2}}(m+T)^{1/4} \leq \frac{\sqrt{3G}m^{1/4}}{T^{1/2}} + \frac{\sqrt{3G}}{T^{1/4}}, \tag{62}
$$

where the last inequality is due to $(a+b)^{1/4} \leq a^{1/4} + b^{1/4}$ for all $a, b > 0$. Thus we have

$$
\frac{1}{T}\sum_{t=1}^{T}\mathbb{E}\Big[\frac{1}{\gamma}\|\tilde{x}_{t+1} - x_t\| + \frac{1}{\rho}\big(L_0\|y_t - y^*(x_t)\| + \sqrt{2}\|w_t - \bar{\nabla}f(x_t, y_t) - R_t\|\big)\Big]
$$

$$
\leq \frac{2\sqrt{3G}m^{1/4}}{T^{1/2}} + \frac{2\sqrt{3G}}{T^{1/4}}. \tag{63}
$$

Since $\|w_t - \bar{\nabla} f(x_t, y_t) - R_t\| \geq \|w_t - \bar{\nabla} f(x_t, y_t)\| - \|R_t\|$, by the above inequality (63), we can obtain

$$
\begin{aligned}
\frac{1}{T} \sum_{t=1}^T \mathbb{E}[\mathcal{G}_t] &= \frac{1}{T} \sum_{t=1}^T \mathbb{E}\Big[\frac{1}{\gamma}\|\tilde{x}_{t+1} - x_t\| + \frac{1}{\rho}\big(L_0\|y_t - y^*(x_t)\| + \sqrt{2}\|w_t - \bar{\nabla} f(x_t, y_t)\|\big)\Big] \\
&\leq \frac{2\sqrt{3G}m^{1/4}}{T^{1/2}} + \frac{2\sqrt{3G}}{T^{1/4}} + \frac{\sqrt{2}}{T} \sum_{t=1}^T \mathbb{E}\|R_t\| \\
&= \frac{2\sqrt{3G}m^{1/4}}{T^{1/2}} + \frac{2\sqrt{3G}}{T^{1/4}} + \frac{\sqrt{2}}{T},
\end{aligned}
\tag{64}
$$

where the last inequality is due to $\|R_t\| \leq \frac{1}{T}$ for all $t \geq 1$ by choosing $K = \frac{L_g}{\mu} \log(C_{gxy} C_{fy} T/\mu)$ in Algorithm 1. According to the above inequality (11), we have

$$
\frac{1}{T} \sum_{t=1}^T \mathbb{E}\|\mathcal{G}_\mathcal{X}(x_t, \nabla F(x_t), \gamma)\| \leq \frac{1}{T} \sum_{t=1}^T \mathbb{E}[\mathcal{G}_t] \leq \frac{2\sqrt{3G}m^{1/4}}{T^{1/2}} + \frac{2\sqrt{3G}}{T^{1/4}} + \frac{\sqrt{2}}{T}.
\tag{65}
$$

$\square$

**Theorem 6.** *(Restatement of Theorem 3) Under the above Assumptions (1, 2, 4, 6, 7), in the Algorithm 1, given $\mathcal{X} = \mathbb{R}^d$, $\eta_t = \frac{k}{(m+t)^{1/2}}$ for all $t \geq 0$, $\alpha_{t+1} = c_1 \eta_t$, $\beta_{t+1} = c_2 \eta_t$, $m \geq \max\big(k^2, (c_1 k)^2, (c_2 k)^2\big)$, $k > 0$, $\frac{125 L_0^2}{6\mu^2} \leq c_1 \leq \frac{m^{1/2}}{k}$, $\frac{9}{2} \leq c_2 \leq \frac{m^{1/2}}{k}$, $0 < \lambda \leq \min\big(\frac{15 b_l L_0^2}{4 L_1^2 \mu}, \frac{b_l}{6 L_g}\big)$, $0 < \gamma \leq \min\big(\frac{\sqrt{6}\lambda\mu\rho}{\sqrt{6 L_1^2 \lambda^2 \mu^2 + 125 b_u^2 L_0^2 \kappa^2}}, \frac{m^{1/2}\rho}{4 L k}\big)$ and $K = \frac{L_g}{\mu} \log(C_{gxy} C_{fy} T/\mu)$, we have*

$$
\frac{1}{T} \sum_{t=1}^T \mathbb{E}\|\nabla F(x_t)\| \leq \frac{\sqrt{\frac{1}{T} \sum_{t=1}^T \mathbb{E}\|A_t\|^2}}{\rho} \Big(\frac{2\sqrt{6G'm}}{T^{1/2}} + \frac{2\sqrt{6G'}}{T^{1/4}} + \frac{2\sqrt{3}}{T}\Big),
\tag{66}
$$

*where $G' = \frac{\rho(F(x_1) - F^*)}{k\gamma} + \frac{5 b_u L_0^2 \Delta_0}{k\lambda\mu} + \frac{2\sigma^2}{k} + \frac{2m\sigma^2}{k} \ln(m+T) + \frac{4(m+T)}{9kT^2} + \frac{8k}{T}$.*

*Proof.* According to Lemma 7, we have

$$
\begin{aligned}
\mathcal{G}_t &= \frac{1}{\gamma}\|x_t - \tilde{x}_{t+1}\| + \frac{1}{\rho}\big(L_0\|y^*(x_t) - y_t\| + \sqrt{2}\|\bar{\nabla} f(x_t, y_t) - w_t\|\big) \\
&\geq \frac{1}{\gamma}\|x_t - \tilde{x}_{t+1}\| + \frac{1}{\rho}\|\nabla F(x_t) - w_t\| \\
&\overset{(i)}{=} \|A_t^{-1} w_t\| + \frac{1}{\rho}\|\nabla F(x_t) - w_t\| \\
&= \frac{1}{\|A_t\|}\|A_t\|\|A_t^{-1} w_t\| + \frac{1}{\rho}\|\nabla F(x_t) - w_t\| \\
&\geq \frac{1}{\|A_t\|}\|w_t\| + \frac{1}{\rho}\|\nabla F(x_t) - w_t\| \\
&\overset{(ii)}{\geq} \frac{1}{\|A_t\|}\|w_t\| + \frac{1}{\|A_t\|}\|\nabla F(x_t) - w_t\| \\
&\geq \frac{1}{\|A_t\|}\|\nabla F(x_t)\|,
\end{aligned}
\tag{67}
$$

where the equality $(i)$ holds by $\tilde{x}_{t+1} = x_t - \gamma A_t^{-1} w_t$ that can be easily obtained from the step 5 of Algorithm 1 when $\mathcal{X} = \mathbb{R}^d$, and the inequality $(ii)$ holds by $\|A_t\| \geq \rho$ for all $t \geq 1$ due to Assumption 7. Then we have

$$
\|\nabla F(x_t)\| \leq \|A_t\|\mathcal{G}_t.
\tag{68}
$$

According to Cauchy-Schwarz inequality, we have

$$\frac{1}{T}\sum_{t=1}^{T}\mathbb{E}\|\nabla F(x_t)\| \leq \frac{1}{T}\sum_{t=1}^{T}\mathbb{E}[\mathcal{G}_t\|A_t\|] \leq \sqrt{\frac{1}{T}\sum_{t=1}^{T}\mathbb{E}[\mathcal{G}_t^2]}\sqrt{\frac{1}{T}\sum_{t=1}^{T}\mathbb{E}\|A_t\|^2}. \tag{69}$$

Then we have

$$\begin{aligned}
\frac{1}{T}\sum_{t=1}^{T}\mathbb{E}[\mathcal{G}_t^2] &\leq \frac{1}{T}\sum_{t=1}^{T}\mathbb{E}\Big[\frac{3L_0^2\|y_t - y^*(x_t)\|^2}{\rho^2} + \frac{6\|w_t - \bar{\nabla}f(x_t, y_t)\|^2}{\rho^2} + \frac{3}{\gamma^2}\|\tilde{x}_{t+1} - x_t\|^2\Big] \\
&\leq \frac{1}{T}\sum_{t=1}^{T}\mathbb{E}\Big[\frac{3L_0^2\|y_t - y^*(x_t)\|^2}{\rho^2} + \frac{12\|w_t - \bar{\nabla}f(x_t, y_t) - R_t\|^2}{\rho^2} + \frac{12\|R_t\|^2}{\rho^2} + \frac{3}{\gamma^2}\|\tilde{x}_{t+1} - x_t\|^2\Big] \\
&\leq \frac{24G}{T}(m + T)^{1/2} + \frac{1}{T}\sum_{t=1}^{T}\frac{12\|R_t\|^2}{\rho^2} \\
&\leq \frac{24G}{T}(m + T)^{1/2} + \frac{12}{\rho^2 T^2}, 
\end{aligned} \tag{70}$$

where the third inequality holds by the above inequality (61), and the last inequality holds by $\|R_t\| \leq \frac{1}{T}$ for all $t \geq 1$ by choosing $K = \frac{L_g}{\mu}\log(C_{gxy}C_{fy}T/\mu)$.

By combining the inequalities (69) and (70), we have

$$\begin{aligned}
\frac{1}{T}\sum_{t=1}^{T}\mathbb{E}\|\nabla F(x_t)\| &\leq \sqrt{\frac{1}{T}\sum_{t=1}^{T}\mathbb{E}[\mathcal{G}_t^2]}\sqrt{\frac{1}{T}\sum_{t=1}^{T}\mathbb{E}\|A_t\|^2} \\
&\leq \frac{\sqrt{\frac{1}{T}\sum_{t=1}^{T}\mathbb{E}\|A_t\|^2}}{\rho}\Big(\frac{2\sqrt{6G'm}}{T^{1/2}} + \frac{2\sqrt{6G'}}{T^{1/4}} + \frac{2\sqrt{3}}{T}\Big),
\end{aligned} \tag{71}$$

where $G' = \rho^2 G$.

$\square$

## A.6 CONVERGENCE ANALYSIS OF VR-BIADAM ALGORITHM

In the subsection, we detail convergence analysis of VR-BiAdam algorithm.

**Lemma 12.** *Under the above Assumptions (1, 3, 4), assume the stochastic gradient estimators $v_t$ and $w_t$ be generated from Algorithm 2, we have*

$$\begin{aligned}
\mathbb{E}\|\nabla_y g(x_{t+1}, y_{t+1}) - v_{t+1}\|^2 &\leq (1 - \alpha_{t+1})\mathbb{E}\|\nabla_y g(x_t, y_t) - v_t\|^2 + 2\alpha_{t+1}^2\sigma^2 \\
&\quad + 4L_g^2\eta_t^2\big(\mathbb{E}\|\tilde{x}_{t+1} - x_t\|^2 + \mathbb{E}\|\tilde{y}_{t+1} - y_t\|^2\big),
\end{aligned} \tag{72}$$

$$\begin{aligned}
\mathbb{E}\|w_{t+1} - \bar{\nabla}f(x_{t+1}, y_{t+1}) - R_{t+1}\|^2 &\leq (1 - \beta_{t+1})\mathbb{E}\|w_t - \bar{\nabla}f(x_t, y_t) - R_t\|^2 + 2\beta_{t+1}^2\sigma^2 \\
&\quad + 4L_K^2\eta_t^2\big(\|\tilde{x}_{t+1} - x_t\|^2 + \|\tilde{y}_{t+1} - y_t\|^2\big),
\end{aligned} \tag{73}$$

*where $L_K^2 = 2L_f^2 + 6C_{gxy}^2 L_f^2\frac{K}{2\mu L_g - \mu^2} + 6C_{fy}^2 L_{gxy}^2\frac{K}{2\mu L_g - \mu^2} + 6C_{gxy}^2 L_f^2\frac{K^3 L_{gyy}^2}{(L_g - \mu)^2(2\mu L_g - \mu^2)}$.*

*Proof.* Without loss of generality, we only prove the term $\mathbb{E}\|w_{t+1} - \bar{\nabla}f(x_{t+1}, y_{t+1}) - R_{t+1}\|^2$. The other term is similar for this term. Since $w_{t+1} = \bar{\nabla}f(x_{t+1}, y_{t+1}; \bar{\xi}_{t+1}) + (1 - \beta_{t+1})\big(w_t -$

$\bar{\nabla}f(x_t, y_t; \bar{\xi}_{t+1}))$, we have

$$\mathbb{E}\|w_{t+1} - \bar{\nabla}f(x_{t+1}, y_{t+1}) - R_{t+1}\|^2 \tag{74}$$
$$= \mathbb{E}\|\bar{\nabla}f(x_{t+1}, y_{t+1}; \bar{\xi}_{t+1}) + (1 - \beta_{t+1})(w_t - \bar{\nabla}f(x_t, y_t; \bar{\xi}_{t+1})) - \bar{\nabla}f(x_{t+1}, y_{t+1}) - R_{t+1}\|^2$$
$$= \mathbb{E}\|(1 - \beta_{t+1})(w_t - \bar{\nabla}f(x_t, y_t) - R_t) + \beta_{t+1}(\bar{\nabla}f(x_{t+1}, y_{t+1}; \bar{\xi}_{t+1}) - \bar{\nabla}f(x_{t+1}, y_{t+1}) - R_{t+1})$$
$$\quad + (1 - \beta_{t+1})(\bar{\nabla}f(x_{t+1}, y_{t+1}; \bar{\xi}_{t+1}) - \bar{\nabla}f(x_{t+1}, y_{t+1}) - R_{t+1} - (\bar{\nabla}f(x_t, y_t; \bar{\xi}_t)) - \bar{\nabla}f(x_t, y_t) - R_t))\|^2$$
$$= (1 - \beta_{t+1})^2\mathbb{E}\|w_t - \bar{\nabla}f(x_t, y_t) - R_t\|^2 + \mathbb{E}\|\beta_{t+1}(\bar{\nabla}f(x_{t+1}, y_{t+1}; \bar{\xi}_{t+1}) - \bar{\nabla}f(x_{t+1}, y_{t+1}) - R_{t+1})$$
$$\quad + (1 - \beta_{t+1})(\bar{\nabla}f(x_{t+1}, y_{t+1}; \bar{\xi}_{t+1}) - \bar{\nabla}f(x_{t+1}, y_{t+1}) - R_{t+1} - (\bar{\nabla}f(x_t, y_t; \bar{\xi}_t) - \bar{\nabla}f(x_t, y_t) - R_t))\|^2$$
$$\leq (1 - \beta_{t+1})^2\mathbb{E}\|w_t - \bar{\nabla}f(x_t, y_t) - R_t\|^2 + 2\beta_{t+1}^2\mathbb{E}\|\bar{\nabla}f(x_{t+1}, y_{t+1}; \bar{\xi}_{t+1}) - \bar{\nabla}f(x_{t+1}, y_{t+1}) - R_{t+1}\|^2$$
$$\quad + 2(1 - \beta_{t+1})^2\|\bar{\nabla}f(x_{t+1}, y_{t+1}; \bar{\xi}_{t+1}) - \bar{\nabla}f(x_{t+1}, y_{t+1}) - R_{t+1} - (\bar{\nabla}f(x_t, y_t; \bar{\xi}_t)) - \bar{\nabla}f(x_t, y_t) - R_t)\|^2$$
$$\leq (1 - \beta_{t+1})^2\mathbb{E}\|w_t - \bar{\nabla}f(x_t, y_t) - R_t\|^2 + 2\beta_{t+1}^2\sigma^2 + 2(1 - \beta_{t+1})^2\|\bar{\nabla}f(x_{t+1}, y_{t+1}; \bar{\xi}_{t+1})) - \bar{\nabla}f(x_t, y_t; \bar{\xi}_t)\|^2$$
$$\leq (1 - \beta_{t+1})^2\mathbb{E}\|w_t - \bar{\nabla}f(x_t, y_t) - R_t\|^2 + 2\beta_{t+1}^2\sigma^2 + 4(1 - \beta_{t+1})^2 L_K^2(\|x_{t+1} - x_t\|^2 + \|y_{t+1} - y_t\|^2)$$
$$\leq (1 - \beta_{t+1})\mathbb{E}\|w_t - \bar{\nabla}f(x_t, y_t) - R_t\|^2 + 2\beta_{t+1}^2\sigma^2 + 4L_K^2\eta_t^2(\|\tilde{x}_{t+1} - x_t\|^2 + \|\tilde{y}_{t+1} - y_t\|^2),$$

where the third equality holds by $\mathbb{E}_{\bar{\xi}}[\bar{\nabla}f(x_{t+1}, y_{t+1}; \bar{\xi}_{t+1})] = \bar{\nabla}f(x_{t+1}, y_{t+1}) + R_{t+1}$ and $\mathbb{E}_{\bar{\xi}}[\bar{\nabla}f(x_t, y_t; \bar{\xi}_t))] = \bar{\nabla}f(x_t, y_t) + R_t$; the third last inequality holds by the inequality $\mathbb{E}\|\zeta - \mathbb{E}[\zeta]\|^2 \leq \mathbb{E}\|\zeta\|^2$; the second last inequality is due to Lemma 4; the last inequality holds by $0 < \beta_{t+1} \leq 1$ and $x_{t+1} = x_t + \eta_t(\tilde{x}_{t+1} - x_t), y_{t+1} = y_t + \eta_t(\tilde{y}_{t+1} - y_t)$.

$\square$

**Theorem 7.** *(Restatement of Theorem 2) Under the above Assumptions (1, 3, 4, 6, 7), in the Algorithm 2, given $\mathcal{X} \subset \mathbb{R}^d$, $\eta_t = \frac{k}{(m+t)^{1/3}}$ for all $t \geq 0$, $\alpha_{t+1} = c_1\eta_t^2$, $\beta_{t+1} = c_2\eta_t^2$, $m \geq \max(2, k^3, (c_1k)^3, (c_2k)^3)$, $k > 0$, $c_1 \geq \frac{2}{3k^3} + \frac{125L_0^2}{6\mu^2}$, $c_2 \geq \frac{2}{3k^3} + \frac{9}{2}$, $0 < \lambda \leq \min(\frac{15b_lL_0^2}{16L_2^2\mu}, \frac{b_l}{6L_g})$, $0 < \gamma \leq \min(\frac{\sqrt{6}\lambda\mu\rho}{2\sqrt{24L_2^2\lambda^2\mu^2 + 125b_u^2L_0^2\kappa^2}}, \frac{m^{1/3}\rho}{4Lk})$ and $K = \frac{L_g}{\mu}\log(C_{gxy}C_{fy}T/\mu)$, we have*

$$\frac{1}{T}\sum_{t=1}^T \mathbb{E}\|\mathcal{G}_\mathcal{X}(x_t, \nabla F(x_t), \gamma)\| \leq \frac{1}{T}\sum_{t=1}^T \mathbb{E}[\mathcal{G}_t] \leq \frac{2\sqrt{3M}m^{1/6}}{T^{1/2}} + \frac{2\sqrt{3M}}{T^{1/3}} + \frac{\sqrt{2}}{T}, \tag{75}$$

*where $M = \frac{F(x_1) - F^*}{\rho k\gamma} + \frac{5b_lL_0^2\Delta_0}{\rho^2 k\lambda\mu} + \frac{2m^{1/3}\sigma^2}{\rho^2 k^2} + \frac{2k^2(c_1^2 + c_2^2)\sigma^2\ln(m+T)}{\rho^2} + \frac{6k(m+T)^{1/3}}{\rho^2 T}$, $\Delta_0 = \|y_1 - y^*(x_1)\|^2$ and $L_2^2 = L_g^2 + L_K^2$.*

*Proof.* Since $\eta_t = \frac{k}{(m+t)^{1/3}}$ on $t$ is decreasing and $m \geq k^3$, we have $\eta_t \leq \eta_0 = \frac{k}{m^{1/3}} \leq 1$ and $\gamma \leq \frac{m^{1/3}\rho}{4Lk} \leq \frac{\rho}{2L\eta_0} \leq \frac{\rho}{2L\eta_t}$ for any $t \geq 0$. Due to $0 < \eta_t \leq 1$ and $m \geq (c_1k)^3$, we have $\alpha_{t+1} = c_1\eta_t^2 \leq c_1\eta_t \leq \frac{c_1k}{m^{1/3}} \leq 1$. Similarly, due to $m \geq (c_2k)^3$, we have $\beta_{t+1} \leq 1$. According to Lemma 12, we have

$$\frac{1}{\eta_t}\mathbb{E}\|\nabla_y g(x_{t+1}, y_{t+1}) - v_{t+1}\|^2 - \frac{1}{\eta_{t-1}}\mathbb{E}\|\nabla_y g(x_t, y_t) - v_t\|^2 \tag{76}$$
$$\leq (\frac{1 - \alpha_{t+1}}{\eta_t} - \frac{1}{\eta_{t-1}})\mathbb{E}\|\nabla_y g(x_t, y_t) - v_t\|^2 + 4L_g^2\eta_t(\|\tilde{x}_{t+1} - x_t\|^2 + \|\tilde{y}_{t+1} - y_t\|^2) + \frac{2\alpha_{t+1}^2\sigma^2}{\eta_t}$$
$$= (\frac{1}{\eta_t} - \frac{1}{\eta_{t-1}} - c_1\eta_t)\mathbb{E}\|\nabla_y g(x_t, y_t) - v_t\|^2 + 4L_g^2\eta_t(\|\tilde{x}_{t+1} - x_t\|^2 + \|\tilde{y}_{t+1} - y_t\|^2) + 2c_1^2\eta_t^3\sigma^2,$$

where the second equality is due to $\alpha_{t+1} = c_1\eta_t^2$. Similarly, we have

$$\frac{1}{\eta_t}\mathbb{E}\|w_{t+1} - \bar{\nabla}f(x_{t+1}, y_{t+1}) - R_{t+1}\|^2 - \frac{1}{\eta_{t-1}}\mathbb{E}\|w_t - \bar{\nabla}f(x_t, y_t) - R_t\|^2 \tag{77}$$
$$\leq (\frac{1 - \beta_{t+1}}{\eta_t} - \frac{1}{\eta_{t-1}})\mathbb{E}\|w_t - \bar{\nabla}f(x_t, y_t) - R_t\|^2 + 4L_K^2\eta_t(\|\tilde{x}_{t+1} - x_t\|^2 + \|\tilde{y}_{t+1} - y_t\|^2) + \frac{2\beta_{t+1}^2\sigma^2}{\eta_t}$$
$$= (\frac{1}{\eta_t} - \frac{1}{\eta_{t-1}} - c_2\eta_t)\mathbb{E}\|w_t - \bar{\nabla}f(x_t, y_t) - R_t\|^2 + 4L_K^2\eta_t(\|\tilde{x}_{t+1} - x_t\|^2 + \|\tilde{y}_{t+1} - y_t\|^2) + 2c_2^2\eta_t^3\sigma^2.$$

By $\eta_t = \frac{k}{(m+t)^{1/3}}$, we have

$$
\frac{1}{\eta_t} - \frac{1}{\eta_{t-1}} = \frac{1}{k}\left((m+t)^{\frac{1}{3}} - (m+t-1)^{\frac{1}{3}}\right) \leq \frac{1}{3k(m+t-1)^{2/3}} \leq \frac{1}{3k(m/2+t)^{2/3}}
$$

$$
\leq \frac{2^{2/3}}{3k(m+t)^{2/3}} = \frac{2^{2/3}}{3k^3}\frac{k^2}{(m+t)^{2/3}} = \frac{2^{2/3}}{3k^3}\eta_t^2 \leq \frac{2}{3k^3}\eta_t, \tag{78}
$$

where the first inequality holds by the concavity of function $f(x) = x^{1/3}$, *i.e.*, $(x+y)^{1/3} \leq x^{1/3} + \frac{y}{3x^{2/3}}$; the second inequality is due to $m \geq 2$, and the last inequality is due to $0 < \eta_t \leq 1$.

Let $c_1 \geq \frac{2}{3k^3} + \frac{125L_0^2}{6\mu^2}$, we have

$$
\frac{1}{\eta_t}\mathbb{E}\|\nabla_y g(x_{t+1}, y_{t+1}) - v_{t+1}\|^2 - \frac{1}{\eta_{t-1}}\mathbb{E}\|\nabla_y g(x_t, y_t) - v_t\|^2 \tag{79}
$$

$$
\leq -\frac{125L_0^2\eta_t}{6\mu^2}\mathbb{E}\|\nabla_y g(x_t, y_t) - v_t\|^2 + 4L_g^2\eta_t\left(\|\tilde{x}_{t+1} - x_t\|^2 + \|\tilde{y}_{t+1} - y_t\|^2\right) + 2c_1^2\eta_t^3\sigma^2.
$$

Let $c_2 \geq \frac{2}{3k^3} + \frac{9}{2}$, we have

$$
\frac{1}{\eta_t}\mathbb{E}\|w_{t+1} - \bar{\nabla} f(x_{t+1}, y_{t+1}) - R_{t+1}\|^2 - \frac{1}{\eta_{t-1}}\mathbb{E}\|w_t - \bar{\nabla} f(x_t, y_t) - R_t\|^2 \tag{80}
$$

$$
\leq -\frac{9\eta_t}{2}\mathbb{E}\|w_t - \bar{\nabla} f(x_t, y_t) - R_t\|^2 + 4L_K^2\eta_t\left(\|\tilde{x}_{t+1} - x_t\|^2 + \|\tilde{y}_{t+1} - y_t\|^2\right) + 2c_2^2\eta_t^3\sigma^2.
$$

According to Lemmas 7 and 9, we have

$$
F(x_{t+1}) - F(x_t) \tag{81}
$$

$$
\leq \frac{2\eta_t\gamma}{\rho}\|w_t - \bar{\nabla} f(x_t, y_t)\|^2 + \frac{L_0^2\eta_t\gamma}{\rho}\|y^*(x_t) - y_t\|^2 - \frac{\rho\eta_t}{2\gamma}\|\tilde{x}_{t+1} - x_t\|^2
$$

$$
\leq \frac{4\eta_t\gamma}{\rho}\|w_t - \bar{\nabla} f(x_t, y_t) - R_t\|^2 + \frac{4\eta_t\gamma}{\rho}\|R_t\|^2 + \frac{L_0^2\eta_t\gamma}{\rho}\|y^*(x_t) - y_t\|^2 - \frac{\rho\eta_t}{2\gamma}\|\tilde{x}_{t+1} - x_t\|^2.
$$

According to Lemma 10, we have

$$
\|y_{t+1} - y^*(x_{t+1})\|^2 - \|y_t - y^*(x_t)\|^2 \tag{82}
$$

$$
\leq -\frac{\eta_t\mu\lambda}{4b_t}\|y_t - y^*(x_t)\|^2 - \frac{3\eta_t}{4}\|\tilde{y}_{t+1} - y_t\|^2 + \frac{25\eta_t\lambda}{6\mu b_t}\|\nabla_y g(x_t, y_t) - v_t\|^2 + \frac{25\kappa^2\eta_t b_t}{6\mu\lambda}\|\tilde{x}_{t+1} - x_t\|^2.
$$

Next, we define a *Lyapunov* function, for any $t \geq 1$

$$
\Theta_t = \mathbb{E}\Big[F(x_t) + \frac{5b_t L_0^2\gamma}{\lambda\mu\rho}\|y_t - y^*(x_t)\|^2 + \frac{\gamma}{\rho\eta_{t-1}}\big(\|v_t - \nabla_y g(x_t, y_t)\|^2 + \|w_t - \bar{\nabla} f(x_t, y_t) - R_t\|^2\big)\Big].
$$

For notational simplicity, let $L_2^2 = L_g^2 + L_K^2$. Then we have

$$\Theta_{t+1} - \Theta_t$$

$$= F(x_{t+1}) - F(x_t) + \frac{5b_t L_0^2 \gamma}{\lambda \mu \rho}\left(\|y_{t+1} - y^*(x_{t+1})\|^2 - \|y_t - y^*(x_t)\|^2\right) + \frac{\gamma}{\rho}\left(\frac{1}{\eta_t}\mathbb{E}\|v_{t+1} - \nabla_y g(x_{t+1}, y_{t+1})\|^2\right.$$

$$- \frac{1}{\eta_{t-1}}\mathbb{E}\|v_t - \nabla_y g(x_t, y_t)\|^2 + \frac{1}{\eta_t}\mathbb{E}\|w_{t+1} - \bar{\nabla}f(x_{t+1}, y_{t+1}) - R_{t+1}\|^2 - \frac{1}{\eta_{t-1}}\mathbb{E}\|w_t - \bar{\nabla}f(x_t, y_t) - R_t\|^2\bigg)$$

$$\leq \frac{L_0^2 \eta_t \gamma}{\rho}\|y^*(x_t) - y_t\|^2 + \frac{4\eta_t \gamma}{\rho}\|w_t - \bar{\nabla}f(x_t, y_t) - R_t\|^2 + \frac{4\eta_t \gamma}{\rho}\|R_t\|^2 - \frac{\rho \eta_t}{2\gamma}\|\tilde{x}_{t+1} - x_t\|^2$$

$$+ \frac{5b_t L_0^2 \gamma}{\lambda \mu \rho}\left(-\frac{\eta_t \mu \lambda}{4b_t}\|y_t - y^*(x_t)\|^2 - \frac{3\eta_t}{4}\|\tilde{y}_{t+1} - y_t\|^2 + \frac{25\eta_t \lambda}{6\mu b_t}\|\nabla_y g(x_t, y_t) - v_t\|^2 + \frac{25\kappa^2 \eta_t b_t}{6\mu \lambda}\|\tilde{x}_{t+1} - x_t\|^2\right)$$

$$+ \frac{\gamma}{\rho}\left(-\frac{125 L_0^2 \eta_t}{6\mu^2}\mathbb{E}\|\nabla_y g(x_t, y_t) - v_t\|^2 + 4L_g^2 \eta_t\left(\|\tilde{x}_{t+1} - x_t\|^2 + \|\tilde{y}_{t+1} - y_t\|^2\right) + 2c_1^2 \eta_t^3 \sigma^2\right.$$

$$\left. - \frac{9\eta_t}{2}\mathbb{E}\|w_t - \bar{\nabla}f(x_t, y_t) - R_t\|^2 + 4L_K^2 \eta_t\left(\|\tilde{x}_{t+1} - x_t\|^2 + \|\tilde{y}_{t+1} - y_t\|^2\right) + 2c_2^2 \eta_t^3 \sigma^2\right)$$

$$= -\frac{\gamma \eta_t}{4\rho}\left(L_0^2\|y_t - y^*(x_t)\|^2 + 2\mathbb{E}\|w_t - \bar{\nabla}f(x_t, y_t) - R_t\|^2\right) - \left(\frac{\rho}{2\gamma} - \frac{4L_2^2 \gamma}{\rho} - \frac{125 b_t^2 L_0^2 \kappa^2 \gamma}{6\mu^2 \lambda^2 \rho}\right)\eta_t\|\tilde{x}_{t+1} - x_t\|^2$$

$$- \left(\frac{15 b_t L_0^2 \gamma}{4\lambda \mu \rho} - \frac{4L_2^2 \gamma}{\rho}\right)\eta_t\|\tilde{y}_{t+1} - y_t\|^2 + \frac{4\eta_t \gamma}{\rho}\|R_t\|^2 + \frac{2(c_1^2 + c_2^2)\gamma \sigma^2}{\rho}\eta_t^3$$

$$\leq -\frac{\gamma \eta_t}{4\rho}\left(L_0^2\|y_t - y^*(x_t)\|^2 + 2\mathbb{E}\|w_t - \bar{\nabla}f(x_t, y_t) - R_t\|^2\right) - \frac{\rho \eta_t}{4\gamma}\|\tilde{x}_{t+1} - x_t\|^2$$

$$+ \frac{4\eta_t \gamma}{\rho}\|R_t\|^2 + \frac{2(c_1^2 + c_2^2)\gamma \sigma^2}{\rho}\eta_t^3, \tag{83}$$

where the first inequality holds by the above inequalities (79), (80), (81) and (82); the last inequality is due to $0 < \gamma \leq \frac{\sqrt{6}\lambda \mu \rho}{2\sqrt{24 L_2^2 \lambda^2 \mu^2 + 125 b_u^2 L_0^2 \kappa^2}} \leq \frac{\sqrt{6}\lambda \mu \rho}{2\sqrt{24 L_2^2 \lambda^2 \mu^2 + 125 b_t^2 L_0^2 \kappa^2}}$, $0 < \lambda \leq \frac{15 b_t L_0^2}{16 L_2^2 \mu} \leq \frac{15 b_t L_0^2}{16 L_2^2 \mu}$ for all $t \geq 1$.

Let $\Phi_t = L_0^2\|y_t - y^*(x_t)\|^2 + 2\|w_t - \bar{\nabla}f(x_t, y_t) - R_t\|^2$, then we have

$$\frac{\gamma \eta_t}{4\rho}\mathbb{E}\left[\Phi_t + \frac{\rho \eta_t}{4\gamma}\|\tilde{x}_{t+1} - x_t\|^2\right] \leq \Theta_t - \Theta_{t+1} + \frac{4\eta_t \gamma}{\rho}\|R_t\|^2 + \frac{2(c_1^2 + c_2^2)\gamma \sigma^2}{\rho}\eta_t^3. \tag{84}$$

Taking average over $t = 1, 2, \cdots, T$ on both sides of (84), we have

$$\frac{1}{T}\sum_{t=1}^{T}\mathbb{E}\left[\frac{\eta_t}{4}\Phi_t + \frac{\rho^2 \eta_t}{4\gamma^2}\|\tilde{x}_{t+1} - x_t\|^2\right] \leq \sum_{t=1}^{T}\frac{\rho(\Theta_t - \Theta_{t+1})}{T\gamma} + \frac{4}{T}\sum_{t=1}^{T}\eta_t\|R_t\|^2 + \frac{2(c_1^2 + c_2^2)\sigma^2}{T}\sum_{t=1}^{T}\eta_t^3.$$

Given $x_1 \in \mathcal{X}$ and $y_1 \in \mathcal{Y}$, let $\Delta_0 = \|y_1 - y^*(x_1)\|^2$, we have

$$\Theta_1 = \mathbb{E}\left[F(x_1) + \frac{5b_1 L_0^2 \gamma}{\lambda \mu \rho}\|y_1 - y^*(x_1)\|^2 + \frac{\gamma}{\rho \eta_0}\left(\|v_1 - \nabla_y g(x_1, y_1)\|^2 + \|w_1 - \bar{\nabla}f(x_1, y_1) - R_1\|^2\right)\right]$$

$$\leq F(x_1) + \frac{5b_1 L_0^2 \gamma \Delta_0}{\lambda \mu \rho} + \frac{2\gamma \sigma^2}{\rho \eta_0}, \tag{85}$$

where the last inequality holds by Assumption 2. Since $\eta_t$ is decreasing, i.e., $\eta_T^{-1} \geq \eta_t^{-1}$ for any $0 \leq t \leq T$, we have

$$\frac{1}{T} \sum_{t=1}^{T} \mathbb{E}\Big[\frac{\Phi_t}{4} + \frac{\rho^2}{4\gamma^2}\|\tilde{x}_{t+1} - x_t\|^2\Big] \tag{86}$$

$$\leq \frac{\rho}{T\gamma\eta_T} \sum_{t=1}^{T} \big(\Theta_t - \Theta_{t+1}\big) + \frac{2(c_1^2 + c_2^2)\sigma^2}{T\eta_T} \sum_{t=1}^{T} \eta_t^3 + \frac{4}{T} \sum_{t=1}^{T} \eta_t \|R_t\|^2$$

$$\leq \frac{\rho}{T\gamma\eta_T} \Big(F(x_1) + \frac{5b_1 L_0^2 \gamma \Delta_0}{\lambda\mu\rho} + \frac{2\gamma\sigma^2}{\rho\eta_0} - F^*\Big) + \frac{2(c_1^2 + c_2^2)\sigma^2}{T\eta_T} \sum_{t=1}^{T} \eta_t^3 + \frac{4}{T^2} \sum_{t=1}^{T} \eta_t$$

$$\leq \frac{\rho(F(x_1) - F^*)}{T\gamma\eta_T} + \frac{5b_1 L_0^2 \Delta_0}{\lambda\mu\eta_T T} + \frac{2\sigma^2}{\eta_0\eta_T T} + \frac{2(c_1^2 + c_2^2)\sigma^2}{T\eta_T} \int_1^T \frac{k^3}{m+t}dt + \frac{4}{T^2} \int_1^T \frac{k}{(m+t)^{1/3}}dt$$

$$\leq \frac{\rho(F(x_1) - F^*)}{T\gamma\eta_T} + \frac{5b_1 L_0^2 \Delta_0}{\lambda\mu\eta_T T} + \frac{2\sigma^2}{\eta_0\eta_T T} + \frac{2k^3(c_1^2 + c_2^2)\sigma^2}{T\eta_T} \ln(m+T) + \frac{6k}{T^2}(m+T)^{2/3}$$

$$= \Big(\frac{\rho(F(x_1) - F^*)}{k\gamma} + \frac{5b_1 L_0^2 \Delta_0}{k\lambda\mu} + \frac{2m^{1/3}\sigma^2}{k^2} + 2k^2(c_1^2 + c_2^2)\sigma^2 \ln(m+T) + \frac{6k(m+T)^{1/3}}{T}\Big)\frac{(m+T)^{1/3}}{T},$$

where the second inequality holds by the above inequality (85). Let $M = \frac{F(x_1) - F^*}{\rho k\gamma} + \frac{5b_1 L_0^2 \Delta_0}{\rho^2 k\lambda\mu} + \frac{2m^{1/3}\sigma^2}{\rho^2 k^2} + \frac{2k^2(c_1^2 + c_2^2)\sigma^2 \ln(m+T)}{\rho^2} + \frac{6k(m+T)^{1/3}}{\rho^2 T}$, we have

$$\frac{1}{T} \sum_{t=1}^{T} \mathbb{E}\Big[\frac{\Phi_t}{4\rho^2} + \frac{1}{4\gamma^2}\|\tilde{x}_{t+1} - x_t\|^2\Big] \leq \frac{M}{T}(m+T)^{1/3}. \tag{87}$$

According to Jensen's inequality, we have

$$\frac{1}{T} \sum_{t=1}^{T} \mathbb{E}\Big[\frac{1}{2\gamma}\|\tilde{x}_{t+1} - x_t\| + \frac{1}{2\rho}\big(L_0\|y_t - y^*(x_t)\| + \sqrt{2}\|w_t - \bar{\nabla}f(x_t, y_t) - R_t\|\big)\Big]$$

$$\leq \Big(\frac{3}{T} \sum_{t=1}^{T} \big(\frac{1}{4\gamma^2}\|\tilde{x}_{t+1} - x_t\|^2 + \frac{L_0^2}{4\rho^2}\|y_t - y^*(x_t)\|^2 + \frac{2}{4\rho^2}\mathbb{E}\|w_t - \bar{\nabla}f(x_t, y_t) - R_t\|^2\big)\Big)^{1/2}$$

$$= \Big(\frac{3}{T} \sum_{t=1}^{T} \big(\frac{\Phi_t}{4\rho^2} + \frac{1}{4\gamma^2}\|\tilde{x}_{t+1} - x_t\|^2\big)\Big)^{1/2}$$

$$\leq \frac{\sqrt{3M}}{T^{1/2}}(m+T)^{1/6} \leq \frac{\sqrt{3M}m^{1/6}}{T^{1/2}} + \frac{\sqrt{3M}}{T^{1/3}}, \tag{88}$$

where the last inequality is due to $(a+b)^{1/6} \leq a^{1/6} + b^{1/6}$ for all $a, b > 0$. Thus we have

$$\frac{1}{T} \sum_{t=1}^{T} \mathbb{E}\Big[\frac{1}{\gamma}\|\tilde{x}_{t+1} - x_t\| + \frac{1}{\rho}\big(L_0\|y_t - y^*(x_t)\| + \sqrt{2}\|w_t - \bar{\nabla}f(x_t, y_t) - R_t\|\big)\Big]$$

$$\leq \frac{2\sqrt{3M}m^{1/6}}{T^{1/2}} + \frac{2\sqrt{3M}}{T^{1/3}}. \tag{89}$$

Since $\|w_t - \bar{\nabla}f(x_t, y_t) - R_t\| \geq \|w_t - \bar{\nabla}f(x_t, y_t)\| - \|R_t\|$, by the above inequality (89), we can obtain

$$\frac{1}{T} \sum_{t=1}^{T} \mathbb{E}\Big[\frac{1}{\gamma}\|\tilde{x}_{t+1} - x_t\| + \frac{1}{\rho}\big(L_0\|y_t - y^*(x_t)\| + \sqrt{2}\|w_t - \bar{\nabla}f(x_t, y_t) - R_t\|\big)\Big]$$

$$\leq \frac{2\sqrt{3M}m^{1/6}}{T^{1/2}} + \frac{2\sqrt{3M}}{T^{1/3}} + \frac{\sqrt{2}}{T} \sum_{t=1}^{T} \|R_t\|$$

$$= \frac{2\sqrt{3M}m^{1/6}}{T^{1/2}} + \frac{2\sqrt{3M}}{T^{1/3}} + \frac{\sqrt{2}}{T}, \tag{90}$$

where the last inequality is due to $\|R_t\| \leq \frac{1}{T}$ for all $t \geq 1$ by choosing $K = \frac{L_g}{\mu} \log(C_{gxy} C_{fy} T/\mu)$ in Algorithm 2.

According to the above inequality (11), we have

$$\frac{1}{T} \sum_{t=1}^{T} \mathbb{E}\|\mathcal{G}_\mathcal{X}(x_t, \nabla F(x_t), \gamma)\| \leq \frac{1}{T} \sum_{t=1}^{T} \mathbb{E}[\mathcal{G}_t] \leq \frac{2\sqrt{3M} m^{1/6}}{T^{1/2}} + \frac{2\sqrt{3M}}{T^{1/3}} + \frac{\sqrt{2}}{T}. \tag{91}$$

$\square$

**Theorem 8.** *(Restatement of Theorem 4) Under the above Assumptions (1, 3, 4, 6, 7), in the Algorithm 2, given $\mathcal{X} = \mathbb{R}^d$, $\eta_t = \frac{k}{(m+t)^{1/3}}$ for all $t \geq 0$, $\alpha_{t+1} = c_1 \eta_t^2$, $\beta_{t+1} = c_2 \eta_t^2$, $m \geq \max\left(2, k^3, (c_1 k)^3, (c_2 k)^3\right)$, $k > 0$, $c_1 \geq \frac{2}{3k^3} + \frac{125 L_0^2}{6\mu^2}$, $c_2 \geq \frac{2}{3k^3} + \frac{9}{2}$, $0 < \lambda \leq \min\left(\frac{15 b_l L_0^2}{16 L_2^2 \mu}, \frac{b_l}{6 L_g}\right)$, $0 < \gamma \leq \min\left(\frac{\sqrt{6}\lambda\mu\rho}{2\sqrt{24 L_2^2 \lambda^2 \mu^2 + 125 b_u^2 L_0^2 \kappa^2}}, \frac{m^{1/3}\rho}{4Lk}\right)$ and $K = \frac{L_g}{\mu} \log(C_{gxy} C_{fy} T/\mu)$, we have*

$$\frac{1}{T} \sum_{t=1}^{T} \mathbb{E}\|\nabla F(x_t)\| \leq \frac{\sqrt{\frac{1}{T} \sum_{t=1}^{T} \mathbb{E}\|A_t\|^2}}{\rho} \left(\frac{2\sqrt{6M'm}}{T^{1/2}} + \frac{2\sqrt{6M'}}{T^{1/3}} + \frac{2\sqrt{3}}{T}\right), \tag{92}$$

*where $M' = \frac{\rho(F(x_1) - F^*)}{k\gamma} + \frac{5 b_1 L_0^2 \Delta_0}{k\lambda\mu} + \frac{2m^{1/3}\sigma^2}{k^2} + 2k^2(c_1^2 + c_2^2)\sigma^2 \ln(m+T) + \frac{6k(m+T)^{1/3}}{T}$.*

*Proof.* According to Lemma 7, we have

$$\begin{aligned}
\mathcal{G}_t &= \frac{1}{\gamma}\|x_t - \tilde{x}_{t+1}\| + \frac{1}{\rho}\left(L_0\|y^*(x_t) - y_t\| + \sqrt{2}\|\bar{\nabla}f(x_t, y_t) - w_t\|\right) \\
&\geq \frac{1}{\gamma}\|x_t - \tilde{x}_{t+1}\| + \frac{1}{\rho}\|\nabla F(x_t) - w_t\| \\
&\overset{(i)}{=} \|A_t^{-1} w_t\| + \frac{1}{\rho}\|\nabla F(x_t) - w_t\| \\
&= \frac{1}{\|A_t\|}\|A_t\|\|A_t^{-1} w_t\| + \frac{1}{\rho}\|\nabla F(x_t) - w_t\| \\
&\geq \frac{1}{\|A_t\|}\|w_t\| + \frac{1}{\rho}\|\nabla F(x_t) - w_t\| \\
&\overset{(ii)}{\geq} \frac{1}{\|A_t\|}\|w_t\| + \frac{1}{\|A_t\|}\|\nabla F(x_t) - w_t\| \\
&\geq \frac{1}{\|A_t\|}\|\nabla F(x_t)\|,
\end{aligned} \tag{93}$$

where the equality $(i)$ holds by $\tilde{x}_{t+1} = x_t - \gamma A_t^{-1} w_t$ that can be easily obtained from the step 5 of Algorithm 2 when $\mathcal{X} = \mathbb{R}^d$, and the inequality $(ii)$ holds by $\|A_t\| \geq \rho$ for all $t \geq 1$ due to Assumption 7. Then we have

$$\|\nabla F(x_t)\| \leq \|A_t\|\mathcal{G}_t. \tag{94}$$

According to Cauchy-Schwarz inequality, we have

$$\frac{1}{T} \sum_{t=1}^{T} \mathbb{E}\|\nabla F(x_t)\| \leq \frac{1}{T} \sum_{t=1}^{T} \mathbb{E}[\mathcal{G}_t \|A_t\|] \leq \sqrt{\frac{1}{T} \sum_{t=1}^{T} \mathbb{E}[\mathcal{G}_t^2]} \sqrt{\frac{1}{T} \sum_{t=1}^{T} \mathbb{E}\|A_t\|^2}. \tag{95}$$

Then we have

$$\frac{1}{T}\sum_{t=1}^{T}\mathbb{E}[\mathcal{G}_t^2] \leq \frac{1}{T}\sum_{t=1}^{T}\mathbb{E}\Big[\frac{3L_0^2\|y_t - y^*(x_t)\|^2}{\rho^2} + \frac{6\|w_t - \bar{\nabla}f(x_t, y_t)\|^2}{\rho^2} + \frac{3}{\gamma^2}\|\tilde{x}_{t+1} - x_t\|^2\Big]$$

$$\leq \frac{1}{T}\sum_{t=1}^{T}\mathbb{E}\Big[\frac{3L_0^2\|y_t - y^*(x_t)\|^2}{\rho^2} + \frac{12\|w_t - \bar{\nabla}f(x_t, y_t) - R_t\|^2}{\rho^2} + \frac{12\|R_t\|^2}{\rho^2} + \frac{3}{\gamma^2}\|\tilde{x}_{t+1} - x_t\|^2\Big]$$

$$\leq \frac{24M}{T}(m + T)^{1/3} + \frac{1}{T}\sum_{t=1}^{T}\frac{12\|R_t\|^2}{\rho^2}$$

$$\leq \frac{24M}{T}(m + T)^{1/3} + \frac{12}{\rho^2 T^2}, \tag{96}$$

where the third inequality holds by the above inequality (87), and the last inequality holds by $\|R_t\| \leq \frac{1}{T}$ for all $t \geq 1$ by choosing $K = \frac{L_g}{\mu}\log(C_{gxy}C_{fy}T/\mu)$.

By combining the above inequalities (95) and (96), we have

$$\frac{1}{T}\sum_{t=1}^{T}\mathbb{E}\|\nabla F(x_t)\| \leq \sqrt{\frac{1}{T}\sum_{t=1}^{T}\mathbb{E}[\mathcal{G}_t^2]}\sqrt{\frac{1}{T}\sum_{t=1}^{T}\mathbb{E}\|A_t\|^2}$$

$$\leq \frac{\sqrt{\frac{1}{T}\sum_{t=1}^{T}\mathbb{E}\|A_t\|^2}}{\rho}\Big(\frac{2\sqrt{6M'm}}{T^{1/2}} + \frac{2\sqrt{6M'}}{T^{1/3}} + \frac{2\sqrt{3}}{T}\Big), \tag{97}$$

where $M' = \rho^2 M$.

$\square$

