# OpenReview forum: "BiAdam: Fast Adaptive Bilevel Optimization Methods"
_ICLR.cc/2023/Conference — Submitted to ICLR 2023_

### Official Review · Reviewer_LYRV · 2022-10-24

**Confidence:** 4
**Correctness:** 4
**Technical Novelty And Significance:** 3
**Empirical Novelty And Significance:** 3
**Recommendation:** 8

**Clarity, Quality, Novelty And Reproducibility:**

This paper is well written. The novelty of this paper is that it proposed the adaptive gradient-based methods for nonconvex-strongly-convex bilevel optimization and it also provided solid convergence convergence.

**Strength And Weaknesses:**

**Strength:**

This paper proposed a class of efficient adaptive bilevel optimization methods, which build on the unified adaptive matrices and momentum techniques. Moreover, it provides the solid theoretical analysis for the proposed algorithms, and proved the proposed algorithm reaches the best known sample complexity.

**Weakness:**

The proposed algorithms have many tuning parameters.

**Summary Of The Paper:**

This paper proposed a class of efficient adaptive bilevel optimization methods based on momentum techniques to solve the nonconvex-strongly-convex bilevel optimization problems. Moreover, it studied the convergence properties of the proposed methods, and provided the solid convergence analysis. It also conducted the empirical experiments on data hyper-cleaning and hyper-representation learning tasks to verify the efficiency of the proposed algorithms.

**Summary Of The Review:**

This paper studied the adaptive bilevel methods for the nonconvex-strongly-convex bilelve optimization. One of the main contribution of this paper the the theoretical analysis for the proposed adaptive gradient-based methods for bilelve optimization. Experimentally, the proposed algorithms outperform various algorithms in data hyper-cleaning and hyper-representation learning tasks.

Some Questions:

1. Why use the momentum steps in the lines 5-6 of Algorithms?

2. Why the norm of adaptive matrix has a lower bound in Assumption 7?

3. Why the authors choose the decreasing parameter $\eta_t$ ? Could we choose the
constant parameter $\eta_t$ in the proposed algorithms ?

4. How to choose the tuning parameters in the proposed Algorithms ?

5. The convergence analysis does not show the advantage of the adaptive gradient method since
it does not exploit the specific coordinate-wise adaptive stepsize used by Adagrad? In the best case, the convergence can be as good as the non-adaptive gradient descent ascent.

---

> ### Author Response · Authors · 2022-11-09
> **Responses to Reviewer LYRV**
>
> Thanks so much for your positive comments.
>
> **R1:** Our momentum steps in the lines 5-6 not only accelerate our algorithms, but also let our algorithms easily
> use the unified adaptive matrices to flexibly incorporate various adaptive
> learning rates to update variables in both UL and LL problems.
>
> **R2:** From the formulations (4)-(5) at page 4 of our paper, we can choose positive tuning parameters $\rho>0$ and $\varepsilon>0$.
> In fact, the existing adaptive methods implicitly give the lower bounds of adaptive learning rates (Please see the page 6 of **[d]**
> (https://proceedings.neurips.cc/paper/2021/file/4be5a36cbaca8ab9d2066debfe4e65c1-Paper.pdf).
>
> **R3:** In our convergence analysis, we choose decreasing parameter $\eta_t$. Thus, we can give the tuning parameters
> $\alpha_t$ and $\beta_t$ depended on this parameter $\eta_t$. In this case, our convergence analysis framework is
> relatively simple. In fact, we can also choose the constant parameter
> $\eta_t$ in the experiments.
>
> **R4:** In our algorithms, we choose the parameters $\eta_t$, $\alpha_t$ and $\beta_t$ based on the given forms in the convergence analysis. Specifically, in our BiAdam algorithm, we use $\eta_t = \frac{k}{(m+t)^{1/2}}$,
> $\alpha_{t+1}=c_1\eta_t$ and $\beta_{t+1}=c_2\eta_t$. In our VR-BiAdam algorithm, we use $\eta_t = \frac{k}{(m+t)^{1/3}}$,
> $\alpha_{t+1}=c_1\eta^2_t$ and $\beta_{t+1}=c_2\eta^2_t$. To further choose the tuning parameters in these parameters,
> we use the grid search to find the optimal tuning parameters $k,m,c_1,c_2$. **NOTE THAT**: in the experiments, good performances of our algorithms are robust to these tuning parameters $k,m,c_1,c_2$.
>
> **R5:** It is well known that most of adaptive methods enjoy the same complexity as the non-adaptive versions but converges faster in practice.  Only some specific adaptive methods show a faster convergence rate (lower complexity) than the non-adaptive versions under some specific conditions such as the sparse gradient condition used in Adagrad. We propose an adaptive gradient-based algorithm framework for bilevel optimization based on the general adaptive matrices. Meanwhile, we provide a convergence analysis framework for our algorithms. In our convergence analysis, thus, we do not assume some specific adaptive learning rates and some specific conditions such as sparse gradients.
>
> **[d]** Feihu Huang, Junyi Li, and Heng Huang. Super-adam: faster and universal framework of adaptive
> gradients. Advances in Neural Information Processing Systems, 34:9074–9085, 2021.
>  (https://proceedings.neurips.cc/paper/2021/file/4be5a36cbaca8ab9d2066debfe4e65c1-Paper.pdf)

---

> > ### Comment · Reviewer_LYRV · 2022-11-13
> > **Thanks for your reply**
> >
> > Thanks a lot for your clarification. The response well solved my concerns, so I would like to keep my score to support this paper.
> >
> >
> > Best,
> >
> > Reviewer LYRV

---

> > > ### Author Response · Authors · 2022-11-16
> > > **Responses to Reviewer LYRV**
> > >
> > > Thanks so much for your reply and positive comment.

---

### Official Review · Reviewer_5EBw · 2022-10-24

**Confidence:** 4
**Correctness:** 4
**Technical Novelty And Significance:** 3
**Empirical Novelty And Significance:** 3
**Recommendation:** 8

**Clarity, Quality, Novelty And Reproducibility:**

The novelty of this article is that it studied the adaptive bilevel methods for the nonconvex-strongly-convex bilevel optimization and provided solid theoretical analysis. This article is well-organized and easy to understand.

**Strength And Weaknesses:**

Strength:

This paper proposed efficient adaptive bilevel optimization methods based on momentum techniques for the nonconvex-strongly-convex bilevel optimization. It provided the solid convergence analysis for the proposed methods, and proved that these methods obtain the best-known complexity.

Weakness:

The parameters in adaptive matrices maybe depend on the tuning parameter in the proposed algorithms.


**Summary Of The Paper:**

This paper proposed efficient adaptive bilevel optimization methods based on the momentum techniques for the nonconvex-strongly-convex bilevel optimization. It provided the solid convergence analysis for the proposed methods, and proved that these methods obtain the best-known complexity. The experimental results on data hyper-cleaning and hyper-representation learning tasks demonstrate the efficiency of the proposed algorithms.

**Summary Of The Review:**

This paper proposed efficient adaptive bilevel optimization methods based on momentum techniques for the nonconvex-strongly-convex bilevel optimization. It provided the solid convergence analysis for the proposed methods, and proved that these methods obtain the best-known complexity. The experimental results on data hyper-cleaning and hyper-representation learning tasks demonstrate the efficiency of the proposed algorithms.

Some Comments:

1. It would be great if the authors would write how they can solve the subproblems in lines 5 and 6 of Algorithms 1-2.

2. What is the intuition to use the momentum steps in the lines 5-6 of Algorithms 1-2?

3. These momentum steps improve the sample complexity in the proposed algorithms?

4. In Theorem 6, the term $\sqrt{\frac{1}{T}\sum_{t=1}^T\mathbb{E}\|A_t\|^2}$ is bounded?

5. How to choose the adaptive learning rates of Algorithms 1-2 in the experiments?

---

> ### Author Response · Authors · 2022-11-09
> **Responses to Reviewer 5EBw**
>
> Thanks so much for your positive comments.
>
> **R1:** When $\mathcal{X}=\mathbb{R}^d$ and $\mathcal{Y}=\mathbb{R}^p$, i.e., the subproblems are **unconstrained** optimizations, so we have
> their closed-form solutions. When $\mathcal{X}\subset \mathbb{R}^d$ and $\mathcal{Y}\subset \mathbb{R}^p$, i.e.,
> the subproblems are **constrained** optimizations, since the adaptive matrices are diagonal matrices,
> we also have their closed-form solutions under some specific constrained sets such as $||x||_1 \leq r$.
>
> **R2:** Our momentum steps in the lines 5-6 not only accelerate our algorithms, but also let our algorithms easily
> use the unified adaptive matrices to flexibly incorporate various adaptive
> learning rates to update variables in both UL and LL problems.
>
> **R3:** From our convergence analysis, this momentum steps do not improve the sample complexity in the proposed algorithms,
> but let algorithms converge faster in practice.
>
> **R4:** Yes, this term is bounded if give the same conditions such as bounded stochastic gradients in the existing adaptive methods
> such as Adam (Please see the Remark 3 at the page 15 of our paper).
>
> **R5:** In our algorithms, we choose the parameters $\eta_t$, $\alpha_t$ and $\beta_t$ based on the given forms in the convergence analysis.  Specifically, in our BiAdam algorithm, we use $\eta_t = \frac{k}{(m+t)^{1/2}}$,
> $\alpha_{t+1}=c_1\eta_t$ and $\beta_{t+1}=c_2\eta_t$. In our VR-BiAdam algorithm, we use $\eta_t = \frac{k}{(m+t)^{1/3}}$,
> $\alpha_{t+1}=c_1\eta^2_t$ and $\beta_{t+1}=c_2\eta^2_t$. To further choose the tuning parameters in these parameters,
> we use the grid search to find the optimal tuning parameters $k,m,c_1,c_2$. **NOTE THAT**: in the experiments, good performances of our algorithms  are robust to these tuning parameters $k,m,c_1,c_2$.

---

### Official Review · Reviewer_7cUX · 2022-10-25

**Confidence:** 2
**Clarity, Quality, Novelty And Reproducibility:** Paper is clear, and the idea seems new
**Correctness:** 4
**Technical Novelty And Significance:** 3
**Empirical Novelty And Significance:** 2
**Recommendation:** 5

**Strength And Weaknesses:**

Major concerns:
- I have doubt about the practical impact of the proposed work, as many bilevel optimization problems, it requires to select $3$ sequences of stepsize $\eta_t, \alpha_t, \beta_t$. To me, it seems very unclear how to select these parameters in practice.
- I also have doubt about the theoretical impact of the paper "Our VR-BiAdam algorithm reaches the best known sample complexity of $1/\epsilon^3$". A recent paper [1] managed to prove $1/\epsilon^2$ rate for a single loop algorithm. Could authors comment on this?

[1] Dagréou, M., Ablin, P., Vaiter, S. and Moreau, T., 2022. A framework for bilevel optimization that enables stochastic and global variance reduction algorithms. NeurIPS2022


**Summary Of The Paper:**

Inspired from Adam, authors propose adaptive bilevel algorithms, and a variance reduced variation, coined "biAdam" and VR-BiAdam. They show respectively $1/\epsilon^4$ and $1/\epsilon^3$ convergence rate. Authors propose experiments on data hypercleaning and hyperrepresentation learning


**Summary Of The Review:**

I have doubts on the practical and theoretical impact of the paper

---

> ### Author Response · Authors · 2022-11-09
> **Responses to Reviewer 7cUX**
>
> **C1:** I have doubt about the practical impact of the proposed work, as many bilevel optimization problems,...
>
> **R1:** Thanks for your comment.
> In our algorithms, we choose the parameters $\eta_t$, $\alpha_t$ and $\beta_t$ based on the given forms in the convergence analysis.
> Specifically, in our BiAdam algorithm, we use $\eta_t = \frac{k}{(m+t)^{1/2}}$,
> $\alpha_{t+1}=c_1\eta_t$ and $\beta_{t+1}=c_2\eta_t$. In our VR-BiAdam algorithm, we use $\eta_t = \frac{k}{(m+t)^{1/3}}$,
> $\alpha_{t+1}=c_1\eta^2_t$ and $\beta_{t+1}=c_2\eta^2_t$. To further choose the tuning parameters in these parameters,
> we use the grid search to find the optimal tuning parameters $k,m,c_1,c_2$. **NOTE THAT**: in the experiments, good performances of our algorithms are robust to these tuning parameters $k,m,c_1,c_2$.
>
> --------------------------------------------------------------------------------------------------------
> ---------------------------------------------------------------------------------------------------------------
>
>
> **C2:** I also have doubt about the theoretical impact of the paper "Our VR-BiAdam algorithm reaches the best known sample complexity of ...
>
> **R2:** Thanks for your comment. our VR-BiAdam has the best known sample (gradient) complexity of $O(\epsilon^{-3})$
> for finding an $\epsilon$-stationary point (i.e., $\mathbb{E}||\nabla F(x)||\leq \epsilon$
>   or its equivalent variants), which reaches the near-optimal gradient complexity of $O(\epsilon^{-3})$ in finding an $\epsilon$-stationary point of nonconvex stochastic optimization **[a]**. You said "A recent paper [1] managed to prove $1/\epsilon^2$ rate for a single loop algorithm."
> Sorry, I can not find the paper [1]. I think this $1/\epsilon^2$ is a convergence rate, not a sample (or gradient) complexity,
> since [a] prove the gradient complexity of $O(\epsilon^{-3})$ is a lower bound for for nonconvex **stochastic**
> optimization. In addition, this $1/\epsilon^2$ may be sample (or gradient) complexity for **deterministic** optimization.
>
> **[a]** Y. Arjevani, Y. Carmon, J. C. Duchi, D. J. Foster, N. Srebro, and B. Woodworth. Lower bounds for
> non-convex stochastic optimization. arXiv preprint arXiv:1912.02365, 2019.

---

> > ### Comment · Reviewer_7cUX · 2022-11-14
> > **Reference**
> >
> > My bad, I was thinking to this paper (I edited my response)
> >
> > [1] Dagréou, M., Ablin, P., Vaiter, S. and Moreau, T., 2022. A framework for bilevel optimization that enables stochastic and global variance reduction algorithms. NeurIPS2022
> >
> > For "matching" the lower bound, I am not sure that this is meaningful, because this is not a general non-convex problems, it has a lot of structure (the strongly convex inner problem for instance). Could you comment on this?

---

> > > ### Author Response · Authors · 2022-11-16
> > > **Responses to Reviewer 7cUX  (2.2)**
> > >
> > > **C2:** For "matching" the lower bound, I am not sure that this is meaningful,
> > > because this is not a general non-convex problems, it has a lot of structure (the strongly convex inner problem for instance).
> > > Could you comment on this?
> > >
> > > **R2:** Thanks for your comment. In fact, in the problem (1) of our paper, we only assume the **outer objective function**
> > >
> > > $F(x)=f(x,y^*(x))=\mathbb{E}_{\xi}\big[f(x,y^*(x);\xi)\big]$
> > >
> > > is a general nonconvex function **without some specific structures**, and assume the **inner objective function**
> > > $g(x,y)=\mathbb{E}_{\zeta}\big[g(x,y;\zeta)\big]$
> > > is a differentiable and strongly convex function in variable $y$, which is common in many bilevel optimization methods including the SABA algorithm in [1].
> > >
> > > **NOTE THAT** the single-level problem $\min_x f(x)=\mathbb{E}_{\xi}\big[f(x;\xi)\big]$ can be seen as **a specific case** of the bilevel problem (1) in our paper.
> > > For example, $F(x)=f(x,y^*(x))=af(x)+b$, where $a>0$ and $b\geq 0$ are constants, i.e., here $y^*(x)=c$ is independent on variable $x$, where $c$ is a constant.
> > >
> > > **[2]** shows the stochastic algorithms in solving the single-level **nonconvex stochastic** problem
> > >
> > > $\min_x f(x)=\mathbb{E}_{\xi}\big[f(x;\xi)\big]$
> > >
> > > has a lower bound complexity $O(\epsilon^{-3})$ for finding an $\epsilon$-stationary point (i.e., $\mathbb{E}||\nabla f(x)||\leq \epsilon$).
> > > Since the single-level problem $\min_x f(x)=\mathbb{E}_{\xi}\big[f(x;\xi)\big]$ can be seen as a specific case of the bilevel problem (1) of our paper,
> > > the stochastic algorithms in solving the bilevel stochastic problem (1) of our paper
> > > **also has a lower bound complexity $O(\epsilon^{-3})$** for finding an $\epsilon$-stationary point (i.e., $\mathbb{E}||\nabla F(x)||\leq \epsilon$).
> > >
> > > **[2]** Y. Arjevani, Y. Carmon, J. C. Duchi, D. J. Foster, N. Srebro, and B. Woodworth.
> > > Lower bounds for non-convex stochastic optimization. arXiv preprint arXiv:1912.02365, 2019.
> > >
> > > From our **Theorem 2** (at page 8 of our paper) and **Theorem 4** (at page 14 of our paper), we can find that
> > > our **VR-BiAdam** algorithm has a sample complexity $\tilde{O}(\epsilon^{-3})$ for finding an $\epsilon$-stationary point of
> > > stochastic bilevel problem (1) of our paper, i.e.,
> > > to **constrained** optimization ($\mathcal{X}\subset\mathbb{R}^{d}$), from **our Theorem 2**,
> > >
> > > $\frac{1}{T} \sum_{t=1}^T \mathbb{E}||\mathcal{G}_\mathcal{X}(x_t,\nabla F(x_t),\gamma)|| \leq
> > > \tilde{O}(\frac{1}{T^{1/3}})= \epsilon$,
> > >
> > > to **unconstrained** optimization ($\mathcal{X}=\mathbb{R}^{d}$), from **our Theorem 4**, we have
> > >
> > > $\frac{1}{T}\sum_{t=1}^T\mathbb{E}||\nabla F(x_t)||  \leq \tilde{O}(\frac{1}{T^{1/3}})= \epsilon$.
> > >
> > > From **Theorems 1-2** of **[1]** (at the page 7 of [1] https://arxiv.org/pdf/2201.13409.pdf), we can find that the
> > >  **SOBA** algorithm of [1] has a sample complexity $O(\epsilon^{-4})$ for finding an $\epsilon$-stationary point of
> > > finite-sum bilevel **problem (1) of [1]**, i.e.,
> > > $\frac{1}{T}\sum_{t=1}^T\mathbb{E}||\nabla h(x_t)||\leq \epsilon$, where $h(x)$ is the **outer objective function** of the finite-sum
> > > bilevel problem (1) of [1].
> > > **NOTE THAT** in **Theorem 1 of [1]**, it shows
> > >
> > > $\frac{1}{T}\sum_{t=1}^T\mathbb{E}||\nabla h(x_t)||^2=O(T^{-\frac{1}{2}})$,
> > >
> > > which **is equivalent to** the following form
> > >
> > > $\frac{1}{T}\sum_{t=1}^T\mathbb{E}||\nabla h(x_t)||=O(T^{-\frac{1}{4}})= \epsilon$.
> > >
> > > In **Theorem 2 of [1]**, it shows
> > >
> > > $\inf_{t\leq T}\mathbb{E}||\nabla h(x_t)||^2=O(\log(T)T^{-\frac{1}{2}})=\tilde{O}(T^{-\frac{1}{2}})$,
> > >
> > > which is equivalent to the following form
> > >
> > > $\inf_{t\leq T}\sum_{t=1}^T\mathbb{E}||\nabla h(x_t)||=\tilde{O}(T^{-\frac{1}{4}}) = \epsilon$.
> > >
> > >
> > > From **Theorem 3** of **[1]**(at the page 7 of [1] https://arxiv.org/pdf/2201.13409.pdf), we can find that the
> > >  **SABA** algorithm of [1] has a sample complexity $O(N^{2/3}\epsilon^{-2})$ for finding an $\epsilon$-stationary point of
> > > finite-sum bilevel problem (1) of [1], i.e.,
> > > $\frac{1}{T}\sum_{t=1}^T\mathbb{E}||\nabla h(x_t)||\leq \epsilon$.
> > >  **NOTE THAT** in **Theorem 3 of [1]**, it shows
> > >
> > > $\frac{1}{T}\sum_{t=1}^T\mathbb{E}||\nabla h(x_t)||^2=O(N^{2/3}T^{-1})$,
> > >
> > > which is **equivalent to** the following form
> > >
> > > $\frac{1}{T}\sum_{t=1}^T\mathbb{E}||\nabla h(x_t)||=O(\frac{N^{1/3}}{T^{1/2}})= \epsilon$.
> > >
> > > In **big data setting, when $N\gg \epsilon^{-3/2}$**,
> > > clearly, our **VR-BiAdam** algorithm has a **lower** sample complexity $\tilde{O}(\epsilon^{-3})$
> > > than the convergence rate $O(N^{2/3}\epsilon^{-2})$
> > > of **SABA** algorithm of [1]. **When $N = O(\epsilon^{-3/2})$**, the sample complexity $\tilde{O}(\epsilon^{-3})$
> > > of our VR-BiAdam algorithm is **similar** to the sample complexity $O(N^{2/3}\epsilon^{-2})$
> > > of SABA algorithm of [1]. In **small data setting, when $N \ll \epsilon^{-3/2}$**, our VR-BiAdam algorithm has
> > > a **higher** sample complexity $\tilde{O}(\epsilon^{-3})$
> > > than the sample complexity $O(N^{2/3}\epsilon^{-2})$
> > > of SABA algorithm of [1].

---

> > > > ### Author Response · Authors · 2022-11-16
> > > > **Responses to Reviewer 7cUX (2.3)**
> > > >
> > > > **NOTE THAT** the **SABA** algorithm of [1] focuses on solving the **finite-sum** bilevel problem, where $N \ (0<N<+\infty)$ denotes all sample size.
> > > > While our BiAdam and VR-BiAdam algorithms focus on solving the **stochastic** bilevel problem,
> > > > where all sample size $N=+\infty$.
> > > > Clearly, **our paper and [2]** mainly consider the stochastic optimization in the **big data setting**, i.e., $N\gg \epsilon^{-3/2}$.
> > > >
> > > > Since the **SABA** algorithm builds on the **SAGA** requiring
> > > > to compute the full gradient at the first step and store a gradient table on all samples, clearly, the **SABA** algorithm
> > > > can not solve the **stochastic** bilevel problem (1) in our paper, where all sample size $N=+\infty$.
> > > > In summary, **for fair and meaningful comparison, we need to use the same or equivalent convergence metric
> > > > under the same or equivalent conditions in solving the same problems**.
> > > >
> > > > In addition, in **small data setting, i.e., $N \ll \epsilon^{-3/2}$**, the sample complexity $O(N^{2/3}\epsilon^{-2})$
> > > > of SABA algorithm of [1] is lower than
> > > > than the sample complexity $\tilde{O}(\epsilon^{-3})$ of our VR-BiAdam algorithm. Clearly, this is a meaningless.
> > > > In the **small data setting, i.e., $N \ll \epsilon^{-3/2}$**,
> > > > we can directly use the **deterministic** gradient algorithm to solve this finite-sum problem **without storing the
> > > > gradient table on all samples**.

---

> > > ### Author Response · Authors · 2022-11-16
> > > **Responses to Reviewer 7cUX (2.1)**
> > >
> > > Thanks so much for your reply.
> > >
> > > **Thank you for introducing this interesting and good paper [1] to me**. In the new version of our paper,
> > > we have introduced this paper in the introduction and related work sections, **E.g.**, "More recently, Dagr ́eou et al. (2022) developed a novel framework for bilevel optimization based on the linear system, and proposed a fast SABA algorithm
> > > for finite-sum bilevel problems based on the variance reduced technique of SAGA (Defazio et al.,2014)."
> > >
> > >
> > > **C1:** I also have doubt about the theoretical impact of the paper "Our VR-BiAdam algorithm reaches the
> > > best known sample complexity of $1/\epsilon^3$". A recent paper [1] managed to prove $1/\epsilon^2$ rate for a single loop algorithm.
> > > Could authors comment on this?
> > >
> > > **R1:** From our **Theorem 2** (at page 8 of our paper) and **Theorem 4** (at page 14 of our paper), we can find that
> > > our VR-BiAdam algorithm has a convergence rate $\tilde{O}(\frac{1}{T^{1/3}})$, where $T$ is the iteration number.
> > > To **constrained** optimization ($\mathcal{X}\subset\mathbb{R}^{d}$), from our **Theorem 2**, we have
> > >
> > > $\frac{1}{T} \sum_{t=1}^T \mathbb{E}||\mathcal{G}_\mathcal{X}(x_t,\nabla F(x_t),\gamma)|| \leq
> > > \tilde{O}(\frac{1}{T^{1/3}}),$
> > >
> > > to **unconstrained** optimization ($\mathcal{X}=\mathbb{R}^{d}$), from our **Theorem 4**, we have
> > >
> > > $\frac{1}{T}\sum_{t=1}^T\mathbb{E}||\nabla F(x_t)||  \leq \tilde{O}(\frac{1}{T^{1/3}}).$
> > >
> > > **NOTE THAT for fair comparison, we need to use the same or equivalent convergence metric**.
> > > From **Theorems 1-2** of **[1]** (at the page 7 of [1] https://arxiv.org/pdf/2201.13409.pdf), we can find that the
> > >  **SOBA** algorithm of [1] has a convergence rate $O(\frac{1}{T^{1/4}})$ based on the metric
> > > $\frac{1}{T}\sum_{t=1}^T\mathbb{E}||\nabla h(x_t)||$,
> > > where $T$ is the iteration number. **NOTE THAT** in **Theorem 1** of **[1]**, it shows
> > >
> > > $\frac{1}{T}\sum_{t=1}^T\mathbb{E}||\nabla h(x_t)||^2=O(T^{-\frac{1}{2}})$,
> > >
> > > which is **equivalent to** the following form
> > >
> > > $\frac{1}{T}\sum_{t=1}^T\mathbb{E}||\nabla h(x_t)||=O(T^{-\frac{1}{4}})$.
> > >
> > > In **Theorem 2** of **[1]**, it shows
> > >
> > > $\inf_{t\leq T}\mathbb{E}||\nabla h(x_t)||^2=O(\log(T)T^{-\frac{1}{2}})=\tilde{O}(T^{-\frac{1}{2}})$,
> > >
> > > which is **equivalent to** the following form
> > >
> > > $\inf_{t\leq T}\sum_{t=1}^T\mathbb{E}||\nabla h(x_t)||=\tilde{O}(T^{-\frac{1}{4}})$.
> > >
> > > Meanwhile, $\inf_{t\leq T}\sum_{t=1}^T\mathbb{E}||\nabla h(x_t)||\leq \frac{1}{T}\sum_{t=1}^T\mathbb{E}||\nabla h(x_t)||$.
> > > Thus, our VR-BiAdam algorithm has a faster convergence rate $\tilde{O}(T^{-\frac{1}{3}})$
> > > than the convergence rate $\tilde{O}(T^{-\frac{1}{4}})$
> > > of SOBA algorithm of [1].
> > >
> > > From **Theorem 3** of **[1]** (at the page 7 of [1] https://arxiv.org/pdf/2201.13409.pdf), we can find that the
> > >  **SABA** algorithm of [1] has a convergence rate $O(\frac{N^{1/3}}{T^{1/2}})$ based on the metric
> > > $\frac{1}{T}\sum_{t=1}^T\mathbb{E}||\nabla h(x_t)||$, where $T$ is the iteration number.
> > >  **NOTE THAT** in Theorem 3 of [1], it shows
> > >
> > > $\frac{1}{T}\sum_{t=1}^T\mathbb{E}||\nabla h(x_t)||^2=O(N^{2/3}T^{-1})$,
> > >
> > > which is **equivalent to** the following form
> > >
> > > $\frac{1}{T}\sum_{t=1}^T\mathbb{E}||\nabla h(x_t)||=O(N^{1/3}T^{-1/2})$.
> > >
> > > **NOTE THAT** the **SABA** algorithm of [1] only focuses on solving the **finite-sum** bilevel problem. Here $N=n+m$
> > > denotes all sample size. In **big data setting, when $N\gg T^{1/2}$**,
> > > clearly, our **VR-BiAdam** algorithm has a faster convergence rate $\tilde{O}(\frac{1}{T^{1/3}})$
> > > than the convergence rate $O(\frac{N^{1/3}}{T^{1/2}})$
> > > of **SABA** algorithm of [1]. **When $N = O(T^{1/2})$**, the convergence rate $\tilde{O}(\frac{1}{T^{1/3}})$
> > > of our VR-BiAdam algorithm is similar to the convergence rate $O(\frac{N^{1/3}}{T^{1/2}})$
> > > of SABA algorithm of [1]. In **small data setting, when $N \ll T^{1/2}$**, our VR-BiAdam algorithm has
> > > a slower convergence rate $\tilde{O}(\frac{1}{T^{1/3}})$
> > > than the convergence rate $O(\frac{N^{1/3}}{T^{1/2}})$
> > > of SABA algorithm of [1].
> > >
> > > However, our BiAdam and VR-BiAdam algorithms focuses on solving the **stochastic** bilevel problem (1) in our paper,
> > > where all sample size $N=+\infty$.
> > > Clearly, **our paper and [2]** mainly consider the stochastic optimization in the big data setting, e.g., $N\gg T^{1/2}$.
> > > Meanwhile, since the **SABA** algorithm builds on the **SAGA** requiring
> > > to compute full gradient at the first step and store a gradient table on all samples, clearly, the SABA algorithm
> > > can not solve the **stochastic** bilevel problem (1) in our paper, where all sample size $N=+\infty$.
> > >
> > > In our paper, we only consider the **general nonconvex** condition on function $F(x)$, and
> > > do not consider the PL-assumption. Thus, we only compare our **VR-BiAdam** algorithm with the **SOBA** and
> > > **SABA** algorithms of **[1]** without the PL-assumption on the function $h(x)$.
> > > In summary, **for fair comparison, we need to use the same or equivalent convergence metric
> > > under the same or equivalent conditions in solving the same problems**.

---

### Official Review · Reviewer_LXpt · 2022-10-26

**Confidence:** 3
**Correctness:** 3
**Technical Novelty And Significance:** 2
**Empirical Novelty And Significance:** 2
**Recommendation:** 3

**Clarity, Quality, Novelty And Reproducibility:**

clarity: typo in the abstract, "mate learning" -> "meta learning.



**Strength And Weaknesses:**

strengths:


a) The complexities matches the previous results, and the algorithm does not need a mini-batch


b) The algorithm can be used in the setting with constraints.


weaknesses:


a) The novelty in this paper is quite limited. The results are built on the large body of bi-level optimization literature and super-ADAM. Noticeable, there are already many works for bi-level optimization can achieve the same complexities.

b) The adaptive framework is not very meaningful here. Note that all the adaptivity is absorbed in the matrices $A_t$ and $B_t$ in the algorithms. The stepsize can be considered to be inversely related with eigenvalues of matrices. In Assumption 7, it assumes $A_t \succeq \rho I_d$ and $B_t=b I_p\left(b_u \geq b \geq b_l>0\right)$, which basically indicates stepsize for $x$ is upper bounded and stepsize for $y$ is lower and upper bounded. Then by controlling the other parameters in the stepsizes, e.g., $\lambda$ and $\gamma$, the stepsizes  can be treated just like non-adaptive stepsizes. Therefore, the analysis does not need too much novelty to accommodate adaptive stepsize, e.g., Lemma 10.  This leads to some drawbacks: 1) upper and lower bounds for adaptive matrices $A_t$ and $B_t$ need to be know in order to pick other hyperparameters, 2) usually, $\rho$ in Adam is chosen to be very small just for stability. But when $\rho$ is small, Theorem 1 needs a very small $\gamma$ which may reduce the gain from Adam stepsize.

**Summary Of The Paper:**

This work focus on adaptive frameworks for bilevel optimization. It first introduce BiAdam that achieves $\tilde{O}(\epsilon^{-4})$ complexity. Then it introduces BiAdam with variance reduction and achieves $\tilde{O}(\epsilon^{-3})$ complexity. Both results matches  state-of-the-art complexities.

**Summary Of The Review:**

The results in this paper is reasonable, but it may lack novelty.

---

> ### Author Response · Authors · 2022-11-09
> **(1) Responses to Reviewer LXpt**
>
> **C1:** The novelty in this paper is quite limited....
>
> **R1:** Thanks for your comments. **Our contributions and novelties of our paper** are given as follows:
>
> **(1)**. To the best of our knowledge, our BiAdam and VR-BiAdam methods are the first work to study
> the adaptive bilevel optimization methods with adaptive learning rates based on the momentum techniques.
> **In particular, our algorithms use the the momentum techniques to update variables and estimate stochastic gradients simultaneously.**
> For example, our BiAdam uses the **momentum iteration**  (i.e.,
>
> $x_{t+1} = x_t + \eta_t( \tilde{x}_{t+1} -x_t)$
>
> at the line 5 of our Algorithm 1)  to update variable $x$,  and applies the basic momentum technique to estimate the stochastic gradient $w_t$ (i.e., the line 9 of our Algorithm 1).  Our VR-BiAdam also uses the **momentum iteration** (i.e.,
>
> $x_{t+1} = x_t + \eta_t( \tilde{x}_{t+1} -x_t)$
>
> at the line 5 of our Algorithm 2) to update variable $x$,
> and applies the variance-reduced momentum technique of STROM to estimate the stochastic gradient $w_t$  (i.e., the line 9 of our Algorithm 2).
>
> **(2)**. We provide a solid convergence analysis framework for our BiAdam and VR-BiAdam methods.
> Specifically, we prove that our BiAdam has a sample (gradient) complexity of $O(\epsilon^{-4})$ for finding an $\epsilon$-stationary point.
> Meanwhile, we prove that our VR-BiAdam has a sample (gradient) complexity of $O(\epsilon^{-3})$,
> which reaches the near-optimal gradient complexity of $O(\epsilon^{-3})$ in finding an $\epsilon$-stationary point of
> nonconvex stochastic optimization **[a]**.
>
> **[a]** Y. Arjevani, Y. Carmon, J. C. Duchi, D. J. Foster, N. Srebro, and B. Woodworth. Lower bounds for
> non-convex stochastic optimization. arXiv preprint arXiv:1912.02365, 2019.
>
> **NOTE THAT**: **1)** our methods (i.e., BiAdam and VR-BiAdam) are **totally different from the most related
> momentum-based bilevel optimization methods** (i.,e, MRBO **[b]** and SUSTAIN **[c]**).
>
> **First**, both the MRBO (https://proceedings.neurips.cc/paper/2021/file/71cc107d2e0408e60a3d3c44f47507bd-Paper.pdf)
> and SUSTAIN (https://proceedings.neurips.cc/paper/2021/file/fe2b421b8b5f0e7c355ace66a9fe0206-Paper.pdf)
> algorithms only use the variance-reduced momentum technique of STROM to estimate the stochastic gradients, **while our algorithms use the the momentum techniques to update variables and estimate stochastic gradients simultaneously.**
> For example, our BiAdam uses the **momentum iteration** (i.e.,
>
> $x_{t+1}=x_t + \eta_t(\tilde{x}_{t+1}-x_t)$
>
> at the line 5 of our Algorithm 1)  to update variable $x$,
> and applies the basic momentum technique to estimate the stochastic gradient $w_t$ (i.e., the line 9 of our Algorithm 1).
> Our VR-BiAdam also uses the **momentum iteration** (i.e.,
>
> $x_{t+1}=x_t + \eta_t(\tilde{x}_{t+1}-x_t)$
>
> at the line 5 of our Algorithm 2) to update variable $x$,
> and applies the variance-reduced momentum technique of STROM to estimate the stochastic gradient $w_t$
> (i.e., the line 9 of our Algorithm 2).
>
> **Second**, the MRBO and SUSTAIN  algorithms and other bilevel optimization algorithms
> only focus on the **unconstrained optimization** in both lower level and upper level subproblems, while our algorithms focus on
>  the **constrained optimization** in both lower level and upper level subproblems.
> Clearly, there exist some significant differences between
> our algorithms and these existing algorithm in convergence analysis. For example, in our convergence analysis,
> we use a new convergence metric $\mathcal{G}_t $
>
> $  = \frac{1}{\gamma} \|\tilde{x}_{t+1} - x_t\| + \frac{1}{\rho} \Big( \sqrt{2} \|w_t - \bar{\nabla} f(x_t,y_t)\| + L_0 \|y^*(x_t)-y_t\| \Big)$
>
> and prove it is a upper bound of  the standard convergence metric
>
> $||\mathcal{G}_\mathcal{X}(x_t,\nabla F(x_t),\gamma)||$
>
> for **constrained** optimization.
> Moreover,  we provide the convergence properties of our algorithms for both **constrained optimization** (in Theorems 1-2)
> and **unconstrained optimization** (in Theorems 3-4 at pages 13-14 of our paper)
> in both lower level and upper level subproblems.
>
> **Third**, in the convergence analysis, we use a novel potential function, for any $t\geq 1$,
>
> $\Gamma_t  = \mathbb{E}\big [F(x_t) + \frac{5b_tL^2_0\gamma}{\lambda\mu\rho}\|y_t-y^*(x_t)\|^2  + \frac{\gamma}{\rho} \big( \|v_t - \nabla_y g(x_t,y_t)\|^2  + \|w_t - \bar{\nabla} f(x_t,y_t) -R_t\|^2 \big) \big] $
>
> which depends on the adaptive learning rates $b_t$.
>
> **[b]** Junjie Yang, Kaiyi Ji, and Yingbin Liang. Provably faster algorithms for bilevel optimization.
> Advances in Neural Information Processing Systems, 34:13670–13682, 2021
>
> **[c]** Prashant Khanduri, Siliang Zeng, Mingyi Hong, Hoi-To Wai, Zhaoran Wang, and Zhuoran Yang. A
> near-optimal algorithm for stochastic bilevel optimization via double-momentum.
> Advances in Neural Information Processing Systems, 2021.

---

> > ### Author Response · Authors · 2022-11-09
> > **(2) Responses to Reviewer LXpt**
> >
> > **NOTE THAT**: **2)** our methods (i.e., BiAdam and VR-BiAdam) are **totally different from the existing Super-Adam [d] method**
> > (https://proceedings.neurips.cc/paper/2021/file/4be5a36cbaca8ab9d2066debfe4e65c1-Paper.pdf).
> >
> > Our methods focus on the bilevel optimization with a hierarchical structure, which Super-Adam only
> > focus on the single-level optimization. Clearly, the convergence analysis of our methods has some significant differences
> > from that of Super-Adam. For example, in our convergence analysis, we use a novel potential function, for any $t\geq 1$,
> >
> >  $\Gamma_t  = \mathbb{E} \big [ F(x_t) + \frac{5b_tL^2_0\gamma}{\lambda\mu\rho} \|y_t-y^*(x_t)\|^2 + \frac{\gamma}{\rho} \big( \|v_t - \nabla_y g(x_t,y_t)\|^2  + \|w_t - \bar{\nabla} f(x_t,y_t) -R_t\|^2 \big) \big]$
> >
> > based on a useful Lemma 10 in our paper. Moreover, for **unconstrained optimization**
> > in both lower level and upper level subproblems, we provide a more reasonable upper bound in the convergence properties
> > (in Theorems 3-4) than that of Super-Adam. Specifically, in our Theorems 3-4 (at the pages 13-14 of our paper), our upper bounds include a reasonable $\sqrt{\frac{1}{T}\sum_{t=1}^T\mathbb{E}\|A_t\|^2}$ that is a **scalar**, while in the
> > Corollaries 1-2 of **[d]**, the upper bounds include a term $\max_{1\leq t\leq T}\|H_t\|$ that is a **random variable** since $H_t$ includes
> > the information of stochastic gradients.
> >
> > **[d]** Feihu Huang, Junyi Li, and Heng Huang. Super-adam: faster and universal framework of adaptive
> > gradients. Advances in Neural Information Processing Systems, 34:9074–9085, 2021.
> >  (https://proceedings.neurips.cc/paper/2021/file/4be5a36cbaca8ab9d2066debfe4e65c1-Paper.pdf)
> >
> >
> > **NOTE THAT**: **3)** our VR-BiAdam obtains the best known (gradient) complexity of $O(\epsilon^{-3})$ (Please see Table 1 in our paper), which also reaches the near-optimal gradient complexity of $O(\epsilon^{-3})$ in finding an $\epsilon$-stationary point of
> > nonconvex stochastic optimization **[a]**. **Clearly, our adaptive methods can not obtain a lower complexity
> > than this near-optimal gradient complexity of $O(\epsilon^{-3})$**.
> >
> > It is well known that most of adaptive methods enjoy the same complexity as the non-adaptive versions but converges faster in practice.
> > Only some specific adaptive methods show a faster convergence rate (lower complexity) than the non-adaptive versions under some specific conditions such as the sparse gradient condition used in Adagrad **[e]**. We propose an adaptive gradient-based algorithm framework for bilevel optimization based on the general adaptive matrices. Meanwhile, we provide a convergence analysis framework for our algorithms. In our convergence analysis, thus, we do not assume some specific adaptive learning rates and some specific conditions such as sparse gradients.
> >
> >
> > **[a]** Y. Arjevani, Y. Carmon, J. C. Duchi, D. J. Foster, N. Srebro, and B. Woodworth. Lower bounds for
> > non-convex stochastic optimization. arXiv preprint arXiv:1912.02365, 2019.
> >
> > **[e]** Duchi J, Hazan E, Singer Y. Adaptive subgradient methods for online learning and stochastic optimization.
> > JMLR, 12(7), 2011.
> >
> > ----------------------------------------------------------------------------------------
> > --------------------------------------------------------------------------------
> >
> > **C2:** The adaptive framework is not very meaningful here.... e.g., Lemma 10. This leads to some drawbacks:
> > 1) upper and lower bounds for adaptive matrices  and  need to be know in order to pick other hyperparameters,
> > 2) usually,  in Adam is chosen to be very small just for stability. But when  is small,
> > Theorem 1 needs a very small...
> >
> > **R2:** Thanks for your comments. In fact, most of adaptive methods implicitly assume the bounded adaptive learning rates.
> > Specifically, they implicitly give the lower bounds of adaptive learning rates (Please see the page 6 of **[d]**
> > (https://proceedings.neurips.cc/paper/2021/file/4be5a36cbaca8ab9d2066debfe4e65c1-Paper.pdf) detail this case).
> > Meanwhile, they also implicitly give the upper bounds of adaptive learning rates,
> > e.g., assume the bounded stochastic gradients **[f,g,h]**.
> >
> > **NOTE THAT**: **since our algorithms use the momentum techniques to update variables and estimate stochastic gradients simultaneously, our algorithms can use a relatively large $\rho$**, e.g., we set $\rho=0.0005$ in the experiments.
> > However, Adam only uses the momentum technique to estimate stochastic gradients, so it need a relatively small $\rho=0.00000001$.
> >
> >
> > **[f]** S. J. Reddi, S. Kale, and S. Kumar. On the convergence of adam and beyond. arXiv preprint arXiv:1904.09237, 2019.
> >
> > **[g]** X. Chen, S. Liu, R. Sun, and M. Hong. On the convergence of a class of adam-type algorithms for non-convex
> > optimization. In 7th International Conference on Learning Representations (ICLR), 2019.
> >
> > **[h]** Zehao Dou and Yuanzhi Li. On the one-sided convergence of adam-type algorithms in non-convex non-concave min-max optimization.

---

### Author Response · Authors · 2022-11-09
**Overall Responses (1)**

**Our contributions and novelties of our paper** are given as follows:

**(1)**. To the best of our knowledge, our BiAdam and VR-BiAdam methods are the first work to study
the adaptive bilevel optimization methods with adaptive learning rates based on the momentum techniques.
**In particular, our algorithms use the the momentum techniques to update variables and estimate stochastic gradients simultaneously.**
For example, our BiAdam uses the **momentum iteration**  (i.e.,

$x_{t+1} = x_t + \eta_t( \tilde{x}_{t+1} -x_t)$

at the line 5 of our Algorithm 1)  to update variable $x$,  and applies the basic momentum technique to estimate the stochastic gradient $w_t$ (i.e., the line 9 of our Algorithm 1).  Our VR-BiAdam also uses the **momentum iteration** (i.e.,

$x_{t+1} = x_t + \eta_t( \tilde{x}_{t+1} -x_t)$

at the line 5 of our Algorithm 2) to update variable $x$,
and applies the variance-reduced momentum technique of STROM to estimate the stochastic gradient $w_t$  (i.e., the line 9 of our Algorithm 2).

**(2)**. We provide a solid convergence analysis framework for our BiAdam and VR-BiAdam methods.
Specifically, we prove that our BiAdam has a sample (gradient) complexity of $O(\epsilon^{-4})$ for finding an $\epsilon$-stationary point.
Meanwhile, we prove that our VR-BiAdam has a sample (gradient) complexity of $O(\epsilon^{-3})$,
which reaches the near-optimal gradient complexity of $O(\epsilon^{-3})$ in finding an $\epsilon$-stationary point of
nonconvex stochastic optimization **[a]**.

**[a]** Y. Arjevani, Y. Carmon, J. C. Duchi, D. J. Foster, N. Srebro, and B. Woodworth. Lower bounds for
non-convex stochastic optimization. arXiv preprint arXiv:1912.02365, 2019.

**NOTE THAT**: **1)** our methods (i.e., BiAdam and VR-BiAdam) are **totally different from the most related
momentum-based bilevel optimization methods** (i.,e, MRBO **[b]** and SUSTAIN **[c]**).

**First**, both the MRBO (https://proceedings.neurips.cc/paper/2021/file/71cc107d2e0408e60a3d3c44f47507bd-Paper.pdf)
and SUSTAIN (https://proceedings.neurips.cc/paper/2021/file/fe2b421b8b5f0e7c355ace66a9fe0206-Paper.pdf)
algorithms only use the variance-reduced momentum technique of STROM to estimate the stochastic gradients, **while our algorithms use the the momentum techniques to update variables and estimate stochastic gradients simultaneously.**
For example, our BiAdam uses the **momentum iteration** (i.e.,

$x_{t+1}=x_t + \eta_t(\tilde{x}_{t+1}-x_t)$

at the line 5 of our Algorithm 1)  to update variable $x$,
and applies the basic momentum technique to estimate the stochastic gradient $w_t$ (i.e., the line 9 of our Algorithm 1).
Our VR-BiAdam also uses the **momentum iteration** (i.e.,

$x_{t+1}=x_t + \eta_t(\tilde{x}_{t+1}-x_t)$

at the line 5 of our Algorithm 2) to update variable $x$,
and applies the variance-reduced momentum technique of STROM to estimate the stochastic gradient $w_t$
(i.e., the line 9 of our Algorithm 2).

**Second**, the MRBO and SUSTAIN  algorithms and other bilevel optimization algorithms
only focus on the **unconstrained optimization** in both lower level and upper level subproblems, while our algorithms focus on
 the **constrained optimization** in both lower level and upper level subproblems.
Clearly, there exist some significant differences between
our algorithms and these existing algorithm in convergence analysis. For example, in our convergence analysis,
we use a new convergence metric $\mathcal{G}_t $

$  = \frac{1}{\gamma} \|\tilde{x}_{t+1} - x_t\| + \frac{1}{\rho} \Big( \sqrt{2} \|w_t - \bar{\nabla} f(x_t,y_t)\| + L_0 \|y^*(x_t)-y_t\| \Big)$

and prove it is a upper bound of  the standard convergence metric

$||\mathcal{G}_\mathcal{X}(x_t,\nabla F(x_t),\gamma)||$

for **constrained** optimization.
Moreover,  we provide the convergence properties of our algorithms for both **constrained optimization** (in Theorems 1-2)
and **unconstrained optimization** (in Theorems 3-4 at pages 13-14 of our paper)
in both lower level and upper level subproblems.

**Third**, in the convergence analysis, we use a novel potential function, for any $t\geq 1$,

$\Gamma_t  = \mathbb{E}\big [F(x_t) + \frac{5b_tL^2_0\gamma}{\lambda\mu\rho}\|y_t-y^*(x_t)\|^2  + \frac{\gamma}{\rho} \big( \|v_t - \nabla_y g(x_t,y_t)\|^2  + \|w_t - \bar{\nabla} f(x_t,y_t) -R_t\|^2 \big) \big] $

which depends on the adaptive learning rates $b_t$.

**[b]** Junjie Yang, Kaiyi Ji, and Yingbin Liang. Provably faster algorithms for bilevel optimization.
Advances in Neural Information Processing Systems, 34:13670–13682, 2021

**[c]** Prashant Khanduri, Siliang Zeng, Mingyi Hong, Hoi-To Wai, Zhaoran Wang, and Zhuoran Yang. A
near-optimal algorithm for stochastic bilevel optimization via double-momentum.
Advances in Neural Information Processing Systems, 2021.

---

> ### Author Response · Authors · 2022-11-09
> **Overall Responses (2)**
>
> **NOTE THAT**: **2)** our methods (i.e., BiAdam and VR-BiAdam) are **totally different from the existing Super-Adam [d] method**
> (https://proceedings.neurips.cc/paper/2021/file/4be5a36cbaca8ab9d2066debfe4e65c1-Paper.pdf).
>
> Our methods focus on the bilevel optimization with a hierarchical structure, which Super-Adam only
> focus on the single-level optimization. Clearly, the convergence analysis of our methods has some significant differences
> from that of Super-Adam. For example, in our convergence analysis, we use a novel potential function, for any $t\geq 1$,
>
>  $\Gamma_t  = \mathbb{E} \big [ F(x_t) + \frac{5b_tL^2_0\gamma}{\lambda\mu\rho} \|y_t-y^*(x_t)\|^2 + \frac{\gamma}{\rho} \big( \|v_t - \nabla_y g(x_t,y_t)\|^2  + \|w_t - \bar{\nabla} f(x_t,y_t) -R_t\|^2 \big) \big]$
>
> based on a useful Lemma 10 in our paper. Moreover, for **unconstrained optimization**
> in both lower level and upper level subproblems, we provide a more reasonable upper bound in the convergence properties
> (in Theorems 3-4) than that of Super-Adam. Specifically, in our Theorems 3-4 (at the pages 13-14 of our paper), our upper bounds include a reasonable $\sqrt{\frac{1}{T}\sum_{t=1}^T\mathbb{E}\|A_t\|^2}$ that is a **scalar**, while in the
> Corollaries 1-2 of **[d]**, the upper bounds include a term $\max_{1\leq t\leq T}\|H_t\|$ that is a **random variable** since $H_t$ includes
> the information of stochastic gradients.
>
> **[d]** Feihu Huang, Junyi Li, and Heng Huang. Super-adam: faster and universal framework of adaptive
> gradients. Advances in Neural Information Processing Systems, 34:9074–9085, 2021.
>  (https://proceedings.neurips.cc/paper/2021/file/4be5a36cbaca8ab9d2066debfe4e65c1-Paper.pdf)
>
>
> **NOTE THAT**: **3)** our VR-BiAdam obtains the best known (gradient) complexity of $O(\epsilon^{-3})$ (Please see Table 1 in our paper),  which reaches the near-optimal gradient complexity of $O(\epsilon^{-3})$ in finding an $\epsilon$-stationary point of
> nonconvex stochastic optimization **[a]**. **Clearly, our adaptive methods can not obtain a lower complexity
> than this near-optimal gradient complexity of $O(\epsilon^{-3})$**.
>
> In fact, in the problem (1) of our paper, we only assume the **outer objective function**
>
> $F(x)=f(x,y^*(x))=\mathbb{E}_{\xi}\big[f(x,y^*(x);\xi)\big]$
>
> is a general nonconvex function **without some specific structures**, and assume the **inner objective function**
> $g(x,y)=\mathbb{E}_{\zeta}\big[g(x,y;\zeta)\big]$
> is a differentiable and strongly convex function in variable $y$, which is common in many bilevel optimization methods.
>
> **NOTE THAT** the single-level problem $\min_x f(x)=\mathbb{E}_{\xi}\big[f(x;\xi)\big]$ can be seen as **a specific case** of the bilevel problem (1) in our paper.
> For example, $F(x)=f(x,y^*(x))=af(x)+b$, where $a>0$ and $b\geq 0$ are constants, i.e., here $y^*(x)=c$ is independent on variable $x$, where $c$ is a constant.
>
> **[a]** shows the stochastic algorithms in solving the single-level **nonconvex stochastic** problem
>
> $\min_x f(x)=\mathbb{E}_{\xi}\big[f(x;\xi)\big]$
>
> has a lower bound complexity $O(\epsilon^{-3})$ for finding an $\epsilon$-stationary point (i.e., $\mathbb{E}||\nabla f(x)||\leq \epsilon$).
> Since the single-level problem $\min_x f(x)=\mathbb{E}_{\xi}\big[f(x;\xi)\big]$ can be seen as a specific case of the bilevel problem (1) of our paper,
> the stochastic algorithms in solving the bilevel stochastic problem (1) of our paper
> **also has a lower bound complexity $O(\epsilon^{-3})$** for finding an $\epsilon$-stationary point (i.e., $\mathbb{E}||\nabla F(x)||\leq \epsilon$).
>
> It is well known that most of adaptive methods enjoy the same complexity as the non-adaptive versions but converges faster in practice.
> Only some specific adaptive methods show a faster convergence rate (lower complexity) than the non-adaptive versions under some specific conditions such as the sparse gradient condition used in Adagrad **[e]**. We propose an adaptive gradient-based algorithm framework for bilevel optimization based on the general adaptive matrices. Meanwhile, we provide a convergence analysis framework for our algorithms. In our convergence analysis, thus, we do not assume some specific adaptive learning rates and some specific conditions such as sparse gradients.
>
>
> **[a]** Y. Arjevani, Y. Carmon, J. C. Duchi, D. J. Foster, N. Srebro, and B. Woodworth. Lower bounds for
> non-convex stochastic optimization. arXiv preprint arXiv:1912.02365, 2019.
>
> **[e]** Duchi J, Hazan E, Singer Y. Adaptive subgradient methods for online learning and stochastic optimization.
> JMLR, 12(7), 2011.

---

### Author Response · Authors · 2022-11-22
**We are very glad to continue discussion with you.**

Dear Reviewers,

Thanks a lot for your reviews.
We are very glad to continue discussion with you.
If you still have any problems, please let us know. Thanks.

Best wishes,

Authors

---

### Decision · Program_Chairs · 2023-01-20

**Decision:**

Reject

**Justification For Why Not Higher Score:**

The novelty and practical applicability of the algorithm is limited.

**Justification For Why Not Lower Score:**

N/A

**Metareview: Summary, Strengths And Weaknesses:**

The paper studies stochastic bilevel optimization problems where the outer objective function is possibly non-convex and the inner objective function is strongly convex. Prior works only consider the unconstrained setting and propose algorithms with non-adaptive step sizes. The paper introduces an algorithmic framework based on Adam that leverages adaptive step sizes and it supports general constraints. The core algorithm is then combined with momentum-based variance reduction. The convergence guarantees for the core algorithm and its variance-reduced counterpart match the best-known convergence guarantees.

The reviewers' opinions were divided, with two reviewers being positive about the paper and supporting accept and two reviewers being concerned about the novelty of the contribution and its practical applicability. The ability to handle general constraints was appreciated by the reviewers. The main concerns about the novelty were that there already exist several works for bilevel optimization that attain the same convergence guarantee and, although incorporating adaptive step sizes into the framework is a valuable contribution, it is fairly standard and straightforward to do. The main concerns about the practical applicability were that the algorithm has many parameters that need to be tuned and assumes knowledge of suitable upper and lower bounds on the eigenvalues of the adaptive matrices.

**Summary Of Ac-Reviewer Meeting:**

We discussed the strengths of the paper's contribution and its practical applicability. After the discussion, the consensus among the attending reviewers was that the novelty and practical applicability of the algorithm are limited, and the main reasons are summarized above. The reviewers that were positive about the paper did not attend the meeting.